# CONTEXTUAL BANDITS WITH CONCAVE REWARDS, AND AN APPLICATION TO FAIR RANKING

**Virginie Do**
PSL University & Meta AI
virginiedo@meta.com

**Elvis Dohmatob, Matteo Pirotta, Alessandro Lazaric, Nicolas Usunier**
Meta AI
{dohmatob,pirotta,lazaric,usunier}@meta.com

## ABSTRACT

We consider CONTEXTUAL BANDITS with CONCAVE REWARDS (CBCR), a multi-objective bandit problem where the desired trade-off between the rewards is defined by a known concave objective function, and the reward vector depends on an observed stochastic context. We present the first algorithm with provably vanishing regret for CBCR without restrictions on the policy space, whereas prior works were restricted to finite policy spaces or tabular representations. Our solution is based on a geometric interpretation of CBCR algorithms as optimization algorithms over the convex set of expected rewards spanned by all stochastic policies. Building on Frank-Wolfe analyses in constrained convex optimization, we derive a novel reduction from the CBCR regret to the regret of a *scalar-reward* bandit problem. We illustrate how to apply the reduction off-the-shelf to obtain algorithms for CBCR with both linear and general reward functions, in the case of non-combinatorial actions. Motivated by fairness in recommendation, we describe a special case of CBCR with rankings and fairness-aware objectives, leading to the first algorithm with regret guarantees for contextual combinatorial bandits with fairness of exposure.

## 1 INTRODUCTION

Contextual bandits are a popular paradigm for online recommender systems that learn to generate personalized recommendations from user feedback. These algorithms have been mostly developed to maximize a single scalar reward which measures recommendation performance for users. Recent fairness concerns have shifted the focus towards item producers whom are also impacted by the exposure they receive (Biega et al., 2018; Geyik et al., 2019), leading to optimize trade-offs between recommendation performance for users and fairness of exposure for items (Singh & Joachims, 2019; Zehlike & Castillo, 2020). More generally, there is an increasing pressure to insist on the multi-objective nature of recommender systems (Vamplew et al., 2018; Stray et al., 2021), which need to optimize for several engagement metrics and account for multiple stakeholders' interests (Mehrotra et al., 2020; Abdollahpouri et al., 2019). In this paper, we focus on the problem of contextual bandits with multiple rewards, where the desired trade-off between the rewards is defined by a known concave objective function, which we refer to as CONTEXTUAL BANDITS with CONCAVE REWARDS (CBCR). Concave rewards are particularly relevant to fair recommendation, where several objectives can be expressed as (known) concave functions of the (unknown) utilities of users and items (Do et al., 2021).

Our CBCR problem is an extension of BANDITS with CONCAVE REWARDS (BCR) (Agrawal & Devanur, 2014) where the vector of multiple rewards depends on an observed stochastic context. We address this extension because contexts are necessary to model the user/item features required for personalized recommendation. Compared to BCR, the main challenge of CBCR is that optimal policies depend on the entire distribution of contexts and rewards. In BCR, optimal policies are distributions over actions, and are found by direct optimization in policy space (Agrawal & Devanur, 2014; Berthet & Perchet, 2017). In CBCR, stationary policies are mappings from a continuous context space to distributions over actions. This makes existing BCR approaches inapplicable to CBCR because the policy space is not amenable to tractable optimization without further assumptions or restrictions. As a matter of fact, the only prior theoretical work on CBCR is restricted to a finite policy set (Agrawal et al., 2016).

We present *the first algorithms with provably vanishing regret* for CBCR without restriction on the policy space. Our main theoretical result is a reduction where the CBCR regret of an algorithm is

bounded by its regret on a proxy bandit task with *single* (scalar) reward. This reduction shows that it is straightforward to turn *any* contextual (scalar reward) bandits into algorithms for CBCR. We prove this reduction by first re-parameterizing CBCR as an optimization problem in the space of feasible rewards, and then revealing connections between Frank-Wolfe (FW) optimization in reward space and a decision problem in action space. This bypasses the challenges of optimization in policy space.

To illustrate how to apply the reduction, we provide two example algorithms for CBCR with non-combinatorial actions, one for linear rewards based on LinUCB (Abbasi-Yadkori et al., 2011), and one for *general reward functions* based on the SquareCB algorithm (Foster & Rakhlin, 2020) which uses online regression oracles. In particular, we highlight that our reduction can be used together with any exploration/exploitation principle, while previous FW approaches to BCR relied exclusively on upper confidence bounds (Agrawal & Devanur, 2014; Berthet & Perchet, 2017; Cheung, 2019).

Since fairness of exposure is our main motivation for CBCR, we show how our reduction also applies to the *combinatorial* task of fair ranking with contextual bandits, leading to the *first algorithm with regret guarantees* for this problem, and we show it is *computationally efficient*. We compare the empirical performance of our algorithm to relevant baselines on a music recommendation task.

**Related work.** Agrawal et al. (2016) address a restriction of CBCR to a finite set of policies, where explicit search is possible. Cheung (2019) use FW for reinforcement learning with concave rewards, a similar problem to CBCR. However, they rely on a tabular setting where there are few enough policies to compute them explicitly. Our approach is the only one to apply to CBCR without restriction on the policy space, by removing the need for explicit representation and search of optimal policies.

Our work is also related to fairness of exposure in bandits. Most previous works on this topic either do not consider rankings (Celis et al., 2018; Wang et al., 2021; Patil et al., 2020; Chen et al., 2020), or apply to combinatorial bandits without contexts (Xu et al., 2021). Both these restrictions are impractical for recommender systems. Mansoury et al. (2021); Jeunen & Goethals (2021) propose heuristics with experimental support that apply to both ranking and contexts in this space, but they lack theoretical guarantees. We present the first algorithm with regret guarantees for fair ranking with contextual bandits. We provide a more detailed discussion of the related work in Appendix A.

## 2 MAXIMIZATION OF CONCAVE REWARDS IN CONTEXTUAL BANDITS

**Notation.** For any $n \in \mathbb{N}$, we denote by $[\![n]\!] = \{1, \ldots, n\}$. The dot product of two vectors $x$ and $y$ in $\mathbb{R}^n$ is either denoted $x^\intercal y$ or using braket notation $\langle x \,|\, y \rangle$, depending on which one is more readable.

**Setting.** We define a stochastic contextual bandit (Langford & Zhang, 2007) problem with $D$ rewards. At each time step $t$, the environment draws a context $x_t \sim P$, where $x \in \mathcal{X} \subseteq \mathbb{R}^q$ and $P$ is a probability measure over $\mathcal{X}$. The learner chooses an action $a_t \in \mathcal{A}$ where $\mathcal{A} \subseteq \mathbb{R}^K$ is the action space, and receives a noisy multi-dimensional reward $r_t \in \mathbb{R}^D$, with expectation $\mathbb{E}[r_t | x_t, a_t] = \mu(x_t) a_t$, where $\mu : \mathcal{X} \to \mathbb{R}^{D \times K}$ is the matrix-value contextual expected reward function.[1] The trade-off between the $D$ cumulative rewards is specified by a known concave function $f : \mathbb{R}^D \to \mathbb{R} \cup \{\pm\infty\}$. Let $\overline{\mathcal{A}}$ denote the convex hull of $\mathcal{A}$ and $\overline{\pi} : \mathcal{X} \to \overline{\mathcal{A}}$ be a stationary policy,[2] then the optimal value for the problem is defined as $f^* = \sup_{\overline{\pi}:\mathcal{X} \to \overline{\mathcal{A}}} f\left( \mathbb{E}_{x \sim P}\left[ \mu(x) \overline{\pi}(x) \right] \right)$.

We rely on either of the following assumptions on $f$:

**Assumption A** $f$ *is closed proper concave*[3] *on $\mathbb{R}^D$ and $\mathcal{A}$ is a compact subset of $\mathbb{R}^K$. Moreover, there is a compact convex set $\mathcal{K} \subseteq \mathbb{R}^D$ such that*

- (Bounded rewards) $\forall (x, a) \in \mathcal{X} \times \mathcal{A}, \mu(x)a \in \mathcal{K}$ *and for all* $t \in \mathbb{N}_*$, $r_t \in \mathcal{K}$ *with probability* $1$.
- (Local Lipschitzness) $f$ *is $L$-Lipschitz continuous with respect to $\|.\|_2$ on an open set containing $\mathcal{K}$.*

**Assumption B** *Assumption A holds and $f$ has $C$-Lipschitz-continuous gradients w.r.t. $\|.\|_2$ on $\mathcal{K}$.*

---

[1] Notice that linear structure between $\mu(x_t)$ and $a_t$ is standard in combinatorial bandits (Cesa-Bianchi & Lugosi, 2012) and it reduces to the usual multi-armed bandit setting when $\mathcal{A}$ is the canonical basis of $\mathbb{R}^K$.

[2] In the multi-armed setting, stationary policies return a distribution over arms given a context vector. In the combinatorial setup, $\overline{\pi}(x) \in \overline{\mathcal{A}}$ is the average feature vector of a stochastic policy over $\mathcal{A}$. For the benchmark, we are only interested in expected rewards so there is to need to specify the full distribution over $\mathcal{A}$.

[3] This means that $f$ is concave and upper semi-continuous, is never equal to $+\infty$ and is finite somewhere.

The most general version of our algorithm, described in Appendix D, removes the need for the smoothness assumption using smoothing techniques. We describe an example in Section 3.3. In the rest of the paper, we denote by $D_\mathcal{K} = \sup_{z, z' \in \mathcal{K}} \|z - z'\|_2$ the diameter of $\mathcal{K}$, and use $\tilde{C} = \frac{C}{2} D_\mathcal{K}^2$.

We now give two examples of this problem setting, motivated by real-world applications in recommender systems, and which satisfy Assumption A.

**Example 1 (Optimizing multiple metrics in recommender systems.)** *Mehrotra et al. (2020) formalized the problem of optimizing $D$ engagement metrics (e.g. clicks, streaming time) in a bandit-based recommender system. At each $t$, $x_t$ represents the current user's features. The system chooses one arm among $K$, represented by a vector $a_t$ in the canonical basis of $\mathbb{R}^K$ which is the action space $\mathcal{A}$. Each entry of the observed reward vector $(r_{t,i})_{i=1}^D$ corresponds to a metric's value. The trade-off between the metrics is defined by the Generalized Gini Function: $f(z) = \sum_{i=1}^D w_i z_i^\uparrow$, where $(z_i^\uparrow)_{i=1}^D$ denotes the values of $z$ sorted increasingly and $w \in \mathbb{R}^D$ is a vector of non-increasing weights.*

**Example 2 (Fairness of exposure in rankings.)** *The goal is to balance the traditional objective of maximizing user satisfaction in recommender systems and the inequality of exposure between item producers (Singh & Joachims, 2018; Zehlike & Castillo, 2020). For a recommendation task with $m$ items to rank, this leads to a problem with $D = m + 1$ objectives, which correspond to the $m$ items' exposures, plus the user satisfaction metric. The context $x_t \in \mathcal{X} \subset \mathbb{R}^{md}$ is a matrix where each $x_{t,i} \in \mathbb{R}^d$ represents a feature vector of item $i$ for the current user. The action space $\mathcal{A}$ is combinatorial, i.e. it is the space of rankings represented by permutation matrices:*

$$\mathcal{A} = \left\{ a \in \{0,1\}^{m \times m} : \forall i \in [\![m]\!], \sum_{k=1}^m a_{i,k} = 1 \text{ and } \forall k \in [\![m]\!], \sum_{i=1}^m a_{i,k} = 1 \right\} \qquad (1)$$

*For $a \in \mathcal{A}$, $a_{i,k} = 1$ if item $i$ is at rank $k$. Even though we use a double-index notation and call $a$ a* permutation matrix, *we flatten $a$ as a vector of dimension $K = m^2$ for consistency of notation.*

*We now give a concrete example for $f$, which is concave as usual for objective functions in fairness of exposure (Do et al., 2021). It is inspired by Morik et al. (2020), who study trade-offs between average user utility and inequality[4] of item exposure:*

$$f(z) = \underbrace{z_{m+1}}_{\text{user utility}} - \beta \underbrace{\frac{1}{2m} \sum_{i=1}^m \sum_{j=1}^m |z_i - z_j|}_{\text{inequality of item exposure}} \qquad \text{where } \beta > 0 \text{ is a trade-off parameter.} \qquad (2)$$

**The learning problem.** In the bandit setting, $P$ and $\mu$ are unknown and the learner can only interact online with the environment. Let $h_T = (x_t, a_t, r_t)_{t \in [\![T-1]\!]}$ be the history of contexts, actions, and reward observed up to time $T - 1$ and $\delta' > 0$ be a confidence level, then at step $t$ a bandit algorithm $\mathfrak{A}$ receives in input the history $h_t$, the current context $x_t$, and it returns a distribution over actions $\mathcal{A}$ and selects an action $a_t \sim \mathfrak{A}(h_t, x_t, \delta')$. The objective of the algorithm is to minimize the regret

$$R_T = f^* - f(\hat{s}_T) \qquad \text{where } \hat{s}_T = \frac{1}{T} \sum_{t=1}^T r_t. \qquad (3)$$

Note that our setting subsumes classical stochastic contextual bandits: when $D = 1$ and $f(z) = z$, maximizing $f(\hat{s}_T)$ amounts to maximizing a cumulative scalar reward $\sum_{t=1}^T r_t$. In Lem. 9 (App. C.3), we show that alternative definitions of regret, with different choices of comparator or performance measure, would yield a difference of order $O(1/\sqrt{T})$, and hence not substantially change our results.

## 3 A GENERAL REDUCTION-BASED APPROACH FOR CBCR

In this section we describe our general approach for CBCR. We first derive our key reduction from CBCR to a specific scalar-reward bandit problem. We then instantiate our algorithm to the case of linear and general reward functions for smooth objectives $f$. Finally, we extend to the case of non-smooth objective functions using Moreau-Yosida regularization (Rockafellar & Wets, 2009).

---

[4]$\text{Gini}(z_1, \ldots, z_m) = \frac{1}{2m} \sum_{i=1}^m \sum_{j=1}^m |z_i - z_j|$ is an unnormalized Gini coefficient.

## 3.1 REDUCTION FROM CBCR TO SCALAR-REWARD CONTEXTUAL BANDITS

There are two challenges in the CBCR problem: 1) the computation of the optimal policy $\sup_{\overline{\pi}:\mathcal{X}\to\overline{\mathcal{A}}} f\left(\mathbb{E}_{x\sim P}\left[\mu(x)\overline{\pi}(x)\right]\right)$ even with known $\mu$; 2) the learning problem when $\mu$ is unknown.

**1: Reparameterization of the optimization problem.** The first challenge is that optimizing directly in policy space for the benchmark problem $\sup_{\overline{\pi}:\mathcal{X}\to\overline{\mathcal{A}}} f\left(\mathbb{E}_{x\sim P}\left[\mu(x)\overline{\pi}(x)\right]\right)$ is intractable without any restriction, because the policy space includes all mappings from the continuous context space $\mathcal{X}$ to distributions over actions. Our solution is to rewrite the optimization problem as a standard convex constrained problem by introducing the convex set $\mathcal{S}$ of feasible rewards:

$$\mathcal{S} = \left\{ \mathbb{E}_{x\sim P}\left[\mu(x)\overline{\pi}(x)\right] \middle| \overline{\pi} : \mathcal{X} \to \overline{\mathcal{A}} \right\} \text{ so that } f^* = \sup_{\overline{\pi}:\mathcal{X}\to\overline{\mathcal{A}}} f\left(\mathbb{E}_{x\sim P}\left[\mu(x)\overline{\pi}(x)\right]\right) = \max_{s\in\mathcal{S}} f(s).$$

Under Assumption A, $\mathcal{S}$ is a compact subset of $\mathcal{K}$ (see Lemma 7 in App. C) so $f$ attains its maximum over $\mathcal{S}$. We have thus reduced the complex initial optimization problem to a concave optimization problem over a compact convex set.

**2: Reducing the learning problem to scalar-reward bandits.** Unfortunately, since $P$ and $\mu$ are unknown, the set $\mathcal{S}$ is unknown. This precludes the possibility of directly using standard constrained optimization techniques, including gradient descent with projections onto $\mathcal{S}$. We consider Frank-Wolfe, a projection-free optimization method robust to approximate gradients (Lacoste-Julien et al., 2013; Kerdreux et al., 2018). At each iteration $t$ of FW, the update direction is given by the linear subproblem: $\text{argmax}_{s\in\mathcal{S}}\langle\nabla f(z_{t-1})\,|\,s\rangle$, where $z_{t-1}$ is the current iterate. Our main technical tool, Lemma 1, allows to connect the FW subproblem in the unknown reward space $\mathcal{S}$ to a workable decision problem in the action space (see Lemma 13 in Appendix E for a proof):

**Lemma 1** *Let $\mathbb{E}_t\big[.\big]$ be the expectation conditional on $h_t$. Let $z_t \in \mathcal{K}$ be a function of contexts, actions and rewards up to time $t$. Under Assumption A, we have:*

$$\forall t \in \mathbb{N}_*, \mathbb{E}_t\big[\max_{a\in\mathcal{A}}\langle\nabla f(z_{t-1})\,|\,\mu(x_t)a\rangle\big] = \max_{s\in\mathcal{S}}\langle\nabla f(z_{t-1})\,|\,s\rangle. \tag{4}$$

*For all $\delta \in (0,1]$, with probability at least $1-\delta$, we have:*

$$\sum_{t=1}^{T}\left(\max_{s\in\mathcal{S}}\langle\nabla f(z_{t-1})\,|\,s\rangle - \max_{a\in\mathcal{A}}\langle\nabla f(z_{t-1})\,|\,\mu(x_t)a\rangle\right) \le LD_{\mathcal{K}}\sqrt{2T\ln(\delta^{-1})}. \tag{5}$$

Lemma 1 shows that FW for CBCR operates closely to a sequence of decision problems of the form $(\max_{a\in\mathcal{A}}\langle\nabla f(z_{t-1})\,|\,\mu(x_t)a\rangle)_{t=1}^{T}$. However, we have yet to address the problem that $P$ and $\mu$ are unknown. To solve this issue, we introduce a **reduction to scalar-reward contextual bandits**. We can notice that solving for the sequence of actions maximizing $\sum_{t=1}^{T}\langle\nabla f(z_{t-1})\,|\,\mu(x_t)a\rangle$ corresponds to solving a contextual bandit problem with adversarial contexts and stochastic rewards. Formally, using $z_t = \hat{s}_t$[5], we define the extended context $\tilde{x}_t = (\nabla f(\hat{s}_{t-1}), x_t)$, the average scalar reward $\tilde{\mu}(\tilde{x}_t) = \nabla f(\hat{s}_{t-1})^\intercal\mu(x_t)$ and the observed scalar reward $\tilde{r}_t = \langle\nabla f(\hat{s}_{t-1})\,|\,r_t\rangle$. This fully defines a contextual bandit problem with *scalar reward*. Then, the objective of the algorithm is to minimize the following *scalar regret*:

$$R_T^{\text{scal}} = \sum_{t=1}^{T}\max_{a\in\mathcal{A}}\tilde{\mu}(\tilde{x}_t)^\intercal a - \sum_{t=1}^{T}\tilde{r}_t = \sum_{t=1}^{T}\max_{a\in\mathcal{A}}\langle\nabla f(\hat{s}_{t-1})\,|\,\mu(x_t)a\rangle - \sum_{t=1}^{T}\langle\nabla f(\hat{s}_{t-1})\,|\,r_t\rangle. \tag{6}$$

In this framework, the only information observed by the learning algorithm is $\tilde{h}_t := \big(\tilde{x}_{t'}, a_{t'}, \tilde{r}_{t'}\big)_{t'\in[\![t-1]\!]}$. This regret minimization problem has been extensively studied (see e.g., Slivkins, 2019, Chap. 8 for an overview). The following key reduction result[6] relates $R_T^{\text{scal}}$ to $R_T$, the regret of the original CBCR problem:

---

[5]For simplicity, we presented our reduction with $z_t = \hat{s}_t$ but other choices of $z_t$ are possible (see Appendix D). The important point is that the reduction works without restricting $z_t$ to $\mathcal{S}$.

[6]In practice, this result is used in conjunction with an upper bound $\overline{R}^{\text{scal}}(T,\delta')$ on $R_T^{\text{scal}}$ that holds with probability $\ge 1-\delta'$, which gives $R_T \le \overline{R}^{\text{scal}}(T,\delta')/T + O(\sqrt{\ln(1/\delta)/T})$ with probability at least $1-\delta-\delta'$ using the union bound.

**Theorem 2** *Under Assmpt. B, for every $T \in \mathbb{N}_*$ and $\delta > 0$, algorithm $\mathfrak{A}$ satisfies, with prob. $\geq 1 - \delta$:*

$$R_T = f^* - f(\hat{s}_T) \leq \frac{R_T^{\text{scal}} + LD_{\mathcal{K}}\sqrt{2T\ln(1/\delta)} + \tilde{C}\ln(eT)}{T}. \quad (7)$$

The reduction shown in Thm. 2 hints us at how to use or adapt scalar bandit algorithms for CBCR. In particular, any algorithm with sublinear regret will lead to a vanishing regret for CBCR. Since the worst-case regret of contextual bandits is $\Omega(\sqrt{T})$ (Dani et al., 2008), we obtain near minimax optimal algorithms for CBCR. We illustrate this with two algorithms derived from our reduction in Sec. 3.2.

*Proof sketch of Theorem 2: CBCR and Frank-Wolfe algorithms (full proof in Appendix E).*
Although the set $\mathcal{S}$ is not known, the standard telescoping sum argument for the analysis of Frank-Wolfe algorithms (see Lemma 14 in Appendix E, and e.g., (Berthet & Perchet, 2017, Lemma 12) for similar derivations) gives that under Assumption B, denoting $g_t = \nabla f(\hat{s}_{t-1})$:

$$TR_T \leq \sum_{t=1}^{T} \max_{s \in \mathcal{S}} \langle g_t \,|\, s - r_t \rangle + \tilde{C}\ln(eT).$$

The result is true for every sequence $(r_t)_{t \in [\![T]\!]} \in \mathcal{K}^T$, and only tracks the trajectory of $\hat{s}_t$ in reward space. We introduce now the reference of the scalar regret:

$$TR_T = \sum_{t=1}^{T} \big( \max_{s \in \mathcal{S}} \langle g_t \,|\, s \rangle - \max_{a \in \mathcal{A}} \langle g_t \,|\, \mu(x_t)a \rangle \big) + \underbrace{\sum_{t=1}^{T} \max_{a \in \mathcal{A}} \langle g_t \,|\, \mu(x_t)a - r_t \rangle}_{= R_T^{\text{scal}}} + \tilde{C}\ln(eT) \quad (8)$$

Lemma 1 bounds the leftmost term, from which Theorem 2 immediately follows using (8). $\qquad\square$

## 3.2 PRACTICAL APPLICATION: TWO ALGORITHMS FOR MULTI-ARMED CBCR

To illustrate the effectiveness of the reduction from CBCR to scalar-reward bandits, we focus on the case where the action space $\mathcal{A}$ is the canonical basis of $\mathbb{R}^K$ (as in Example 1). We first study the case of linear rewards. Then, for general reward functions, we introduce the FW-SquareCB algorithm, the first example of a FW-based approach combined with an exploration principle other than optimism. This shows our approach has a much broader applicability to solve (C)BCR than previous strategies.

**From LinUCB to FW-LinUCB (details in Appendix G).** We consider a CBCR with linear reward function, i.e., $\mu(x) = \theta x$ where $\theta \in \mathbb{R}^{D \times d}$ (recall we have $D$ rewards) and $x \in \mathbb{R}^{d \times K}$, where $d$ is the number of features. Let $\tilde{\theta} := \text{flatten}(\theta)$ and $g_t = \nabla f(\hat{s}_{t-1})$. Using $[.;.]$ to denote the vertical concatenation of matrices, the expected reward for action $a$ in context $x$ at time $t$ can be written $\langle g_t \,|\, \mu(x)a \rangle = g_t^{\mathsf{T}} \theta x a = \langle \tilde{\theta} \,|\, \tilde{x}_t a \rangle$ where $\tilde{x}_t \in \mathbb{R}^{Dd \times K}$ is the *extended* context with entries $\tilde{x}_t = [g_{t,0}x_t; \ldots; g_{t,D}x_t] \in \mathbb{R}^{Dd \times K}$. This is an instance of a linear bandit problem, where at each time $t$, action $a$ is associated to the vector $\tilde{x}_t a$ and its expected reward is $\langle \tilde{\theta} \,|\, \tilde{x}_t a \rangle$. As a result, we can immediately derive a LinUCB-based algorithm for linear CBCR by leveraging the equivalence FW-LinUCB$(h_t, x_t, \delta') = \text{LinUCB}(\tilde{h}_t, \tilde{x}_t, \delta')$. LinUCB's regret guarantees imply $R_T^{\text{scal}} = O(d\sqrt{T})$ with high probability, which, in turn give a $O(1/\sqrt{T})$ for $R_T$.

**From SquareCB to FW-SquareCB (details in Appendix H).** We now consider a CBCR with general reward function $\mu(x)$. The SquareCB algorithm (Foster & Rakhlin, 2020) is a randomized exploration strategy that delegates the learning of rewards to an arbitrary online regression algorithm. The scalar regret of SquareCB is bounded depending on the regret of the base regression algorithm.

For FW-SquareCB, we have access to an online regression oracle $\hat{\mu}_t$, an estimate of $\mu$ which is a function of $h_t$, which has regression regret bounded by $R_{\text{oracle}}(T)$. The exploration strategy of FW-SquareCB follows the same principles as SquareCB: let $g_t = \nabla f(\hat{s}_{t-1})$ and denote $\underline{\hat{\mu}}_t = g_t^{\mathsf{T}} \hat{\mu}_t(x_t)$, so that $\underline{\hat{\mu}}_t^{\mathsf{T}} a = \langle g_t \,|\, \hat{\mu}_t(x_t)a \rangle$. Let $\mathfrak{A}_t = \text{FW-SquareCB}(h_t, x_t, \delta')$ defined as

$$\forall a \in \mathcal{A}, \mathfrak{A}_t(a) = \begin{cases} \frac{1}{K + \gamma_t\left(\underline{\hat{\mu}}_t^* - \underline{\hat{\mu}}_t^{\mathsf{T}}a\right)} & \text{if } a \neq \underline{a}_t \\ 1 - \sum_{\substack{a \in \mathcal{A} \\ a \neq \underline{a}_t}} \mathfrak{A}_t(a) & \text{if } a = \underline{a}_t \end{cases} \quad \text{where } \underline{a}_t \in \underset{a \in \mathcal{A}}{\text{argmax}}\, \underline{\hat{\mu}}_t^{\mathsf{T}}a \text{ and } \underline{\hat{\mu}}_t^* = \underline{\hat{\mu}}_t\underline{a}_t$$

Then FW-SquareCB has $R_T$ in $O(\sqrt{R_{\text{oracle}}(T)}/\sqrt{T})$ with high probability.

Table 1: Regret bounds depending on assumptions and base algorithm $\mathfrak{A}$, for multi-armed bandits with $K$ arms (in dimension $d$ for LinUCB). See Appendix G and H for the full details.

| Algorithm (FW-<bandit>) | Assumptions (informal) | Bound on $R_T$ (simplified, using $\delta' = \delta$) |
|---|---|---|
| FW-LinUCB | $\mu(x)a = \theta x a$ for $\theta \in \mathbb{R}^{D \times d}, x \in \mathbb{R}^{d \times K}$ | $\dfrac{L D_{\mathcal{K}} d D \ln\left(\left(1 + \frac{TLD_{\mathcal{K}}}{dD}\right)/\delta\right)}{\sqrt{T}}$ |
| FW-SquareCB | $\sum_{t=1}^{T} \|\hat\mu_t(x_t)a_t - \mu(x_t)a_t\|_2^2 \le R_{\text{oracle}}(T)$ | $\dfrac{L\sqrt{K\left(R_{\text{oracle}}(T) + D_{\mathcal{K}}^2 \ln(T/\delta)\right)}}{\sqrt{T}}$ |

### 3.3 THE CASE OF NONSMOOTH $f$

When $f$ is nonsmooth, we use a smoothing technique where the scalar regret is not measured using $\nabla f(\hat s_{t-1})$, but rather using gradients of a sequence $(f_t)_{t \in \mathbb{N}}$ of smooth approximations of $f$, whose smoothness decrease over time (see e.g., Lan, 2013, for applications of smoothing to FW). We provide a comprehensive treatment of smoothing in our general approach described in Appendix D, while specific smoothing techniques are discussed in Appendix F.

We now describe the use of Moreau-Yosida regularization (Rockafellar & Wets, 2009, Def. 1.22): $f_t(z) = \max_{y \in \mathbb{R}^D} \left( f(y) - \frac{\sqrt{t+1}}{2\beta_0} \|y - z\|_2^2 \right)$. It is well-known that $f_t$ is concave and $L$-Lipschitz whenever $f$ is, and $f_t$ is $\frac{\sqrt{t+1}}{\beta_0}$-smooth (see Lemma 15 in Appendix F). A related smoothing method was used by Agrawal & Devanur (2014) for (non-contextual) BCR. Our treatment of smoothing is more systematic than theirs, since we use a smoothing factor $\beta_0/\sqrt{t+1}$ that decreases over time rather than a fixed smoothing factor that depends on a pre-specified horizon. Our regret bound for CBCR is based on a scalar regret $R_T^{\text{scal,sm}}$ where $\nabla f_{t-1}(\hat s_{t-1})$ is used instead of $\nabla f(\hat s_{t-1})$:

$$R_T^{\text{scal,sm}} = \sum_{t=1}^{T} \max_{a \in \mathcal{A}} \langle \nabla f_{t-1}(\hat s_{t-1}) \,|\, \mu(x_t)a \rangle - \sum_{t=1}^{T} \langle \nabla f_{t-1}(\hat s_{t-1}) \,|\, r_t \rangle. \tag{9}$$

**Theorem 3** *Under Assumptions A, for every $z_0 \in \mathcal{K}$, every $T \ge 1$ and every $\delta > 0, \delta' > 0$, Algorithm $\mathfrak{A}$ satisfies, with probability at least $1 - \delta - \delta'$:*

$$R_T \le \frac{R_T^{\text{scal,sm}}}{T} + \frac{L D_{\mathcal{K}}}{\sqrt{T}}\left(\frac{D_{\mathcal{K}}}{L\beta_0} + 3\frac{L\beta_0}{D_{\mathcal{K}}} + \sqrt{2 \ln \frac{1}{\delta}}\right). \tag{10}$$

The proof is given in Appendix F. Taking $\beta_0 = \frac{D_{\mathcal{K}}}{L}$ leads to a simpler bound where $\frac{D_{\mathcal{K}}}{L\beta_0} + 3\frac{L\beta_0}{D_{\mathcal{K}}} = 4$.

## 4 CONTEXTUAL RANKING BANDITS WITH FAIRNESS OF EXPOSURE

In this section, we apply our reduction to the combinatorial bandit task of fair ranking, and obtain the first algorithm with regret guarantees in the contextual setting. This task is described in Example 2 (Sec. 2). We remind that there is a fixed set of $m$ items to rank at each timestep $t$, and that actions are flattened permutation matrices ($\mathcal{A}$ is defined in Ex. 2, Eq. (1)). The context $x_t \sim P$ is a matrix $x_t = (x_{t,i})_{i \in [\![m]\!]}$ where each $x_{t,i} \in \mathbb{R}^d$ represents a feature vector of item $i$ for the current user.

**Observation model.** The *user utility* $u(x_t)$ is given by a position-based model with position weights $b(x_t) \in [0,1]^m$ and expected value for each item $v(x_t) \in [0,1]^m$. Denoting $u(x_t)$ the flattened version of $v(x_t)b(x_t)^\mathsf{T} \in \mathbb{R}^{m \times m}$, the user utility is (Lagrée et al., 2016; Singh & Joachims, 2018):

$$\langle u(x_t) \,|\, a \rangle = \sum_{i=1}^{m} v_i(x_t) \sum_{k=1}^{m} a_{i,k} b_k(x_t).$$

In this model, $b_k(x_t) \in [0,1]$ is the probability that the user observes the item at rank $k$. The quantity $\sum_{k=1}^{m} a_{i,k} b_k(x_t)$ is thus the probability that the user observes item $i$ given ranking $a$. We denote

---

**Algorithm 1:** FW-LinUCBRank: linear contextual bandits for fair ranking.

**input :** $\delta' > 0, \lambda > 0, \hat{s}_0 \in \mathcal{K}$ $V_0 = \lambda \boldsymbol{I}_d, y_0 = \boldsymbol{0}_d, \hat{\theta}_0 = \boldsymbol{0}_d$

1 **for** $t = 1, \ldots$ **do**

2     Observe context $x_t \sim P$

3     $\forall i, \hat{v}_{t,i} \leftarrow \hat{\theta}_{t-1}^{\mathsf{T}} x_{t,i} + \alpha_t\left(\frac{\delta'}{3}\right) \|x_{t,i}\|_{V_{t-1}^{-1}}$        // UCB on $v_i(x_t)$ (def. of $\alpha_t$ in Lem. 26, App. I)

4     $a_t \leftarrow$ top-$\overline{k}\{\frac{\partial f}{\partial z_{m+1}}(\hat{s}_{t-1})\hat{v}_{t,i} + \frac{\partial f}{\partial z_i}(\hat{s}_{t-1})\}_{i=1}^m$        // FW linear optimization step

5     Observe exposed items $e_t \in \{0,1\}^m$ and user feedback $c_t \in \{0,1\}^m$

6     Update $\hat{s}_t \leftarrow \hat{s}_{t-1} + \frac{1}{t}(r_t - \hat{s}_{t-1})$

7     $V_t \leftarrow V_{t-1} + \sum_{i=1}^m e_{t,i} x_{t,i} x_{t,i}^{\mathsf{T}}, \quad y_t \leftarrow y_{t-1} + \sum_{i=1}^m c_{t,i} x_{t,i}$ and $\hat{\theta}_t \leftarrow V_t^{-1} y_t$        // regression

8 **end**

---

$\overline{k} = \max_{x \in \mathcal{X}} \|b(x)\|_0 \leq m$ the maximum rank that can be exposed to any user. In most practical applications, $\overline{k} \ll m$. As formalized in Assumption D below, the position weights $b_k(x)$ are always non-increasing with $k$ since the user browses the recommended items in order of their rank. We use a linear assumption for item values, where $D_{\mathcal{X}}$ and $D_\theta$ are known constants:

**Assumption C** $\sup_{x \in \mathcal{X}} \|x\|_2 \leq D_{\mathcal{X}}$ *and* $\exists \theta \in \mathbb{R}^d, \|\theta\|_2 \leq D_\theta$ *s.t.* $\forall x \in \mathcal{X}, \forall i \in [\![m]\!], v_i(x) = \theta^{\mathsf{T}} x_i.$

We propose an observation model where values $v_i(x)$ *and* position weights $b(x)$ are *unknown*. However, we assume that at each time step $t$, after computing the ranking $a_t$, we have two types of feedback: first, $e_{t,i} \in \{0,1\}$ is 1 if item $i$ has been exposed to the user, and 0 otherwise. Second $c_{t,i} \in \{0,1\}$ which represents a binary like/dislike feedback from the user. We have

$$\mathbb{E}[e_{t,i}|x_t, a_t] = \sum_{k=1}^m a_{t,i,k} b_k(x_t) \qquad \mathbb{E}[c_{t,i}|x_t, e_{t,i}] = \begin{cases} v_i(x_t) & \text{if } e_{t,i} = 1 \\ 0 & \text{if } e_{t,i} = 0 \end{cases} \qquad (11)$$

This observation model captures well applications such as newsfeed ranking on mobile devices or dating applications where only one post/profile is shown at a time. What we gain with this model is that $b(x)$ can *depend arbitrarily on the context* $x$, while previous work on bandits in the position-based model assumes $b$ known and context-independent (Lagrée et al., 2016).[7]

**Fairness of exposure.** There are $D = m + 1$ rewards, i.e., $\mu(x) \in \mathbb{R}^{(m+1) \times m^2}$. Denoting $\mu_i(x)$ the $i$th-row of $\mu(x)$, seen as a column vector, each of the $m$ first rewards is the exposure of a specific item, while the $m + 1$-th reward is the user utility:

$$\forall i \in [\![m]\!], \langle \mu_i(x) \,|\, a \rangle = \sum_{k=1}^m a_{i,k} b_k(x) \qquad \text{and} \qquad \mu_{m+1}(x) = u(x) \qquad (12)$$

The observed reward vector $r_t \in \mathbb{R}^D$ is defined by $\forall i \in [\![m]\!], r_{t,i} = e_{t,i}$ and $r_{t,m+1} = \sum_{i=1}^m c_{t,i}$. Notice that $\mathbb{E}[r_{t,m+1}|x_t] = u(x_t)$. Let $\mathcal{K}$ be the convex hull of $\{z \in \{0,1\}^{m+1} : \sum_{i=1}^m z_i \leq \overline{k}$ and $z_{m+1} \leq \sum_{i=1}^m z_i\}$, we have $D_{\mathcal{K}} \leq \sqrt{\overline{k}}\sqrt{\overline{k}+2} \leq \overline{k} + 1$ and $r_t \in \mathcal{K}$ with probability 1. The objective function $f : \mathbb{R}^D \to \mathbb{R}$ makes a trade-off between average user utility and inequalities in item exposure (we gave an example in Eq. (2)). The remaining assumptions of our framework are that the objective function is non-decreasing with respect to average user utility. This is not required but it is natural (see Example 2) and slightly simplifies the algorithm.

**Assumption D** *The assumptions of the framework described above hold, as well as Assumption B. Moreover,* $\forall z \in \mathcal{K} \frac{\partial f}{\partial z_{m+1}}(z) > 0$, *and* $\forall x \in \mathcal{X}, 1 \geq b_1(x) \geq \ldots \geq b_{\overline{k}}(x) = \ldots = b_m(x) = 0.$

**Algorithm and results.** We present the algorithm in the setting of linear contextual bandits, using LinUCB (Abbasi-Yadkori et al., 2011; Li et al., 2010) as scalar exploration/exploitation algorithm in

---

[7]When $b$ is unknown, depends on the context $x$, and we do not observe $e_t$, several approaches have been proposed to estimate the position weights (see e.g., Fang et al., 2019). Incorporating these approaches in contextual bandits for ranking is likely feasible but out of the scope of this work.

Figure 1: **(left)** Multi-armed CBCR: Objective values on environments from (Mehrotra et al., 2020). **(middle)** Ranking CBCR: Fairness objective value over timesteps on Last.fm data. **(right)** Ranking CBCR: Trade-off between user utility and item inequality after $5 \times 10^6$ iterations on Last.fm data.

Algorithm 1. It builds reward estimates based on Ridge regression with regularization parameter $\lambda$. As in the previous section, we focus on the case where $f$ is smooth but the extension to nonsmooth $f$ is straightforward, as described in Section 3. Appendix I provides the analysis for the general case.

As noted by Do et al. (2021), Frank-Wolfe algorithms are particularly suited for fair ranking in the position-based model. This is illustrated by line 4 of Alg. 1, where for $\tilde{u} \in \mathbb{R}^m$, top-$\overline{k}(\tilde{u})$ outputs a permutation (matrix) of $[\![m]\!]$ that sorts the top-$\overline{k}$ elements of $\tilde{u}$. Alg. 1 is thus *computationally fast*, with a cost dominated by the top-$\overline{k}$ sort. It also has an intuitive interpretation as giving items an adaptive bonus depending on $\nabla f$ (e.g., boosting the scores of items which received low exposure in previous steps). The following result is a consequence of (Do et al., 2021, Theorem 1):

**Proposition 4** *Let $t \in \mathbb{N}_*$ and $\hat{\mu}_t$ such that $\forall i \in [\![m]\!], \hat{\mu}_{t,i} = \mu_i(x_t)$ and $\hat{\mu}_{t,m+1} = \hat{v}_t b(x_t)^\intercal$ viewed as a column vector, with $\hat{v}$ defined in line 3 of Algorithm 1. Then, under Assumption D, $a_t$ defined on line 4 of Algorithm 1 satisfies: $\langle \nabla f(\hat{s}_{t-1}) \mid \hat{\mu}_t a_t \rangle = \underset{a \in \mathcal{A}}{\mathrm{argmax}} \langle \nabla f(\hat{s}_{t-1}) \mid \hat{\mu}_t a \rangle.$*

The proposition says that even though computing $a_t$ as in line 4 of Alg. 1 does not require the knowledge of $b(x_t)$, we still obtain the optimal update direction according to $\hat{\mu}_t$. Together with the usage of the observed reward $r_t$ in FW iterates (instead of e.g., $\hat{\mu}_t a_t$ as would be done by Agrawal & Devanur (2014)), this removes the need for explicit estimates of $\mu(x_t)$. This is how our algorithm works *without knowing the position weights $b(x_t)$*, which are then allowed to depend on the context.

The usage of $\hat{v}_t$ to compute $a_t$ follows the usual confidence-based approach to explore/exploitation principles for linear bandits, which leads to the following result (proven in Appendix I):

**Theorem 5** *Under Assumptions B, C and D, for every $\delta' > 0$, every $T \in \mathbb{N}_*$, every $\lambda \geq D_\mathcal{X}^2 \overline{k}$, with probability at least $1 - \delta'$, Algorithm 1 has scalar regret bounded by*

$$R_T^{\mathrm{scal}} = O\left( L\sqrt{T\overline{k}} \sqrt{d \ln(T/\delta')} \left( \sqrt{d \ln(T/\delta')} + D_\theta \sqrt{\lambda} + \sqrt{\overline{k}/d} \right) \right). \tag{13}$$

*Thus, considering only $d, T, \overline{k}$ and $\delta = \delta'$ Alg. 1 has regret $R_T \leq O\left( \frac{d\overline{k} \ln(T/\delta)}{\sqrt{T}} \right)$ w.p. at least $1 - \delta$.*

## 5 EXPERIMENTS

We present two experimental evaluations of our approach, which are fully detailed in App. B.

### 5.1 MULTI-ARMED CBCR: APPLICATION TO MULTI-OBJECTIVE BANDITS

We first focus on the multi-objective recommendation task of Example 1 where $f(z) = \sum_{i=1}^{D} w_i z_i^\uparrow$.

**Algorithms.** We evaluate our two instantiations presented in Sec. 3.2 with the Moreau-Yosida smoothing technique of Sec. 3.3: (i) FW-SquareCB with Ridge regression and (ii) FW-LinUCB, where exploration is controlled by a scaling variable $\epsilon$ on the exploration bonus of each arm. We compare them to MOLinCB from (Mehrotra et al., 2020).

**Environments.** We reproduce the synthetic environments of Mehrotra et al. (2020), where the context and reward parameters are generated randomly, and $w_i = \frac{1}{2^{i-1}}$. We set $K = 50$ and $D \in \{5, 20\}$ (we also vary $K$ in App. B). Each simulation is repeated with 100 random seeds.

**Results.** Following (Mehrotra et al., 2020), we evaluate the algorithms' performance by measuring the value of $f(\frac{1}{T} \sum_{t=1}^T \mu(x_t)a_t)$ over time. Our results are shown in Figure 1 (left). We observe that our algorithm FW-SquareCB obtains comparable performance with the baseline MOLinCB. These algorithms converge after $\approx 100$ rounds. In this environment from (Mehrotra et al., 2020), only little exploration is needed, hence FW-LinUCB obtains better performance when $\epsilon$ is smaller ($\epsilon = 0.01$). The advantage of using an FW instantiation for the multi-objective bandit optimization task is that unlike MOLinCB, its convergence is also supported by our theoretical regret guarantees.

## 5.2 RANKING CBCR: APPLICATION TO FAIRNESS OF EXPOSURE IN RANKINGS

We now tackle the ranking problem of Section 4. We show how FW-LinUCBRank allows to fairly distribute exposure among items on a music recommendation task with bandit user feedback.

**Environment.** Following (Patro et al., 2020), we use the Last.fm music dataset from (Cantador et al., 2011), from which we extract the top 50 users and items with the most listening counts. We use a protocol similar to Li et al. (2016) to generate context and rewards from those. We use $\bar{k} = 10$ ranking slots, and exposure weights $b_k(x) = \frac{\log(2)}{1+\log(k)}$. Simulations are repeated with 10 seeds.

**Algorithms.** Our algorithm is FW-LinUCBRank with the nonsmooth objective $f$ of Eq. (2), which trades off between user utility and item inequality. We study other fairness objectives in App. B. Our first baseline is *LinUCBRank* (Ermis et al., 2020), designed for ranking without fairness. Then, we study two baselines with amortized fairness of exposure criteria. Mansoury et al. (2021) proposed a fairness module for UCB-based ranking algorithms, which we plug into *LinUCBRank*. We refer to this baseline as *Unbiased-LinUCBRank*. Finally, the *FairLearn($c, \alpha$)* algorithm (Patil et al., 2020) enforces as fairness constraint that the pulling frequency of each arm be $\geq c$, up to a tolerance $\alpha$. We implement as third baseline a simple adaptation of *FairLearn* to contextual bandits and ranking.

**Dynamics.** Figure 1 (middle) represents the values of $f$ over time achieved by the competing algorithms, for fixed $\beta = 1$. As expected, compared to the fairness-aware and -unaware baselines, our algorithm FW-LinUCBRank reaches the best values of $f$. Interestingly, *Unbiased-LinUCBRank* also obtains high values of $f$ on the first $10^4$ rounds, but its performance starts decreasing after more iterations. This is because *Unbiased-LinUCBRank* is not guaranteed to converge to an optimal trade-off between user fairness and item inequality.

**At convergence.** We analyse the trade-offs achieved after $5 \cdot 10^6$ rounds between user utility and item inequality measured by the Gini index. We vary $\beta$ in the objective $f$ of Eq. (2) for FW-LinUCBRank and the strength $c$ in *FairLearn($c, \alpha$)*, with tolerance $\alpha = 1$. In Fig. 1 (right), we observe that compared to *FairLearn*, FW-LinUCBRank converges to much higher user utility at all levels of inequality among items. In particular, it achieves zero-unfairness at little cost for user utility.

## 6 CONCLUSION

We presented the first general approach to contextual bandits with concave rewards. To illustrate the usefulness of the approach, we show that our results extend randomized exploration with generic online regression oracles to the concave rewards setting, and extend existing ranking bandit algorithms to fairness-aware objective functions. The strength of our reduction is that it can produce algorithms for CBCR from any contextual bandit algorithm, including recent extensions of SquareCB to infinite compact action spaces (Zhu & Mineiro, 2022; Zhu et al., 2022) and future ones.

In our main application to fair ranking, the designer sets a fairness trade-off $f$ to optimize. In practice, they may choose $f$ among a small class by varying hyperparameters (e.g. $\beta$ in Eq. (2)). An interesting open problem is the integration of recent elicitation methods for $f$ (e.g., Lin et al., 2022) in the bandit setting. Another interesting issue is the generalization of our framework to include constraints (Agrawal & Devanur, 2016). Finally, we note that the deployment of our algorithms requires to carefully design the whole machine learning setup, including the specification of reward functions (Stray et al., 2021), the design of online experiments (Bird et al., 2016), while taking feedback loops into account (Bottou et al., 2013; Jiang et al., 2019; Dean & Morgenstern, 2022).

ACKNOWLEDGMENTS

The authors would like to thank Clément Vignac, Marc Jourdan, Yaron Lipman, Levent Sagun and the anonymous reviewers for their helpful comments.

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

## CONTENTS

## A    RELATED WORK

The non-contextual setting of bandits with concave rewards (BCR) has been previously studied by Agrawal & Devanur (2014), and by Busa-Fekete et al. (2017) for the special case of Generalized Gini indices. In BCR, policies are distributions over actions. These approaches perform a direct optimization in policy space, which is not possible in the contextual setup without restrictions or assumptions on optimal policies. Agrawal et al. (2016) study a setting of CBCR where the goal is to find the best policy in a finite set of policies. Because they rely on explicit search in the policy space, they do not resolve the main challenge of the general CBCR setting we address here. Cheung

(2019); Siddique et al. (2020); Mandal & Gan (2022); Geist et al. (2021) address multi-objective reinforcement learning with concave aggregation functions, a problem more general than stochastic contextual bandits. In particular, Cheung (2019) use a FW approach for this problem. However, these works rely on a tabular setting (i.e., finite state and action sets) and explicitly compute policies, which is not possible in our setting where policies are mappings from a continuous context set to distributions over actions. Our work is the only one amenable to contextual bandits with concave rewards by removing the need for an explicit policy representation. Finally, compared to previous FW approaches to bandits with concave rewards, e.g. (Agrawal & Devanur, 2014; Berthet & Perchet, 2017), our analysis is not limited to confidence-based exploration/exploitation algorithms.

CBCR is also related to the broad literature on bandit convex optimization (BCO) (Flaxman et al., 2004; Agarwal et al., 2011; Hazan et al., 2016; Shalev-Shwartz et al., 2012). In BCO, the goal is to minimize a cumulative loss of the form $\sum_{t=1}^{T} \ell_t(\pi_t)$, where the convex loss function $\ell_t$ is *unknown* and the learner only observes the value $\ell_t(\pi_t)$ of the chosen parameter $\pi_t$ at each timestep. Existing approaches to BCO perform gradient-free optimization in the parameter space. While BCR considers global objectives rather than cumulative ones, similar approaches have been used in non-contextual BCR (Berthet & Perchet, 2017) where the parameter space is the convex set of distributions over actions. As we previously highlighted, such parameterization does not apply to CBCR because direct optimization in policy space is infeasible.

CBCR is also related to multi-objective optimization (Miettinen, 2012; Drugan & Nowe, 2013), where the goal is to find all Pareto efficient solutions. (C)BCR, focuses on one point of the Pareto front determined by the concave aggregation function $f$, which is more practical in our application settings where the decision-maker is interested in a specific (e.g., fairness) trade-off.

In recent years, the question of fairness of exposure attracted a lot of attention, and has been mostly studied in a static ranking setting (Geyik et al., 2019; Beutel et al., 2019; Yang & Stoyanovich, 2017; Singh & Joachims, 2018; Patro et al., 2022; Zehlike et al., 2021; Kletti et al., 2022; Diaz et al., 2020; Do & Usunier, 2022; Wu et al., 2022). Existing work on fairness of exposure in bandits focused on local exposure constraints on the probability of pulling an arm at each timestep, either in the form of lower/upper bounds (Celis et al., 2018) or merit-based exposure targets (Wang et al., 2021). In contrast, we consider amortized exposure over time, in line with prior work on fair ranking (Biega et al., 2018; Morik et al., 2020; Usunier et al., 2022), along with fairness trade-offs defined by concave objective functions which are more flexible than fairness constraints (Zehlike & Castillo, 2020; Do et al., 2021; Usunier et al., 2022). Moreover, these works (Celis et al., 2018; Wang et al., 2021) do not address combinatorial actions, while ours applies to ranking in the position-based model, which is more practical for recommender systems (Lagrée et al., 2016; Singh & Joachims, 2018). The methods of (Patil et al., 2020; Chen et al., 2020) aim at guaranteeing a minimal cumulative exposure over time for each arm, but they also do not apply to ranking. In contrast, (Xu et al., 2021; Li et al., 2019) consider combinatorial bandits with fairness, but they do not address the contextual case, which limits their practical application to recommender systems. (Mansoury et al., 2021; Jeunen & Goethals, 2021) propose heuristic algorithms for fairness in ranking in the contextual bandit setting, highlighting the problem's importance for real-world recommender systems, but they lack theoretical guarantees. Using our FW reduction with techniques from contextual combinatorial bandits (Lagrée et al., 2016; Li et al., 2016; Qin et al., 2014), we obtain the first principled bandit algorithms for this problem with provably vanishing regret.

## B    MORE ON EXPERIMENTS

Our experiments are fully implemented in Python 3.9.

### B.1    RANKING CBCR: APPLICATION TO FAIRNESS OF EXPOSURE IN RANKINGS WITH BANDIT FEEDBACK

#### B.1.1    DETAILS OF THE ENVIRONMENT AND ALGORITHMS

**Environment**    Following (Patro et al., 2020) who also address fairness in recommender systems, we use the Last.fm music dataset[8] from (Cantador et al., 2011), which includes the listening counts of

---

[8]`https://www.last.fm`, the dataset is publicly available for non-commercial use.

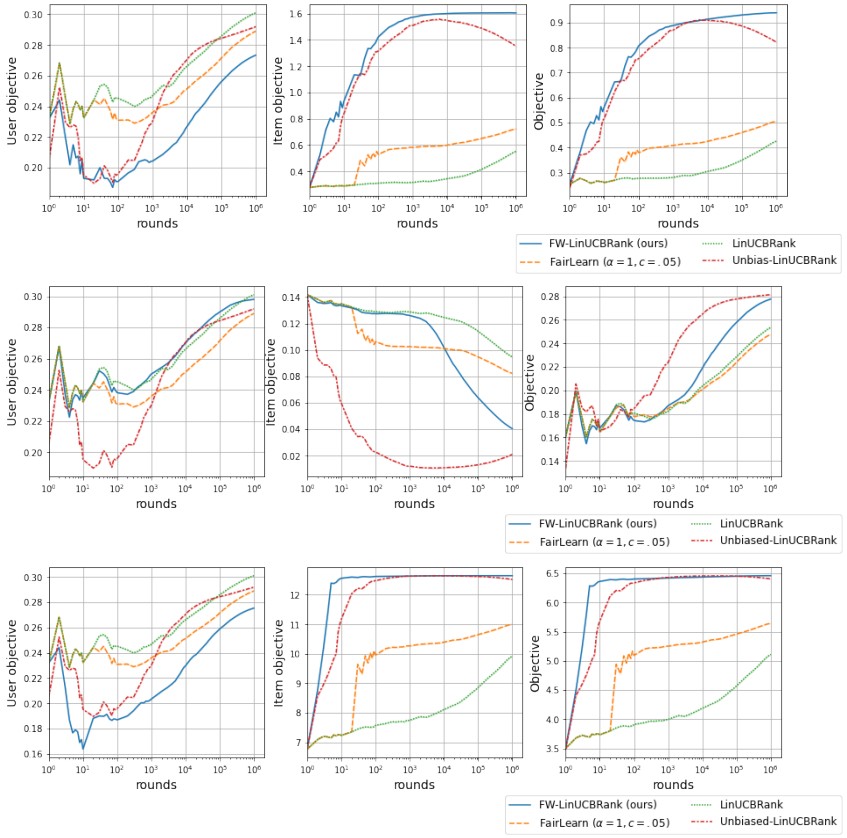

Figure 2: *Lastfm-50*: Objective values over time for (top) *Gini*, (middle) *eq. exposure*, (bottom) *welf*.

$1,892$ users for the tracks of $17,632$ artists, which we identify as the items. For the first environment, which we presented in Section 5 and which we call *Lastfm-50* here, we extract the top $n = 50$ users and $m = 50$ items having the most interactions. In order to examine algorithms at larger scale, we also design another environment, *Lastfm-2k*, where we keep all $n = 1.9k$ users and the top $m = 2.5k$ items having the most interactions. In both cases, to generate contexts and rewards, we follow a protocol similar to other works on linear contextual bandits (Garcelon et al., 2020; Li et al., 2016). Using low-rank matrix factorization with $d'$ latent factors[9], we obtain user factors $u_j \in \mathbb{R}^{d'}$ and item factors $v_i \in \mathbb{R}^{d'}$ for all $j, i \in [\![n]\!] \times [\![m]\!]$. We design the context set as $\mathcal{X} = \{\text{flatten}(u_j v_i^\intercal) : j, i \in [\![n]\!] \times [\![m]\!]\} \subset \mathbb{R}^d$, where $d = d'^2$. At each time step $t$, the environment draws a user $j_t$ uniformly at random from $[\![n]\!]$ and sends context $x_t = \text{flatten}(u_{j_t} v_i^\intercal)$. Given a context $x_t$ and item $i$, clicks are drawn from a Bernoulli distribution: $c_{t,i} \sim \mathcal{B}(u_{j_t}^\intercal v_i)$.

We set $\bar{k} = 10$, and for the position weights, we use the standard weights of the discounted cumulative gain (DCG): $\forall k \in [\![\bar{k}]\!], b_k = \frac{1}{\log_2(1+k)}$ and $b_{\bar{k}+1}, \ldots, b_m = 0$.

**Details of the algorithms** For all algorithms, the regularization parameter of the Ridge regression is set to $\lambda = 0.1$.

The first baseline we consider is the algorithm *LinUCBRank*[10] of (Ermis et al., 2020), which is a top-$\bar{k}$ ranking bandit algorithm without fairness. It is equivalent to using FW-LinUCBRank with $f(s) = s_{m+1}$, which corresponds to the usual top-$\bar{k}$ ranking objective without item fairness. More precisely, at each timestep, the algorithm produces a top-$\bar{k}$ ranking of $\left(\hat{\theta}_{t-1}^\intercal x_{t,i} + \alpha_t(\frac{\delta'}{3})\|x_{t,i}\|_{V_{t-1}^{-1}}\right)_{i=1}^m$.

___

[9]Using the Python library Implicit, MIT License: https://implicit.readthedocs.io/

[10]*LinUCBRank* appears under various names in the literature, including PBMLinUCBRank (Ermis et al., 2020) and CascadeLinUCB (Kveton et al., 2015).

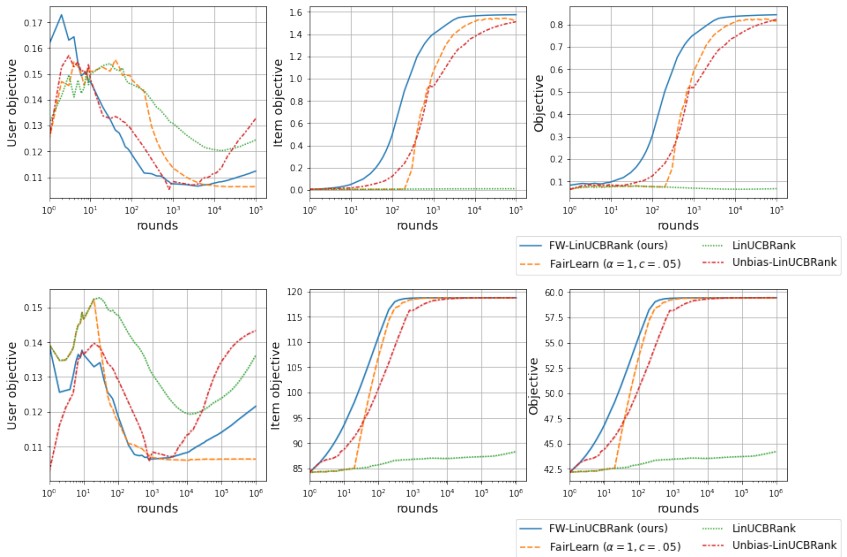

Figure 3: *Lastfm-2k*: Objective values over time for (top) *Gini*, (bottom) *welf*.

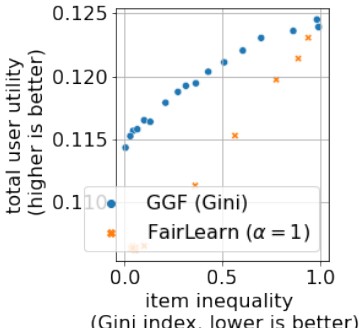

Figure 4: Trade-offs between user utility and inequality on *Lastfm-2k*, after $T = 10^6$ rounds.

We also consider as baselines two bandit algorithms with amortized fairness of exposure criteria. First, Mansoury et al. (2021) proposed a fairness module for cascade ranking bandits, which can be easily adapted to the position-based model (PBM). Their goals include reducing inequality in exposure between items, measured by the Gini index of exposures in their experiments. While they measure the exposure of an item as their recommendation frequency over time, we adapt their module to the PBM by using the observation frequency, i.e. $\sum_{t'=1}^{t} e_{t',i}$ for item $i$ at time $t$. Transposed to our setting, their module consists in a simple modification of *LinUCBRank* by multiplying the exploration bonus of each item $i$ by a factor:

$$\eta_{t,i} = 1 - \frac{\sum_{t'=1}^{t-1} e_{t',i}}{\sum_{t'=1}^{t-1} \frac{1}{k} \sum_{i'=1}^{m} e_{t',i'}}. \tag{14}$$

More precisely, at each timestep, the algorithm produces a top-$\bar{k}$ ranking of $\left( \hat{\theta}_{t-1}^{\mathsf{T}} x_{t,i} + \eta_{t,i} \times \alpha_t(\frac{\delta'}{3}) \|x_{t,i}\|_{V_{t-1}^{-1}} \right)_{i=1}^{m}$. Following (Mansoury et al., 2021), we call this baseline *Unbiased-LinUCBRank*.

Our second baseline with fairness is the *FairLearn$(c, \alpha)$* algorithm of Patil et al. (2020) for stochastic bandits with a fairness constraint on the pulling frequency $N_{t,i}$ of each arm $i$ at each timestep $t$. The constraint is parameterized by a variable $c$ and a tolerance parameter $\alpha$: $\lfloor ct \rfloor - N_{t,i} \leq \alpha$. We adapt *FairLearn$(c, \alpha)$* to ranking by applying the algorithm sequentially for each recommendation slot, while constraining the algorithm not to choose the same item twice for a given ranked list. We also

adapt *FairLearn* to contextual bandits by using LinUCB as underlying learning algorithm. More precisely, for the current timestep and slot, if the constraint is not violated, then the algorithm plays the item with the highest LinUCB upper confidence bound.

**Objectives** To illustrate the flexibility of our approach, we use algorithm FW-LinUCBRank to optimize three existing objectives which trade off between user utility and item fairness, in the form: $f(s) = s_{m+1} + \beta f^{\text{item}}(s_{1:m})$. *Gini* measures item inequality by the Gini index, as in (Biega et al., 2018; Morik et al., 2020; Do & Usunier, 2022), and *eq. exposure* uses the standard deviation (Do et al., 2021):

$$\textit{(Gini)} \ f^{\text{item}}(s) = \sum_{j=1}^{m} \frac{m-j+1}{m} s_j^{\uparrow} \qquad \textit{(eq. expo)} \ f^{\text{item}}(s) = -\frac{1}{m} \sqrt{\sum_{j=1}^{m} \left( s_j - \frac{1}{m} \sum_{j'=1}^{m} s_{j'} \right)^2} \tag{15}$$

Since *Gini* is nonsmooth, we apply the FW-LinUCBRank algorithm for nonsmooth $f$ with Moreau-Yosida regularization, presented in Section 3.3 and detailed in Appendix F.1 (we use $\beta_0 = 1$ in our experiments). To compute the gradient of the Moreau envelope $\bar{f}_t$, we use the algorithm of Do & Usunier (2022) which specifically applies to generalized Gini functions and top-$\bar{k}$ ranking.

We also study additive concave welfare functions (Do et al., 2021; Moulin, 2003) where $\alpha$ is a parameter controlling the degree of redistribution of exposure to the worse-off items:

$$\textit{(Welf)} \ f^{\text{item}}(s) = \sum_{j=1}^{m} s_j^{\alpha}, \ \ \alpha > 0 \tag{16}$$

### B.1.2 ADDITIONAL RESULTS

We now present additional results, which are obtained by repeating each simulation with 10 different random seeds.

**Dynamics** For the three objectives described, Figure 2 represents the values of the user and item objectives (left and middle), and the value of the objective $f$ (right) over time, achieved by the competing algorithms on *Lastfm-50*. We set $\beta = 0.5$ for all objectives and for *welf*, we set $\alpha = 0.5$. We observe that with this value of $\beta$, the item objective $f^{\text{item}}$ is given more importance in $f$ than the user utility.

We observe that for *Gini* and *welf*, *FW-LinUCBRank* achieves the highest value of $f$ across timesteps. This is because unlike *LinUCBRank*, it accounts for the item objective $f^{\text{item}}$. In both cases, *Unbiased-LinUCBRank* achieves a high value of $f$ over time but starts decreasing, after $10^4$ iterations for *Gini* and $5.10^5$ iterations for *welf*. This is because *Unbiased-LinUCBRank* is not designed to converge towards an optimum of $f$. For *eq. exposure*, when $\beta = 0.5$, *Unbiased-LinUCBRank* obtains surprisingly better values of $f$ than *FW-LinUCBRank*. Therefore, depending on the objective to optimize and the timeframe, *Unbiased-LinUCBRank* can be chosen as an alternative to *FW-LinUCBRank*. However, due to its lack of theoretical guarantees, it is more difficult to understand in which cases it may work, and for how many iterations. Furthermore, unlike *Unbiased-LinUCBRank*, *FW-LinUCBRank* can be chosen to optimise a wide variety of functions by varying the tradeoff parameter $\beta$ in all objectives, and $\alpha$ in *welf* to control the degree of redistribution. *Unbiased-LinUCBRank* does not have such controllability and flexibility.

Figure 3 shows the objective values for *Gini* and *welf* on *Lastfm-2k*. We observe similar results where *FW-LinUCBRank* converges more quickly than its competitors ($\approx 5,000$ iterations for *Gini* and $\approx 500$ iterations for *welf*) and obtains the highest values of $f$. For the first $10^5$ iterations of optimizing *Gini*, *Unbiased-LinUCBRank* obtains significantly lower values than *FW-LinUCBRank* on *welf*.

**Fairness trade-off for fixed** $T$ On the larger *Lastfm-2k* dataset, we study the tradeoffs between user utility and item inequality obtained by *FW-LinUCBRank* and *FairLearn* on Figure 4 after $T = 10^6$ rounds. The Pareto frontiers are obtained as follows: *FW-LinUCBRank* optimises for *Gini*, in which we vary $\beta$, and for *FairLearn* we vary the constraint value $c$ at fixed $\alpha = 1$. Figure 1 in Section 5 of the main paper illustrated the same Pareto frontier but for $5\times$ more iterations and on the smaller

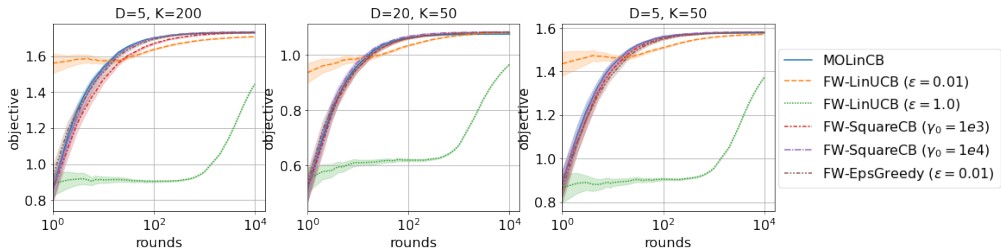

Figure 5: Multi-objective bandits: GGF value achieved on various synthetic environments.

*Lastfm-50* dataset. Although the algorithms might not have converged for this larger dataset, we observe that *FW-LinUCBRank* obtains better trade-offs than *FairLearn*, achieving higher user utility at all levels of inequality. We conclude that even in a setting with more items and shorter learning time, *FW-LinUCBRank* effectively reduces item inequality, at lower cost for user utility than the baseline.

### B.2 MULTI-ARMED CBCR: APPLICATION TO MULTI-OBJECTIVE BANDITS WITH GENERALIZED GINI FUNCTION

We provide the details and additional simulations on the task of optimizing the Generalized Gini aggregation Function (GGF) in multi-objective bandits (Busa-Fekete et al., 2017; Mehrotra et al., 2020). We remind that the goal is to maximize a GGF of the $D$-dimensional rewards, which is a nonsmooth concave aggregation function parameterized by nonincreasing weights $w_1 = 1 \geq \ldots \geq w_D \geq 0$: $f(s) = \sum_{i=1}^{D} w_i s_i^{\uparrow}$, where $(s_i^{\uparrow})_{i=1}^{D}$ denotes the values of $s$ sorted in increasing order.

Mehrotra et al. (2020) study the contextual bandit setting, motivated by music recommendation on Spotify with multiple metrics. They consider atomic actions $a_t \in \mathcal{A}$ (i.e., $\mathcal{A}$ is the canonical basis of $\mathbb{R}^K$) and a linear reward model: $\forall i \in [\![D]\!]$, $\exists \theta_i \in \mathbb{R}^d$, $\mathbb{E}_t[r_{t,i}] = \theta_i^\mathsf{T} x_t^\mathsf{T} a_t$. These are the same assumptions as described in Table 1 of Section 3.2 and in Appendix G.

GGFs are concave functions, but they are nondifferentiable. Therefore, we use the variant of our FW approach for nonsmooth $f$ (see Section 3.3), where we smooth the objective via Moreau-Yosida regularization with parameter $\beta_0 = 0.01$, using the algorithm of (Do & Usunier, 2022) to compute the gradients of the smooth approximations $f_t$.

**Algorithms**   In the main body, we evaluated two instantiations of our FW meta-algorithm, namely FW-LinUCB and FW-SquareCB. The level of exploration in FW-LinUCB is controlled by a variable $\epsilon$. More precisely, the exploration bonus is multiplied by $\sqrt{\epsilon}$, i.e. the UCBs are calculated as: $\hat{\theta}_{t-1,i}^\mathsf{T} x_{t,k} + \sqrt{\epsilon}\, \alpha_t(\delta) \|x_{t,k}\|_{V_{t-1}^{-1}}$. In FW-Square-CB, as detailed in Appendix H, the exploration is controlled by a sequence $(\gamma_t)_{t \geq 1}$, growing as $\sqrt{t}$ (higher $\gamma_t$ means less exploration). We set it to $\gamma_t = \gamma_0 \sqrt{t}$ with $\gamma_0 \in \{10^3, 10^4\}$.

In addition to the two algorithms presented in Section 5, to show the flexibility of our FW approach, we also implement FW-$\epsilon$-greedy, another instantiation of our FW algorithm which uses $\epsilon-$greedy as scalar bandit algorithm.

We compare our algorithms with MOLinCB of Mehrotra et al. (2020), an online gradient descent-style algorithm which was designed for this task, but was introduced without theoretical guarantees, as an extension of the MO-OGDE algorithm of Busa-Fekete et al. (2017) who study the non-contextual problem. We use the default parameters of MOLinCB recommended by Mehrotra et al. (2020).

**Environments**   Since the Spotify dataset of Mehrotra et al. (2020) is not publicly available, we only focus on their simulated, controlled environments. We reproduced these environments exactly as described in Appendix A of their paper. For completeness, we restate the protocol here: we draw a hidden parameter $\theta \in \mathbb{R}^{D \times d}$ uniformly at random in $[0, 1]$, and each element of a context-arm vector $x_{t,k}$ is drawn from $\mathcal{N}(\frac{1}{d}, \frac{1}{d^2})$. Given a context $x_t$ and arm $k_t$, the $D$-dimensional reward is generated

as a draw from $\mathcal{N}(\theta x_{t,k_t}, 0.01(\theta x_{t,k_t})^2)$. We choose $d = 10$ in the data generation and $\lambda = 0.1$ in the Ridge regression, as recommended by Mehrotra et al. (2018).

In Section 5 of the main body, we varied the number of objectives $D \in \{5, 20\}$ and set $K = 50$. Here we also experiment with $K = 200$ to see the effect of varying the number of arms. The GGF weights are set to $w_j = \frac{1}{2^{j-1}}$. Each simulation is repeated with 100 different random seeds.

**Results** The extended results, with more arms and algorithms, are depicted in Figure 5. We observe that FW-$\epsilon$-greedy achieves similar performance to the baseline MOLinCB, with small exploration $\epsilon = 0.01$. FW-SquareCB also achieves comparable performance to MOLinCB when there is little exploration, i.e. with $\gamma_0 = 10^4$ rather than $10^3$. This is coherent with our observation in Section 5 that FW-LinUCB obtains better performance when there is very little exploration on this environment from Mehrotra et al. (2018). Note that there is no forced exploration in their algorithm MOLinCB. Overall, we obtain qualitatively similar results when $K = 200$ compared to $K = 50$.

## C    Proofs of Section 2

In this section we give the missing details of Section 2. For completeness, we remind the definitions of Lipschitz-continuity and super-gradients in the next subsection. Then, we start in Section C.2 the analysis of the structure of the set $\mathcal{S}$ defined in Section 3 of the main paper, and more precisely its support function $g \mapsto \max_{s \in \mathcal{S}} g^{\mathsf{T}} s$. This contains new lemmas that are fundamental for the analysis throughout the paper, in particular in the proof of Lemma 9, which is given in Section C.3.

### C.1    Brief reminder on Lipschitz functions and super-gradients

We remind the following definitions. Let $D$ and $D'$ be two integers, and $f$ a function $f : \mathbb{R}^D \to \mathbb{R}^{D'}$. We have:

- *(Lipschitz continuity)* $f$ is $L$-Lipschitz continuous with respect to $\|.\|_2$ on a set $\mathcal{Z} \subseteq \mathbb{R}^D$ if
$$\forall z, z' \in \mathcal{Z}, \|f(z) - f(z')\|_2 \leq L\|z - z'\|_2. \tag{17}$$
- *(super-gradients)* If $f : \mathbb{R}^D \to \mathbb{R} \cup \{\pm\infty\}$, a super-gradient of $f$ at a point $z \in \mathbb{R}^D$ where $f(z) \in \mathbb{R}$ is a vector $g$ such that for all $z' \in \mathbb{R}^D$, $f(z') \leq f(z) + \langle g \,|\, z' - z \rangle$.

We remind the following results when $f : \mathbb{R}^D \to \mathbb{R} \cup \{\pm\infty\}$ is a proper closed concave function:

- $f$ has non-empty set of super-gradients at every point $z$ where $f(z) \in \mathbb{R}$,
- if $f$ is $L$-Lipschitz on $\mathcal{Z} \subseteq \mathbb{R}^D$ and $\mathcal{Z}$ is open, then for every $z \in \mathcal{Z}$ and every super-gradient $g$ of $f$ at $z$, we have $\|g\|_2 \leq L$.

The assumption of Lipschitz-continuity of $f$ on a set $\mathcal{Z}$ implicitly implies the assumption that $\mathcal{Z}$ is in the domain of $f$.

**Remark 1 (About our Lipschitzness assumptions)** *We use Lipschitzness over an open set containing $\mathcal{K}$ in Assumption A because we use boundedness of the super-gradients of $f$. In fact, a more precise alternative would be to require that super-gradients are bounded uniformly on $\mathcal{K}$ by $L$. We choose the Lipschitz formulation because we believe it is more natural.*

*As a side note, in assumption B, we use Lipschitzness of the gradients on $\mathcal{K}$, not on an open set containing $\mathcal{K}$. This is because smoothness in used in the ascent lemma (see Eq. 50), which uses Inequality 4.3 of Bottou et al. (2018), the proof of which directly uses Lipschitz-continuity of the gradients on $\mathcal{K}$ (Bottou et al., 2018, Appendix B), without relying on an argument of boundedness of gradients.*

### C.2    Preliminaries: the structure of the set $\mathcal{S}$

We denote by $x_{1:T} = (x_1, \ldots, x_T)$ a sequence of contexts of length $T$. Let

$$\mathcal{S} = \left\{ \mathbb{E}_{x \sim P}\left[ \mu(x)\overline{\pi}(x) \right] \middle| \overline{\pi} : \mathcal{X} \to \overline{\mathcal{A}} \right\} \tag{18}$$

$$\forall x_{1:T} \in \mathcal{X}^T, \mathcal{S}(x_{1:T}) = \left\{ \frac{1}{T} \sum_{t=1}^{T} \mu(x_t)\overline{\pi}(x_t) \middle| \overline{\pi} : \mathcal{X} \to \overline{\mathcal{A}} \right\} \tag{19}$$

It is straightforward to show that $\mathcal{S}(x_{1:T}) = \left\{ \frac{1}{T} \sum_{t=1}^T \mu(x_t)\pi_t \,\middle|\, (\pi_1, \ldots, \pi_T) \in \overline{\mathcal{A}}^T \right\}$. These sets are particularly relevant because of the following equality, for every $f : \mathbb{R}^D \to \mathbb{R} \cup \{\pm\infty\}$:

$$f^* = \sup_{\overline{\pi}:\mathcal{X}\to\overline{\mathcal{A}}} f\left( \mathbb{E}_{x\sim P}\left[ \mu(x)\overline{\pi}(x) \right] \right) = \sup_{s\in\mathcal{S}} f(s) \tag{20}$$

and

$$f_T^+ = \sup_{(\pi_t)_{t\in[\![T]\!]}\in\overline{\mathcal{A}}^T} f\left( \frac{1}{T} \sum_{t=1}^T \mu(x_t)\pi_t \right) = \sup_{s\in\mathcal{S}(x_{1:T})} f(s). \tag{21}$$

We study in this section the structure of these sets. We provide here the part of Assumption A that is relevant to this section:

**Assumption $\tilde{\mathbf{A}}$** $\mathcal{A}$ *is a compact subset of* $\mathbb{R}^K$ *and there is a compact convex set* $\mathcal{K} \subseteq \mathbb{R}^D$ *such that* $\forall (x,a) \in \mathcal{X} \times \mathcal{A}, \mu(x)a \in \mathcal{K}.$

We remind the following basic results from convex sets in Euclidian spaces that we use throughout the paper without reference:

**Lemma 6** *Let* $\mathcal{A}$ *be a compact subset of* $\mathbb{R}^K$. *We have:*
- *([Rockafellar & Wets, 2009](), Corollary 2.30) The convex hull* $\overline{\mathcal{A}}$ *of* $a$, *denoted by* $\overline{\mathcal{A}}$, *is compact.*
- *For every* $w \in \mathbb{R}^K, \max_{a\in\mathcal{A}} w^\intercal a = \max_{a\in\overline{\mathcal{A}}} w^\intercal a.$

The following lemma allows us to use maxima instead of suprema over $\mathcal{S}$ and $\mathcal{S}(x_{1:T})$. The proof of this lemma is deferred to Appendix J.1.

**Lemma 7** *Under Assumption* $\tilde{A}$, $\mathcal{S}$ *is compact and* $\forall T \in \mathbb{N}_*, \forall x_{1:T} \in \mathcal{X}^T, \mathcal{S}(x_{1:T})$ *is compact.*

The next result regarding the support functions of $\mathcal{S}$ and $\mathcal{S}(x_{1:T})$ is the key to our approach:

**Lemma 8** *Let* $w \in \mathbb{R}^D$ *and* $T \in \mathbb{N}_*$. *Under Assumption* $\tilde{A}$, *we have*

$$\mathbb{E}_{x_{1:T}\sim P^T}\left[ \max_{s\in\mathcal{S}(x_{1:T})} w^\intercal s \right] = \max_{s\in\mathcal{S}} w^\intercal s. \tag{22}$$

*Moreover, for every* $\delta \in (0,1]$, *we have with probability at least* $1 - \delta$:

$$\max_{s\in\mathcal{S}(x_{1:T})} w^\intercal s \leq \max_{s\in\mathcal{S}} w^\intercal s + \|w\|_2 D_{\mathcal{K}} \sqrt{\frac{2\ln\delta^{-1}}{T}}. \tag{23}$$

*The inequality* $\max_{s\in\mathcal{S}} w^\intercal s \leq \max_{s\in\mathcal{S}(x_{1:T})} w^\intercal s + \|w\|_2 D_{\mathcal{K}} \sqrt{\frac{2\ln\delta^{-1}}{T}}$ *also holds with probability* $1 - \delta$.

*Proof.* The first result is a direct consequence of the maximization of linear functions over the simplex. Using (20) with $f(s) = w^\intercal s$ and the linearity of expectations, we have

$$\max_{s\in\mathcal{S}} w^\intercal s = \max_{\overline{\pi}:\mathcal{X}\to\overline{\mathcal{A}}} \mathbb{E}_{x\sim P}\left[ w^\intercal \mu(x)\overline{\pi}(x) \right]. \tag{24}$$

The optimal policy given $w$, denoted by $\overline{\pi}^w$ is thus obtained by optimizing for every $x$ the dot product between $w^\intercal \mu(x) \in \mathbb{R}^K$ and $\overline{\pi}(x) \in \overline{\mathcal{A}} \subseteq \mathbb{R}^K$. Since, for each $x$, it is a linear optimization, we can find an optimizer in $\mathcal{A}$ (see Lemma 6), which gives:

$$\max_{s\in\mathcal{S}} w^\intercal s = \mathbb{E}_{x\sim P}\big[ \underbrace{w^\intercal \mu(x)\overline{\pi}^w(x)}_{\eta^w(x)} \big] \qquad \text{where } \overline{\pi}^w(x) \in \operatorname*{argmax}_{a\in\mathcal{A}} w^\intercal \mu(x)a, \tag{25}$$

where in the equation above we mean that $\overline{\pi}^w$ is a measurable selection of $x \mapsto \operatorname{argmax}_{a\in\mathcal{A}} w^\intercal \mu(x)a$.

For the same reason, we have $\max_{s\in\mathcal{S}(x_{1:T})} w^\intercal s = \frac{1}{T} \sum_{t=1}^T \eta^w(x_t)$. We obtain

$$\mathbb{E}_{x_{1:T}\sim P^T}\left[ \max_{s\in\mathcal{S}(x_{1:T})} w^\intercal s \right] = \mathbb{E}_{x_{1:T}\sim P^T}\left[ \frac{1}{T} \sum_{t=1}^T \eta^w(x_t) \right] = \mathbb{E}_{x\sim P}\left[ \eta^w(x) \right] = \max_{s\in\mathcal{S}} w^\intercal s. \tag{26}$$

which is the first equality.

For the high-probability inequality, let $X_t = \eta^w(x_t) - \mathbb{E}_{x \sim P}[\eta^w(x)]$. Since the $(x_t)_{t \in [\![T]\!]}$ are independent and identically distributed (i.i.d.), the variables $(X_t)_{t \in [\![T]\!]}$ are also i.i.d., and we have

$$|X_t| \le w^\intercal \bigg( \underbrace{\mu(x_t)\overline{\pi}^w(x_t)}_{\in \mathcal{K}} - \underbrace{\mathbb{E}_{x \sim P}[\mu(x)\overline{\pi}^w(x)]}_{\in \mathcal{K}} \bigg) \le \|w\|_2 D_\mathcal{K} \qquad \text{and } \mathbb{E}[X_t] = 0. \qquad (27)$$

Given $\delta \in (0, 1]$, Hoeffding's inequality applied to $\frac{1}{T} \sum_{t=1}^T X_t$ gives, with probability at least $1 - \delta$:

$$\max_{s \in \mathcal{S}(x_{1:T})} w^\intercal s - \max_{s \in \mathcal{S}} w^\intercal s = \frac{1}{T} \sum_{t=1}^T X_t \le \|w\|_2 D_\mathcal{K} \sqrt{\frac{2 \ln \delta^{-1}}{T}}. \qquad (28)$$

The reverse equation is obtained by applying Hoeffding's inequality to $-\frac{1}{T} \sum_{t=1}^T X_t$. $\qquad \square$

### C.3 Proof of Lemma 9

**Lemma 9** *Under Assumption A, $\forall T \in \mathbb{N}_*, \forall \delta \in (0, 1]$, we have, with probability at least $1 - \delta$:*

$$\left| f_T^+ - f^* \right| \le L D_\mathcal{K} \sqrt{\frac{2 \ln \frac{4e^2}{\delta}}{T}} \qquad \text{where } f_T^+ = \max_{(\pi_1, \ldots, \pi_T) \in \overline{\mathcal{A}}^T} f\bigg( \frac{1}{T} \sum_{t=1}^T \mu(x_t)\pi_t \bigg)$$

*We also have, with probability $1 - \delta$ over contexts, actions, and rewards:*

$$\left| f(s_T) - f(\hat{s}_T) \right| \le L D_\mathcal{K} \sqrt{\frac{2 \ln(2e^2 \delta^{-1})}{T}} \quad \text{where } s_T = \frac{1}{T} \sum_{t=1}^T \mu(x_t) a_t.$$

The first statement shows that the performance of the optimal non-stationary policy over $T$ steps converges to $f^*$ at a rate $O(1/\sqrt{T})$. Furthermore, measuring the algorithm's performance by expected rewards instead of observed rewards would also amount to a difference of order $O(1/\sqrt{T})$. This choice would lead to what is commonly referred to as a *pseudo*-regret. Since the worst-case regret of BCR is $\Omega(1/\sqrt{T})$ (Bubeck & Cesa-Bianchi, 2012), the previous lemma shows that the alternative definitions of regret would not substantially change our results.

*Proof.* We start with the first inequality.

We first prove that w.p. greater than $1 - \delta/2$, we have $f_T^+ \le f^* + L D_\mathcal{K} \sqrt{\frac{2 \ln \frac{2}{\delta}}{T}}$.

Since $f$ is continuous on $\mathcal{K}$ and since $\mathcal{S} \subseteq \mathcal{K}$ and $\mathcal{S}$ is compact by Lemma 7, there is $s^* \in \mathcal{S}$ such that $f^* = f(s^*)$. Similarly, since $\mathcal{S}(x_{1:T})$ is compact, there is $s_T^*$ such that $f(s_T^*) = \max_{s \in \mathcal{S}(x_{1:T})} f(s)$. Using (20), we need to prove that with probability at least $1 - \delta/2$, we have $f(s_T^*) \le f(s^*) + L D_\mathcal{K} \sqrt{\frac{2 \ln \frac{2}{\delta}}{T}}$.

Using the concavity of $f$, let $g^*$ be a supergradient of $f$ at $s^*$. We have

$$f(s_T^*) \le f(s^*) + \langle g^* \,|\, s_T^* - s^* \rangle \qquad (29)$$
$$\le f(s^*) + \max_{s \in \mathcal{S}(x_{1:T})} \langle g^* \,|\, s - s^* \rangle \qquad (30)$$

$$\implies \text{w.p. } \ge 1 - \delta/2: \quad f(s_T^*) \le f(s^*) + \underbrace{\max_{s \in \mathcal{S}} \langle g^* \,|\, s - s^* \rangle}_{\le 0 \text{ by def. of } s^*} + \|g^*\|_2 D_\mathcal{K} \sqrt{\frac{2 \ln \frac{2}{\delta}}{T}}$$

$$\text{(by Lemma 8)}$$

$$\le f(s^*) + L D_\mathcal{K} \sqrt{\frac{2 \ln \frac{2}{\delta}}{T}}. \qquad \text{(by the Lipschitz assumption)}$$

We now prove $f^* \leq f_T^+ + LD_\mathcal{K}\sqrt{\frac{2\ln\frac{4e^2}{\delta}}{T}}$ with probability at least $1 - \delta/2$.

Let $\overline{\pi}^* \in \operatorname{argmax}_{\overline{\pi}:\mathcal{X}\to\overline{\mathcal{A}}} f\left(\mathbb{E}_{x\sim P}\left[\mu(x)\overline{\pi}(x)\right]\right)$ (an optimal policy exists by Lemma 7). Denote by $(X_t = \mu(x_t)\overline{\pi}^*(x_t))_{t\in[\![T]\!]}$ a sequence of independent and identically distributed random variables obtained by sampling $x_t \sim P$.

We have $|X_t - \mathbb{E}X_t| \leq D_\mathcal{K}$ and $\mathbb{E}X_t = s^*$. By the Lipschitz property of $f$, we obtain

$$f(s^*) \leq f\left(\frac{1}{T}\sum_{t=1}^T X_t\right) + L\left\|\frac{1}{T}\sum_{t=1}^T X_t - s^*\right\|_2. \tag{31}$$

Using the version of Azuma's inequality for vector-valued martingale with bounded increments of Hayes (2005, Theorem 1.8) to obtain, for every $\epsilon > 0$:

$$\mathbb{P}\left(\frac{1}{D_\mathcal{K}}\left\|\frac{1}{T}\sum_{t=1}^T X_t - s^*\right\|_2 \geq \epsilon\right) \leq 2e^2 e^{-T\epsilon^2/2}. \tag{32}$$

Setting $\frac{\delta}{2} = 2e^2 e^{-T\epsilon^2/2}$ and solving for $\epsilon$ gives, with probability at least $1 - \delta/2$:

$$f^* \leq f\left(\frac{1}{T}\sum_{t=1}^T X_t\right) + LD_\mathcal{K}\sqrt{\frac{2\ln\frac{4e^2}{\delta}}{T}} \leq f_T^+ + LD_\mathcal{K}\sqrt{\frac{2\ln\frac{4e^2}{\delta}}{T}}. \tag{33}$$

For the second inequality: using $L$-Lipschitzness of $f$, the inequality is a direct consequence of the lemma below, which is itself a direct consequence of (Hayes, 2005, Theorem 1.8). $\qquad\square$

In the following lemma and its proof, we use the two following filtrations:

- $\mathfrak{F} = (\mathfrak{F}_t)_{t\in\mathbb{N}_*}$ where $\mathfrak{F}_t$ is the $\sigma$-algebra generated by $(x_1, a_1, r_1, \ldots, x_{t-1}, a_{t-1}, r_{t-1}, x_t)$,
- $\overline{\mathfrak{F}} = (\overline{\mathfrak{F}}_T)_{T\in\mathbb{N}_*}$ where $\overline{\mathfrak{F}}_T$ is the $\sigma$-algebra generated by $(x_1, a_1, r_1, \ldots, x_{t-1}, a_{t-1}, r_{t-1}, x_t, a_t)$.

Our setup implies that the process $(a_t)_{t\in\mathbb{N}_*}$ is adapted to $\mathfrak{F}$ while $(r_t)_{t\in\mathbb{N}_*}$ is adapted to $\overline{\mathfrak{F}}$.

**Lemma 10** *Under Assumption A, if the actions $(a_1, \ldots, a_T)$ define a process adapted to $(\mathfrak{F}_T)_{T\in\mathbb{N}}$, then, for every $T \in \mathbb{N}$, for every $\delta$, with probability $1 - \delta$, we have:*

$$\|s_T - \hat{s}_T\|_2 \leq D_\mathcal{K}\sqrt{\frac{2\ln\frac{2e^2}{\delta}}{T}} \tag{34}$$

*Proof.* Let $X_T = \sum_{t=1}^T r_t - \mu(x_t)a_t$. We have $\|X_T - X_{T-1}\|_2 \leq D_\mathcal{K}$, and $(X_T)_{T\in\mathbb{N}}$ is a martingale adapted to $(\overline{\mathfrak{F}}_T)_{T\in\mathbb{N}}$ satisfying $X_0 = 0$. We can then use the version of Azuma's inequality for vector-valued martingale with bounded increments of Hayes (2005, Theorem 1.8) to obtain, for every $\epsilon > 0$:

$$\mathbb{P}\left(\left\|\frac{X_T}{D_\mathcal{K}}\right\|_2 \geq \epsilon\right) \leq 2e^2 e^{-\epsilon^2/(2T)}. \tag{35}$$

Solving for $\epsilon$ gives the desired result. $\qquad\square$

## D  THE GENERAL TEMPLATE FRANK-WOLFE ALGORITHM

**A more general framework**  The analysis of the next sections is done within a more general famework than that of the main paper, which is described in Algorithm 2. Similarly to the main paper, the action is drawn according to $a_t \sim \mathfrak{A}(h_t, x_t, \delta')$ (Line 3 of Alg. 2). However, we allow for a generic choice of Frank-Wolfe iterate with respect to which we compute (an extension of) the scalar regret (presented in (36) below). The *update direction* is denoted by $\rho_t$ and is chosen according to a function $\mathfrak{U}(h_{t+1}, \delta')$, a companion function from $\mathfrak{A}(h_t, x_t, \delta')$. Note that the update direction is chosen given $h_{t+1} = (h_t, (x_t, a_t, r_t))$, the history after the actions and rewards have been taken.

---

**Algorithm 2:** Generic Frank-Wolfe algorithm for CBCR.

**input:** initial point $z_0 \in \mathcal{K}$, Approx. RLOO confidence parameter $\delta'$
1 **for** $t = 1 \dots T$ **do**
2  Observe $x_t \sim P$
3  Pull $a_t \sim \mathfrak{A}(h_t, x_t, \delta')$                                                   // Explore/exploit step
4  Observe reward $r_t \in \mathcal{K}$, update temporal average of observed rewards $\hat{s}_t$
5  Let $\rho_t = \mathfrak{U}(h_{t+1}, \delta')$                                                        // Generic Frank-Wolfe update
6  Update $z_t = z_{t-1} + \frac{1}{t}\left(\rho_t - z_{t-1}\right)$
7 **end**

---

**The proofs of the main paper apply to the special case of Alg. 2 where** $\forall t \geq 1, \rho_t = r_t$**.** We then have the FW iterate $z_t$ in Line 6 of the algorithm satisfy $\forall t \geq 1, z_t = \hat{s}_t$.

The reason we study this generalization is to show how our analysis applies in cases where the FW iterate is not the observed reward. In prior work on (non-contextual) BCR, Agrawal & Devanur (2014, Algorithm 4) use an upper-confidence approach and use the upper confidence on the expected reward as update direction. The generalization made by introducing $\mathfrak{U}(h_{t+1}, \delta')$ compared to the main paper allows for our analysis to encompass their approach.

We need to update Assumptions $A$ and $B$ to account for the fact that $\rho_t$ is used in place of $r_t$.

**Assumption $A'$** *$f$ is closed proper concave on $\mathbb{R}^D$ and $\mathcal{A}$ is a compact subset of $\mathbb{R}^K$. Moreover, there is a compact convex set $\mathcal{K} \subseteq \mathbb{R}^D$ such that*

- *(Bounded rewards and iterates) For all $t \in \mathbb{N}_*$, $r_t \in \mathcal{K}$ and $\rho_t \in \mathcal{K}$ with probability 1.*
- *(Local Lipschitzness) $f$ is $L$-Lipschitz continuous with respect to $\|.\|_2$ on an open set containing $\mathcal{K}$.*

**Assumption $B'$** *Assumption $A'$ holds and $f$ has $C$-Lipschitz-continuous gradients w.r.t. $\|.\|_2$ on $\mathcal{K}$.*

In Assumption $A$ we added $\mu(x_t)a_t \in \mathcal{K}$ for clarity, but it is not necessary since $\mu(x_t)a_t \in \mathcal{K}$ with probability 1 is implied by $r_t \in \mathcal{K}$ with probability 1. The difference between Assumption $A'$ and Assumption $A$ is to make sure that the updates $\rho_t$, and thus the iterates $z_t$ belong to $\mathcal{K}$ and are in the domain of definition of $f$. Notice that in the special case of $\rho_t = r_t$, Assumption $A'$ reduces to Assumption $A$ and, similarly, Assumption $B$ reduces to Assumption $B'$. We use the term *smooth* as a synonym of Lipschitz-continuous gradients.

**Analysis for (possibly) non-smooth objective functions**   We are going to present a single analysis that encompasses both the case where $f$ is smooth (Assumption $B$ of the main paper), and the case where $f$ may not be smooth, which we briefly discussed in Section 3.3. In order for our analysis to be agnostic to the type of smoothing used and to also encompass the case where $f$ is smooth, we propose the following assumption, where $(f_t)_{t \in \mathbb{N}}$ is a sequence of smooth approximations of $f$:

**Assumption E** *Assumption $A'$ holds and $\exists (\beta_0, L, M_1, M_2) \in \mathbb{R}_+^4$ such that $(f_t)_{t \in \mathbb{N}}$ satisfy:*

1. *$\forall t \in \mathbb{N}, f_t : \mathbb{R}^D \to \mathbb{R} \cup \{\pm\infty\}$ is proper closed concave on $\mathbb{R}^D$,*
2. *$\forall t \in \mathbb{N}, f_t$ is differentiable on $\mathcal{K}$ with $\sup_{z \in \mathcal{K}} \|\nabla f_t(z)\|_2 \leq L$, and $f_t$ is $\frac{\sqrt{t+1}}{\beta_0}$-smooth on $\mathcal{K}$,*
3. *$\forall t \in \mathbb{N}_*, \forall z \in \mathcal{K}, |f_t(z) - f_{t-1}(z)| \leq \frac{M_1}{t\sqrt{t}}$ and $|f_t(z) - f(z)| \leq \frac{M_2}{\sqrt{t}}$.*

Notice that any function $f$ satisfying Assumption $B$ with coefficient of smoothness $C$ satisfies Assumption $E$ with $\beta_0 = 1/C$, $M_1 = M_2 = 0$. Regarding non-smooth $f$, we discuss in more details in Appendix $F$ specific methods to perform this smoothing, including the Moreau envelope used in Section 3.3.

The generalization of the scalar regret takes into account both the approximation functions $(f_t)_{t \in \mathbb{N}}$ and the general update $z_t$:

$$R_T^{\text{gen}} = \sum_{t=1}^{T} \max_{a \in \mathcal{A}} \langle \nabla f_{t-1}(z_{t-1}) \,|\, \mu(x_t)a \rangle - \sum_{t=1}^{T} \langle \nabla f_{t-1}(z_{t-1}) \,|\, \rho_t \rangle + LT\|z_T - \hat{s}_T\|_2. \tag{36}$$

The general regret bound then takes the following form, where we distinguish between smooth and non-smooth $f$. Recall that $\tilde{C} = CD_{\mathcal{K}}^2/2$.

**Theorem 11** *Under Assumptions B', using $\forall T \in \mathbb{N}, f_T = f$.*

*For every $T \in \mathbb{N}$, every $z_0 \in \mathcal{K}$, every $\delta > 0$, Algorithm 2 satisfies, with probability at least $1 - \delta$:*

$$R_T \leq \frac{R_T^{\text{gen}} + LD_{\mathcal{K}}\sqrt{2T \ln \frac{1}{\delta}} + \tilde{C} \ln(eT)}{T} \tag{37}$$

**Theorem 12** *Under Assumptions E, for every $z_0 \in \mathcal{K}$, every $T \geq 1$ and every $\delta > 0$, Algorithm 2 satisfies, with probability at least $1 - \delta$:*

$$R_T \leq \frac{R_T^{\text{gen}}}{T} + \frac{\frac{D_{\mathcal{K}}^2}{\beta_0} + 4M_1 + 2M_2 + LD_{\mathcal{K}}\sqrt{2 \ln \frac{1}{\delta}}}{\sqrt{T}} \tag{38}$$

The proofs are given in Appendix E.

The worst-case regret of contextual bandits is $\Omega(\sqrt{T})$ (Bubeck & Cesa-Bianchi, 2012; Dani et al., 2008; Lattimore & Szepesvári, 2020), which gives a lower bound for the worst-case regret of CBCR in $\Omega(\frac{1}{\sqrt{T}})$. The dependencies on the problem parameters are all directly derived from the regret bounds $R_T^{\text{gen}}$ of the underlying scalar bandit algorithm (LinUCB, SquareCB, etc.). Therefore we obtain CBCR algorithms that are near minimax optimal as soon as $R_T^{\text{gen}} \leq O(\sqrt{T})$. The residual terms $O(\frac{1}{\sqrt{T}})$ terms are tied to the use of Azuma's inequality (Lemma 13) and FW analysis (using Lipschitz and smoothness parameters), and the dependencies to these parameters match usual convergence guarantees in optimization (Jaggi, 2013; Clarkson, 2010; Lan, 2013). As we rely on a worst-case analysis in deriving our reduction guarantees, it remains an open question whether problem-dependent optimal bounds could be recovered as well.

We make three remarks in order:

**Remark 2 (Why we need a specific result for smooth $f$)** *The result for $C$-smooth $f$ has a better dependency than the general result using $\beta_0 = 1/C$ ($\ln(eT)$ instead of $\sqrt{T}$), which makes a fundamental difference in practice if the smoothness coefficient is close to $\sqrt{T}$. This is why we keep the two results separate.*

**Remark 3 (Comparison to the smoothing as used by Agrawal & Devanur (2014))** *Agrawal & Devanur (2014, Thm 5.4) present an analysis for non-smooth $f$ where, at a high-level, they run the smooth algorithm using $f_T$ instead of a sequence $(f_t)_{t \in \mathbb{N}}$, and then apply the convergence bound for smooth $f$. Our analysis has two advantages:*

1. *Anytime bounds: our approach does not require the horizon to be known in advance.*
2. *Better bound: they obtain a bound on $\sqrt{\ln T/T}$ by suitably choosing the smoothing parameter, whereas we obtain a bound of $1/\sqrt{T}$. In practice, it may not make a difference if $\frac{R_T^{\text{gen}}}{T}$ is itself in $\sqrt{\ln T}/T$, but the advantage of our approach is clear as far as the analysis of FW for (C)BCR is concerned.*

**Remark 4 (About the confidence parameter $\delta'$ in $\mathfrak{A}(h_t, x_t, \delta')$ and $\mathfrak{U}(h_{t+1}, \delta')$)** *In practice, exploration/exploitation algorithms need a confidence parameter that defines the probability of their regret guarantee. For instance, in confidence-based approaches, it is the probability with which the confidence intervals are valid at every time step. In our case, it means that explicit upper bounds on $R_T^{\text{gen}}$ are of the form $\overline{R}^{\text{gen}}(T, \delta')$ which hold with probability $1 - \delta'$, where $\delta'$ is the confidence parameter in $\mathfrak{A}(h_t, x_t, \delta')$. Using the union bound, we obtain bounds of the form $R_T \leq \overline{R}^{\text{gen}}(T, \delta')/T + O\left(\sqrt{\frac{\ln(1/\delta)}{T}}\right)$ that are valid with probability $1 - \delta - \delta'$.*

*Note the difference in the roles of $\delta$ and $\delta'$: $\delta$ is not a parameter of the algorithm, it is only here to account for the randomization over contexts.*

# E PROOFS FOR SECTION 3 AND APPENDIX D

This section contains the proofs for the results of Section 3. All the proofs are made for the more general framework described in Appendix D. The framework of the paper can be recovered as the special case $\forall t \in \mathbb{N}, \rho_t = r_t$ and $z_t = \hat{s}_t$.

*Proof of Lemma 1.* Lemma 1 is the special case of Lemma 13 when $f$ is smooth. Note that every $f$ satisfying Assumption A satisfies the assumptions of Lemma 13. □

*Proof of Theorem 2.* Thm. 2 is a special case of Theorem 11 of Appendix D, using $\forall t \in \mathbb{N}, \rho_t = r_t$ and $z_t = \hat{s}_t$. The proof of Theorem 11 is given in Section E.1. □

**Lemma 13** *Assume that $\forall T, f_T$ is differentiable on $\mathcal{K}$ with $\forall z \in \mathcal{K}, \|\nabla f_T(z)\|_2 \leq L$. Then, for every $z \in \mathcal{K}$, we have:*

$$\mathbb{E}_{x \sim P}\big[\max_{a \in \mathcal{A}}\langle \nabla f_{t-1}(z) \,|\, \mu(x)a\rangle\big] = \max_{\overline{\pi}:\mathcal{X} \to \overline{\mathcal{A}}} \mathbb{E}_{x \sim P}\big[\langle \nabla f_{t-1}(z) \,|\, \mu(x)\overline{\pi}(x)\rangle\big] = \max_{s \in \mathcal{S}}\langle \nabla f_{t-1}(z_{t-1}) \,|\, s\rangle. \quad (39)$$

*Assume furthermore that $z_t$ is a function of contexts, actions and rewards up to time $t$. Let $a_t^* \in \underset{a \in \mathcal{A}}{\mathrm{argmax}}\langle \nabla f_{t-1}(z_{t-1}) \,|\, \mu(x_t)a\rangle$. For all $\delta \in (0,1]$, with probability at least $1 - \delta$, we have:*

$$\sum_{t=1}^{T}\max_{s \in \mathcal{S}}\langle \nabla f_{t-1}(z_{t-1}) \,|\, s - \mu(x_t)a_t^*\rangle \leq LD_{\mathcal{K}}\sqrt{2T\ln\frac{1}{\delta}} \quad (40)$$

*Proof.* Let $z \in \mathcal{K}$. We first prove (39). The first equality in (39) comes from the maximization over functions over the simplex with a linear objective: define

$$\overline{\pi}_t^* : \mathcal{X} \mapsto \overline{\mathcal{A}} \qquad \text{such that } \overline{\pi}_t^*(x) \in \underset{a \in \mathcal{A}}{\mathrm{argmax}}\langle \nabla f_{t-1}(z_{t-1}) \,|\, \mu(x)a\rangle, \quad (41)$$

using some arbitrary tie-breaking rule when the argmax is not unique. We have, for every policy $\overline{\pi}$:

$$\mathbb{E}_{x \sim P}\big[\langle \nabla f_{t-1}(z) \,|\, \mu(x)\overline{\pi}(x)\rangle\big] \leq \mathbb{E}_{x \sim P}\big[\max_{a \in \mathcal{A}}\langle \nabla f_{t-1}(z) \,|\, \mu(x)a\rangle\big] \quad (42)$$

$$\implies \max_{\overline{\pi}:\mathcal{X} \to \overline{\mathcal{A}}} \mathbb{E}_{x \sim P}\big[\langle \nabla f_{t-1}(z) \,|\, \mu(x)\overline{\pi}(x)\rangle\big] \leq \mathbb{E}_{x \sim P}\big[\langle \nabla f_{t-1}(z) \,|\, \mu(x)\overline{\pi}_t^*(x)\rangle\big]. \quad (43)$$

On the other hand, it is clear that

$$\mathbb{E}_{x \sim P}\big[\langle \nabla f_{t-1}(z) \,|\, \mu(x)\overline{\pi}_t^*(x)\rangle\big] \leq \max_{\overline{\pi}:\mathcal{X} \to \overline{\mathcal{A}}} \mathbb{E}_{x \sim P}\big[\langle \nabla f_{t-1}(z) \,|\, \mu(x)\overline{\pi}(x)\rangle\big], \quad (44)$$

and we get the first equality of (39).

The second equality in (39) holds by the definition of $\mathcal{S}$ since for every policy $\overline{\pi}$, we have

$$\mathbb{E}_{x \sim P}\big[\langle \nabla f_{t-1}(z) \,|\, \mu(x)\overline{\pi}(x)\rangle\big] = \langle \nabla f_{t-1}(z) \,|\, \mathbb{E}_{x \sim P}\big[\mu(x)\overline{\pi}(x)\big]\rangle. \quad (45)$$

We now prove (40). Let $\big(\mathbb{E}_t[.]\big)_{t \geq 1}$ be the conditional expectations with respect to the filtration $\widetilde{\mathfrak{F}} = (\widetilde{\mathfrak{F}}_t)_{t \geq 1}$ where $\mathfrak{F}_t$ is the $\sigma$-algebra generated by $(x_t', a_t', r_t')_{t' \in [\![t-1]\!]}$, i.e., contexts, actions and rewards up to time $t-1$, so that we have:

$$\mathbb{E}_t\big[\langle \nabla f_{t-1}(z_{t-1}) \,|\, \mu(x_t)\overline{\pi}_t^*(x_t)\rangle\big] = \mathbb{E}_{x \sim P}\big[\langle \nabla f_{t-1}(z_{t-1}) \,|\, \mu(x)\overline{\pi}_t^*(x)\rangle\big]. \quad (46)$$

Using (39) gives $\mathbb{E}_t\big[\langle \nabla f_{t-1}(z_{t-1}) \,|\, \mu(x_t)\overline{\pi}_t^*(x_t)\rangle\big] = \max_{s \in \mathcal{S}}\langle \nabla f_{t-1}(z_{t-1}) \,|\, s\rangle$, from which we obtain

$$\max_{s \in \mathcal{S}}\langle \nabla f_{t-1}(z_{t-1}) \,|\, s - \mu(x_t)a_t^*\rangle$$

$$= \mathbb{E}_t\big[\langle \nabla f_{t-1}(z_{t-1}) \,|\, \mu(x_t)\overline{\pi}_t^*(x_t)\rangle\big] - \langle \nabla f_{t-1}(z_{t-1}) \,|\, \mu(x_t)\overline{\pi}_t^*(x_t)\rangle$$

$X_T = \sum_{t=1}^{T}\max_{s \in \mathcal{S}}\langle \nabla f_{t-1}(z_{t-1}) \,|\, s - \mu(x_t)a_t^*\rangle$ thus defines a martingale adapted to $\mathfrak{F}$, and, using $X_0 = 0$, we have, for all $t$:

$$|X_t - X_{t-1}| \leq L \sup_{\substack{s \in \mathcal{S} \\ x \in \mathcal{X} \\ a \in \mathcal{A}}}\|s - \mu(x)a\|_2 \leq L \sup_{z,z' \in \mathcal{K}}\|z - z'\|_2 \leq LD_{\mathcal{K}}. \quad (47)$$

The results then follows from Azuma's inequality. □

The next lemma is the main technical tool of the paper. The proof is not technically difficult given the previous result, using the telescoping sum approach of the proof of Lemma 12 of Berthet & Perchet (2017) and organizing the residual terms.

**Lemma 14** *Under Assumption E, denote* $\forall t \in \mathbb{N}$, $f_t^* = \max_{s \in \mathcal{S}} f_t(s)$, *and* $\tilde{R}_t(z) = f_t^* - f_t(z)$.

*Let* $\overline{C}(T)$, $\overline{F}^*(T)$ *in* $\mathbb{R} \cup \{+\infty\}$ *such that,* $\forall T \in \mathbb{N}_*$, *we have:*

$$\sum_{t=1}^{T} \frac{D_{\mathcal{K}}^2}{2} \frac{C_{t-1}}{t} \leq \overline{C}(T), \qquad \sum_{t=1}^{T} t\big(\tilde{R}_t(z_t) - \tilde{R}_{t-1}(z_t)\big) \leq \overline{F}^*(T) \qquad (48)$$

*And let* $\overline{B}(T) = \overline{C}(T) + \overline{F}^*(T)$. *Then, for all* $z_0 \in \mathcal{K}$, $\forall T, \forall \delta > 0, \forall \delta' > 0$, *Algorithm 2 satisfies, with probability at least* $1 - \delta$:

$$f_T^* - f_T(\hat{s}_T) \leq \frac{\overline{B}(T) + R_T^{\mathrm{gen}} + LD_{\mathcal{K}}\sqrt{2T \ln \frac{1}{\delta}}}{T} \qquad (49)$$

*Proof.* We start with the standard ascent lemma using bounded curvature on $\mathcal{K}$ (Bottou et al., 2018, Inequality 4.3), denoting $\tilde{C}_T = \frac{D_{\mathcal{K}}^2}{2} C_T$:

$$f_{t-1}(z_t) \geq f_{t-1}(z_{t-1}) + \frac{1}{t}\langle \nabla f_{t-1}(z_{t-1}) \,|\, \rho_t - z_{t-1}\rangle - \frac{\tilde{C}_{t-1}}{t^2} \qquad (50)$$

$$f_{t-1}^* - f_{t-1}(z_t) \leq f_{t-1}^* - f_{t-1}(z_{t-1}) - \frac{1}{t}\langle \nabla f_{t-1}(z_{t-1}) \,|\, \rho_t - z_{t-1}\rangle + \frac{\tilde{C}_{t-1}}{t^2} \qquad (51)$$

Let us denote by $g_t = \nabla f_{t-1}(z_{t-1})$ and let $a_t^* \in \mathrm{argmax}_{a \in \mathcal{A}}\langle g_t \,|\, \mu(x_t)a\rangle$. We first decompose the middle term:

$$\langle g_t \,|\, \rho_t - z_{t-1}\rangle = \max_{s \in \mathcal{S}}\langle g_t \,|\, s - z_{t-1}\rangle - \max_{s \in \mathcal{S}}\langle g_t \,|\, s - \mu(x_t)a_t^*\rangle - \langle g_t \,|\, \mu(x_t)a_t^* - \rho_t\rangle \qquad (52)$$

$$\geq f_{t-1}^* - f_{t-1}(z_{t-1}) - \underbrace{\max_{s \in \mathcal{S}}\langle g_t \,|\, s - \mu(x_t)a_t^*\rangle}_{\alpha_t} - \underbrace{\langle g_t \,|\, \mu(x_t)a_t^* - \rho_t\rangle}_{\rho_t}$$

$$\text{(by (53) below)}$$

Where the last inequality uses the concavity of $f_t$: for all $s_{t-1}^* \in \mathrm{argmax}_{s \in \mathcal{S}} f_{t-1}(s)$, we have:

$$f_{t-1}^* - f_{t-1}(z_{t-1}) \leq \langle \nabla f_{t-1}(z_{t-1}) \,|\, s_{t-1}^* - z_{t-1}\rangle \leq \max_{s \in \mathcal{S}}\langle \nabla f_{t-1}(z_{t-1}) \,|\, s - z_{t-1}\rangle \qquad (53)$$

and thus we get

$$f_{t-1}^* - f_{t-1}(z_t) \leq \big(f_{t-1}^* - f_{t-1}(z_{t-1})\big)\Big(1 - \frac{1}{t}\Big) + \frac{1}{t}(\alpha_t + \rho_t) + \frac{\tilde{C}_{t-1}}{t^2} \qquad (54)$$

$$\Longrightarrow \qquad t\tilde{R}_t(z_t) \leq (t-1)\tilde{R}_{t-1}(z_{t-1}) + \alpha_t + \rho_t + \frac{\tilde{C}_{t-1}}{t} + t\big(\tilde{R}_t(z_t) - \tilde{R}_{t-1}(z_t)\big) \qquad (55)$$

$$\Longrightarrow \qquad T\tilde{R}_T(z_T) \leq \sum_{t=1}^{T}\alpha_t + \sum_{t=1}^{T}\rho_t + \sum_{t=1}^{T}t\big(\tilde{R}_t(z_t) - \tilde{R}_{t-1}(z_t)\big) + \sum_{t=1}^{T}\frac{\tilde{C}_{t-1}}{t} \qquad (56)$$

Using the Lipschitz property for $f_T$, we finally obtain

$$T\tilde{R}_T(\hat{s}_T) \leq \underbrace{\sum_{t=1}^{T}\alpha_t + \sum_{t=1}^{T}\rho_t + TL\|z_T - \hat{s}_T\|_2}_{\substack{\leq LD_{\mathcal{K}}\sqrt{2T\ln(1/\delta)} + R_T^{\mathrm{gen}} \\ \text{w.p. } \geq 1-\delta \text{ by (36) and Lemma 13.}}} + \underbrace{\sum_{t=1}^{T}t\big(\tilde{R}_t(z_t) - \tilde{R}_{t-1}(z_t)\big)}_{\leq \overline{F}^*(T)} + \underbrace{\sum_{t=1}^{T}\frac{\tilde{C}_{t-1}}{t}}_{\leq \overline{C}(T)} \qquad (57)$$

Which is the desired result. $\qquad\square$

### E.1 PROOFS OF THE MAIN RESULTS

We now prove the results of Appendix D.

*Proof of Theorem 11.* First, notice that since $f$ differentiable on $\mathcal{K}$ (since it is smooth) and since both $z_T$ and $\frac{1}{T}\sum_{t=1}^{T}\mu(x_t)a_t$ are in $\mathcal{K}$, using $\forall t, f_t = f$, we have $R_T = f^* - f(\hat{s}_T) = f_T^* - f_T(\hat{s}_T)$. Using the notation of Lemma 14, we then have $\overline{C}(T) = 0$ and $D(T) = 0$. Also:

$$\sum_{t=1}^{T}\frac{D_{\mathcal{K}}^2}{2}\frac{C_t}{t} = \sum_{t=1}^{T}\frac{\tilde{C}}{t} \leq \tilde{C}(\ln(t)+1) \tag{58}$$

The result then follows from Lemma 14. $\qquad\square$

*Proof of Theorem 12.* Using the notation of Lemma 14, we specify $\overline{C}(T), \overline{F}^*(T)$ in turn.

$$\sum_{t=1}^{T}\frac{D_{\mathcal{K}}^2}{2}\frac{C_{t-1}}{t} = \sum_{t=1}^{T}\frac{D_{\mathcal{K}}^2}{2\beta_0\sqrt{t}} \leq \frac{D_{\mathcal{K}}^2}{\beta_0}\sqrt{T}. \tag{59}$$

For $\overline{F}^*(T)$, we decompose $\tilde{R}_t(z_t) - \tilde{R}_{t-1}(z_t)$ into two terms:

$$\tilde{R}_t(z_t) - \tilde{R}_{t-1}(z_t) = f_t^* - f_{t-1}^* + f_{t-1}(z_t) - f_t(z_t) \leq \frac{2M_1}{t\sqrt{t}} \tag{60}$$

Using $\sum_{t=1}^{T}\frac{1}{\sqrt{t}} \leq 2\sqrt{T}$, we obtain $\overline{F}^*(T) \leq 2M_1\sum_{t=1}^{T}\frac{t}{t\sqrt{t}} \leq 4M_1\sqrt{T}$. Lemma 14 gives

$$f_T^* - f_T(\hat{s}_T) \leq R_T^{\text{gen}} + \frac{\frac{D_{\mathcal{K}}^2}{\beta_0} + 4M_1 + LD_{\mathcal{K}}\sqrt{2\ln(\delta^{-1}/2)}}{\sqrt{T}} \tag{61}$$

To finish the proof, notice that:

$$\left|f^* - f(\hat{s}_T) - \left(f_T^* - f_T(\hat{s}_T)\right)\right| \leq 2\sup_{z'\in\mathcal{K}}|f_T(z') - f(z')| \leq \frac{2M_2}{\sqrt{T}}. \tag{62}$$

The result follows from (61) and (62) using:

$$R_T = f^* - f(\hat{s}_T) \leq f_T^* - f_T(\hat{s}_T) + \frac{2M_2}{\sqrt{T}}. \tag{63}$$

$\qquad\square$

## F SMOOTH APPROXIMATIONS OF NON-SMOOTH FUNCTIONS

We discuss here in more details two specific smoothing techniques: the Moreau envelope, also called Moreau-Yosida regularization in Section F.1, then randomized smoothing in Section F.2. As in Appendices D and E, we focus on the general framework described in Algorithm 2.

*Proof of Theorem 3.* Usinh Theorem 12 above and Lemma 16 below gives the result since

$$\frac{D_{\mathcal{K}}^2}{\beta_0} + 4M_1 + 2M_2 = \frac{D_{\mathcal{K}}^2}{\beta_0} + 3L^2\beta_0 = LD_{\mathcal{K}}\left(\frac{D_{\mathcal{K}}}{L\beta_0} + 3\frac{L\beta_0}{D_{\mathcal{K}}}\right). \tag{64}$$

$\qquad\square$

### F.1 SMOOTHING WITH THE MOREAU ENVELOPE

For functions that are non-smooth, we propose first a smoothing technique based on the Moreau envelope, following the approach described by Lan (2013). Let $f : \mathbb{R}^D \to \mathbb{R} \cup \{\pm\infty\}$ be a closed proper concave function. The Moreau envelope (or Moreau-Yosida regularization) of $f$ with parameter $\beta_T$ (Rockafellar & Wets, 2009, Def. 1.22) is defined as

$$\tilde{f}_\beta(z) = \max_{y \in \mathbb{R}^D} \left( f(y) - \frac{1}{2\beta} \|y - z\|_2^2 \right). \tag{65}$$

For $\beta > 0$, let the proximal operator $\text{prox}_\beta = \text{argmax}_{y \in \mathbb{R}^D} \tilde{f}_\beta(y)$. The basic properties of the Moreau envelope (Rockafellar & Wets, 2009, Th. 2.26) are that if $f : \mathbb{R}^D \to \mathbb{R} \cup \{\pm\infty\}$ is an upper semicontinuous, proper concave function then $\tilde{f}_\beta$ is concave, finite everywhere, continuously differentiable with $\frac{1}{\beta}$-Lipschitz gradients. We also have that the proximal operator $\text{prox}_\beta$ is well-defined (the argmax is attained in a single point) and we have

$$\nabla \tilde{f}_\beta(z) = \frac{1}{\beta}\big(z - \text{prox}_\beta(z)\big). \tag{66}$$

It is immediate to prove the following inequalities for every $z \in \mathbb{R}^n$ and every $\beta > 0$:

$$f(z) \leq \tilde{f}_\beta(z) \leq f(\text{prox}_\beta(z)). \tag{67}$$

The following properties of the Moreau envelope (See (Yurtsever et al., 2018, Appendix A.1) and (Thekumparampil et al., 2020, Lemma 1)) are key to the main results:

**Lemma 15** *Let $\beta > 0$, $f : \mathbb{R}^D \to \mathbb{R} \cup \{\pm\infty\}$ be a proper closed concave function, and $\mathcal{Z} \subseteq \mathbb{R}^D$ be a convex set such that $f$ is locally $L$-Lipschitz-continuous on $\mathcal{Z}$. Then:*

• $\forall z \in \mathcal{Z}$ *such that* $\text{prox}_\beta(z) \in \mathcal{Z}$, *we have* $\|z - \text{prox}_\beta(z)\| \leq L\beta$ *and:*

$$\tilde{f}_\beta(z) - \frac{L^2\beta}{2} \leq f(z) \leq \tilde{f}_\beta(z). \tag{68}$$

• $\forall z \in \mathcal{Z}$ *such that* $\text{prox}_\beta(z) \in \mathcal{Z}$, $\forall \beta > 0$ *and* $\beta' > 0$, *we have:*

$$\tilde{f}_\beta \leq \tilde{f}_{\beta'} + \frac{1}{2}\Big(\frac{1}{\beta'} - \frac{1}{\beta}\Big)\big\|z - \text{prox}_\beta(z)\big\|_2^2 \leq \frac{L^2\beta}{2}\Big(\frac{\beta}{\beta'} - 1\Big) \tag{69}$$

We reformulate the lemma above in the language of Appendix D:

**Lemma 16** *Under Assumption A, assuming furthermore that $f$ is $L$-Lipschitz on $\mathbb{R}^D$.*

*Let $f_t = \tilde{f}_{\beta_t}$ with $\beta_t = \frac{\beta_0}{\sqrt{t+1}}$. Then $f$ and $(f_t)_{t \in \mathbb{N}}$ satisfy Assumption E with the corresponding values of $\beta_0$ and $L$, $M_2 = \frac{L^2\beta_0}{2}$ and $M_1 = \frac{L^2\beta_0}{2}$.*

*Proof.* By Lemma 15, $f_t$ is $L$-Lipschitz on $\mathbb{R}^D$ for every $t$, and we have $M_2 = \frac{L^2\beta_0}{2}$. Moreover, Lemma 15 also gives $0 \leq f_{t-1}(z) - f_t(z) \leq \frac{L^2\beta_0}{2t}(\sqrt{t+1} - \sqrt{t}) \leq \frac{L^2\beta_0}{2t\sqrt{t}}$. and thus $M_1 = \frac{L^2\beta_0}{2}$. $\square$

### F.2 RANDOMIZED SMOOTHING

We now describe the randomized smoothing technique (Lan, 2013; Nesterov & Spokoiny, 2017; Duchi et al., 2012; Yousefian et al., 2012), which consists in convolving $f$ with a probability density function $\Lambda$. Following Lan (2013) who combines Frank-Wolfe with randomized smoothing for nonsmooth optimization, we present our results with $\Lambda$ as the random uniform distribution in the $\ell_2$-ball $\{z \in \mathbb{R}^D : \|z\|_2 \leq 1\}$. Let $\beta > 0$ and $\xi$ a random variable with density $\Lambda$. Then the randomized smoothing approximation of $f$ is defined as:

$$f_\beta(z) := \mathbb{E}_\Lambda[f(x + \beta\xi)] = \int_{\mathbb{R}^D} f(x + \beta y)\Lambda(y)dy. \tag{70}$$

Following (Lan, 2013; Duchi et al., 2012), we abuse notation and take the "gradient" of $f$ inside integrals and expectation below, because $f$ is almost-everywhere differentiable since it is concave. We restate the following well-known properties of randomized smoothing (see e.g., (Yousefian et al., 2012, Lemma 8)):

**Lemma 17** *Let $\beta > 0$ and $f_\beta$ be defined as in Eq. (70).*

- $\forall z \in \mathcal{K}, f(z) \leq f_\beta(z) \leq f(z) + L\beta.$
- $f_\beta$ *is $L$-Lipschitz continuous over $\mathcal{K}$.*
- $f_\beta$ *is continuously differentiable and its gradient is $\frac{L\sqrt{D}}{\beta}$-Lipschitz continuous.*
- $\forall z \in \mathcal{K}, \nabla f_\beta(z) = \mathbb{E}[\nabla f(z + \beta\xi)].$

We obtain the following results, stated in the language of Theorem 12 of Appendix D.

**Lemma 18** *Under Assumption A, assuming furthermore that $f$ is $L$-Lipschitz on $\mathbb{R}^D$.*

*For $t \geq 1$, let $f_t = f_{\beta_t}$ with $\beta_t = \frac{D^{\frac{1}{4}} D_\mathcal{K}}{\sqrt{t+1}}$, and let $\beta_0 = \frac{\sqrt{D} D_\mathcal{K}}{L}$.*

*Then $f$ and $(f_t)_{t \in \mathbb{N}}$ satisfy Assumption E with the corresponding values of $\beta_0$ and $L$, $M_2 = LD^{\frac{1}{4}} D_\mathcal{K}$ and $M_1 = 2LD^{\frac{1}{4}} D_\mathcal{K}$.*

*Proof.* By Lemma 17, $f_t$ is $L$-Lipschitz on $\mathbb{R}^D$ for every $t$, so that $f_t$ has $L$-bounded gradient. Moreover, with this definition of $\beta_0$, $f_t$ is $\frac{\sqrt{t+1}}{\beta_0}$-smooth.

We have $M_2 = LD^{\frac{1}{4}} D_\mathcal{K}$ because:

$$|f_t(z) - f(z)| = |\mathbb{E}[f(z + \beta_t\xi)] - \mathbb{E}[f(z)]| \leq \mathbb{E}[|f(z + \beta_t\xi) - f(z)|] \leq \mathbb{E}[\|L\beta_t\xi\|_2] \leq \frac{LD^{\frac{1}{4}} D_\mathcal{K}}{\sqrt{t}} \tag{71}$$

We also have $M_1 = 2LD^{\frac{1}{4}} D_\mathcal{K}$ because:

$$|f_{t-1} - f_t| \leq \mathbb{E}[|f(x + \beta_{t-1}\xi) - f(x + \beta_t\xi)|] \leq L|\beta_{t-1} - \beta_t|\,\mathbb{E}[\|\xi\|_2] \tag{72}$$

$$= LD^{\frac{1}{4}} D_\mathcal{K}\left(\frac{1}{\sqrt{t}} - \frac{1}{\sqrt{t+1}}\right) \leq \frac{2LD^{\frac{1}{4}} D_\mathcal{K}}{t^{\frac{3}{2}}}. \tag{73}$$

$\square$

# G  FW-LINUCB: UPPER-CONFIDENCE BOUNDS FOR LINEAR BANDITS WITH $K$ ARMS

In this section, we have:

- a finite action space $\mathcal{A}$ which is the canonical basis of $\mathbb{R}^K$, i.e., we focus on the multi-armed bandit setting
- $\mathcal{X} \subseteq \mathbb{R}^{d \times K}$, where $d$ is the dimension of the feature space. Given $x \in \mathcal{X}$, the feature representation of arm $a \in \mathcal{A}$ is given by the matrix-vector product $xa$,
- Given a matrix $\theta \in \mathbb{R}^{D \times d}$, we denote by $\|\theta\|_F$ the frobenius norm of $\theta$, i.e., $\|\theta\|_F = \|\text{flatten}(\theta)\|_2$.

In addition, we make here the following linear assumption on the rewards:

**Assumption F** *There is $\theta \in \mathbb{R}^{D \times d}$ such that $\|\theta\|_F \leq D_\theta$ such that $\forall x \in \mathcal{X}, \mu(x)a = \theta xa$. Moreover, there is $D_\mathcal{X} > 0$ such that $\sup_{\substack{x \in \mathcal{X} \\ a \in \mathcal{A}}} \|xa\|_2 \leq D_\mathcal{X}$.*

We perform the analysis under Assumption E, which is the more general we have. In particular, we assume that we have access to a sequence $(f_t)_{t \in [\![T]\!]}$ of smooth approximations of $f$. We focus on the special case of Algorithm 2 that is described in the main paper, i.e., where $\rho_T = r_t$.

---

**Algorithm 3:** FW-linUCB: linear CBCR with $K$ arms.

---

    **input :** $\delta' > 0, \lambda > 0, \hat{s}_0 \in \mathcal{K}$ $V_0 = \lambda \boldsymbol{I}_{dD}, y_0 = \boldsymbol{0}_{dD}, \hat{\theta}_0 = \boldsymbol{0}_{dD}$

1  **for** $t = 1, \ldots$ **do**

2     Observe context $x_t \sim P$, $x_t \in \mathbb{R}^{d \times K}$

3     $g_t \leftarrow \nabla f_{t-1}(\hat{s}_{t-1}), \tilde{x}_t \leftarrow [g_{t,0}x_t; \ldots; g_{t,D}x_t]$

4     $\forall i \in [\![K]\!], \hat{u}_{t,i} \leftarrow \hat{\theta}_{t-1}^{\mathsf{T}}\tilde{x}_{t,i} + \alpha_t(\frac{\delta'}{2})\|\tilde{x}_{t,i}\|_{V_{t-1}^{-1}}$    // see (75) and (76) for def. of $\|.\|_{V_{t-1}^{-1}}$ and $\alpha_t$.

5     $a_t \leftarrow \operatorname{argmax}_{a \in \mathcal{A}} \hat{u}_t a$

6     Observe reward $r_t$, let $\tilde{r}_t = g_t^{\mathsf{T}} r_t$

7     Update $\hat{s}_t \leftarrow \hat{s}_{t-1} + \frac{1}{t}(r_t - \hat{s}_{t-1})$

8     $V_t \leftarrow V_{t-1} + (\tilde{x}_t a_t)(\tilde{x}_t a_t)^{\mathsf{T}}, \;\; y_t \leftarrow y_{t-1} + \tilde{r}_t \tilde{x}_t a_t$ and $\hat{\theta}_t \leftarrow V_t^{-1} y_t$    // regression

9  **end**

---

**The algorithm.** As hinted in Section 3.2, FW-LinUCB applies the LinUCB algorithm (Abbasi-Yadkori et al., 2011), designed for scalar-reward contextual bandits with adversarial contexts and stochastic rewards, to the following extended rewards and contexts, where we use $[.;.]$ to denote the vertical concatenation of matrices and $g_t = \nabla f_{t-1}(\hat{s}_{t-1})$:

- $\tilde{x}_t \in \mathbb{R}^{Dd \times K}$ is the extended context with entries $\tilde{x}_t = [g_{t,0}x_t; \ldots; g_{t,D}x_t] \in \mathbb{R}^{Dd \times K}$, so that the feature vector of action $a$ at time $t$ is $\tilde{x}_t a$;
- $\tilde{r}_t = g_t^{\mathsf{T}} r_t$ is the scalar observed reward,
- $\tilde{\theta} = \operatorname{flatten}(\theta) \in \mathbb{R}^{dD}$ is the ground-truth parameter vector and $\tilde{\mu}(x) = \tilde{\theta}^{\mathsf{T}} \tilde{x}_t$ is the average reward function.

Notice that under assumption A and F, denoting

$$\widetilde{\mathcal{X}} = \big\{ [g_{t,0}x_t; \ldots; g_{t,D}x_t] : \|g\|_2 \le L, x \in \mathcal{X} \big\} \qquad \text{and } D_{\widetilde{\mathcal{X}}} = \max_{\substack{\tilde{x} \in \widetilde{\mathcal{X}} \\ a \in \mathcal{A}}} \|\tilde{x}a\|_2, \qquad (74)$$

we have $\forall t, \tilde{x}_t \in \widetilde{\mathcal{X}}$ with probability 1 and $D_{\widetilde{\mathcal{X}}} \le LD_{\mathcal{X}}$. Moreover, $|\tilde{r}_t - \tilde{\mu}(x_t)a_t| \le LD_{\mathcal{K}}$, which implies in particular that for every $t \in [\![T]\!]$, $\tilde{r}_t$ is $LD_{\mathcal{K}}/2$-subgaussian.

Given this notation, the FW-LinUCB algorithm is LinUCB applied to the scalar-reward bandit problem above. The algorithm is summarized in Algorithm 3 for completeness, where $\lambda$ is the regularization parameter of the ridge regression, $\hat{\theta}_t$ is the current regression parameters, the matrix $V_t$ and the vector $y_t$ are incremental computations of the relevant matrices to compute $\hat{\theta}_t$. The crucial part of the algorithm is Line 3 which defines an upper confidence bound on $\tilde{\mu}(x_t)a$, denoted by $\hat{u}_t \in \mathbb{R}^K$ and defined by:

$$\forall i \in [\![K]\!], \hat{u}_{t,i} = \hat{\theta}_{t-1}^{\mathsf{T}}\tilde{x}_{t,i} + \alpha_t(\delta'/2)\|\tilde{x}_{t,i}\|_{V_{t-1}^{-1}} \qquad \text{where } \|\tilde{x}_{t,i}\|_{V_{t-1}^{-1}} = \sqrt{\tilde{x}_{t,i}^{\mathsf{T}} V_{t-1}^{-1} \tilde{x}_{t,i}}, \qquad (75)$$

and $\alpha_t$ is defined according to Theorem 2 of Abbasi-Yadkori et al. (2011):

$$\alpha_t(\delta') = \frac{LD_{\mathcal{K}}}{2}\sqrt{dD \ln \Big( \frac{1 + TD_{\widetilde{\mathcal{X}}}^2/\lambda}{\delta'} \Big)} + \sqrt{\lambda}D_\theta. \qquad (76)$$

Under Assumption E, we have with probability $\ge 1 - \delta'/2$: $\forall t \in \mathbb{N}_*, \hat{u}_t a \ge \tilde{\mu}(x_t)a$ (Abbasi-Yadkori et al., 2011, Theorem 2).

**The result.** Let $\tilde{d} = dD$. The regret bound of LinUCB (Abbasi-Yadkori et al., 2011, Theorem 3) and Azuma inequality give:

**Theorem 19** *Under Assumption E, for every $T \in \mathbb{N}_*$, for every $\delta' > 0$, Algorithm 3 satisfies, with probability at least $1 - \delta'$:*

$$R_T^{\text{scal}} \le 4\sqrt{T\tilde{d}\log(1 + TD_{\widetilde{\mathcal{X}}}/\tilde{d})}\Big( \sqrt{\lambda}D_\theta + \frac{LD_{\mathcal{K}}}{2}\sqrt{2\ln(2/\delta') + \tilde{d}\ln\big(1 + TD_{\widetilde{\mathcal{X}}}/(\lambda\tilde{d})\big)} \Big) \qquad (77)$$

$$+ LD_{\mathcal{K}}\sqrt{2\ln(2/\delta')}.$$

---

**Algorithm 4:** FW-SquareCB: contextual bandits with concave rewards and regression oracles

**input :** initial point $\hat{s}_0 \in \mathcal{K}$, exploration parameters $(\gamma_t)_{t\in\mathbb{N}}$. $\mathcal{A}$ is the canonical basis of $\mathbb{R}^K$.

1 **for** $t = 1 \dots$ **do**
2    Observe $x_t \sim P$
3    Compute $\hat{\mu}_t(x_t)$ using RegSq                              // see (81)
4    Let $g_t = \nabla f_{t-1}(\hat{s}_{t-1})$ and $\underline{\hat{\mu}}_t = g_t^\intercal \hat{\mu}_t(x_t) \in \mathbb{R}^K$
5    Let $\underline{a}_t \in \underset{a\in\mathcal{A}}{\arg\max}\, \underline{\hat{\mu}}_t^\intercal a$ and $\underline{\hat{\mu}}_t^* = \underline{\hat{\mu}}_t \underline{a}_t$           // use arbitrary tie breaking rule
6    Let $\forall a \in \mathcal{A}, \mathfrak{A}_t(a) = \begin{cases} \frac{1}{K + \gamma_t\left(\hat{\mu}_t^* - \hat{\mu}_t^\intercal a\right)} & \text{if } a \neq \underline{a}_t \\ 1 - \sum_{\substack{a\in\mathcal{A} \\ a \neq \underline{a}_t}} \mathfrak{A}_t(a) & \text{if } a = \underline{a}_t \end{cases}$    // Exploration/exploitation step
7    Draw $a_t \sim \mathfrak{A}_t$                                   // Action taken at time step $t$
8    Observe reward $r_t$ and update $\hat{s}_t = \hat{s}_{t-1} + \frac{1}{t}(r_t - \hat{s}_{t-1})$
9 **end**

---

*Proof.* Recall that as noted in (6) We decompose the scalar regret $R_T^{\text{scal}}$ into a pseudo regret and a residual term:

$$R_T^{\text{scal}} = \underbrace{\sum_{t=1}^{T} \max_{a\in\mathcal{A}} \tilde{\mu}(\tilde{x}_t)^\intercal a - \sum_{t=1}^{T} \tilde{\mu}(\tilde{x}_t)^\intercal a_t}_{\text{pseudo-regret}} + \sum_{t=1}^{T} \underbrace{\left(\tilde{\mu}(\tilde{x}_t)^\intercal a_t - \tilde{r}_t\right)}_{X_t} \tag{78}$$

The pseudo-regret term is bounded using Theorem 3 by Abbasi-Yadkori et al. (2011). The result applies as-is, except that they assume rewards $|\theta^\intercal \tilde{x}_t| \leq 1$, which is not the case here. The bound is still valid without changes, as in our case we have $|\max_{a\in\mathcal{A}} \tilde{\mu}(\tilde{x}_t)^\intercal a - \tilde{\mu}(\tilde{x}_t)^\intercal a_t| \leq LD_\mathcal{K}$. The steps in the proof where they use the assumption $|\theta^\intercal \tilde{x}_t| \leq 1$ is below Equation 7 (Abbasi-Yadkori et al., 2011, Appendix C), which in our notation and our assumption can be written as:

$$\max_{a\in\mathcal{A}} \tilde{\mu}(\tilde{x}_t)^\intercal a - \tilde{\mu}(\tilde{x}_t)^\intercal a_t \leq \min\left(2\alpha_t(\delta'/2)\|\tilde{x}_t a_t\|_{V_{t-1}^{-1}}, LD_\mathcal{K}\right) \tag{79}$$

$$\leq 2\alpha_t(\delta'/2)\min(\|\tilde{x}_t a_t\|_{V_{t-1}^{-1}}, 1) \tag{80}$$

where the first inequality comes from Abbasi-Yadkori et al. (2011) and the second one is true in our case because $2\alpha_t(\delta') \geq LD_\mathcal{K}$. From here on, the proof of Abbasi-Yadkori et al. (2011)'s regret bound follows the same as the original result.[11] Theorem 3 from Abbasi-Yadkori et al. (2011) gives us the first term of the regret bound of the theorem, which is true with probability at least $1 - \delta'/2$ in our case because we use $\alpha_t(\delta'/2)$.

For the rightmost term, let $\overline{\mathfrak{F}} = \left(\overline{\mathfrak{F}}_t\right)_{t\in\mathbb{N}_*}$ be the filtration where $\overline{\mathfrak{F}}_t$ is the $\sigma$-algebra generated by $(x_1, a_1, r_1, \dots, x_{t-1}, a_{t-1}, r_{t-1}, x_t, a_t)$. Then $(X_t)_{t\in\mathbb{N}_*}$ is a martingale difference sequence adapted to $\overline{\mathfrak{F}}$ with $|X_t| \leq LD_\mathcal{K}$. By Azuma's inequality, we have $\sum_{t=1}^{T} X_t \leq LD_\mathcal{K}\sqrt{2T\overline{k}\ln\frac{2}{\delta'}}$ with probability $1 - \delta'/2$. The final result holds using a union bound. $\qquad\square$

*Bound of Table 1.* The bound is obtained by keeping the main dependencies in $T, \tilde{d}, L$ and $D_\mathcal{K}$, ignoring the dependencies in $\lambda$ and $D_\theta$, and using the fact that $D_{\widetilde{\mathcal{X}}} \leq LD_\mathcal{K}$ (as described below (74)). $\qquad\square$

## H   FW-SQUARECB: CBCR WITH GENERAL REWARD FUCTIONS

The SquareCB algorithm was recently proposed by Foster & Rakhlin (2020) for zero-regret contextual multi-armed bandit *with general reward functions*, based on the notion of online regression oracles.

---

[11]In short, they have different bounds, one involving the varance of $r_t$ and the other one involving average rewards $\tilde{\mu}(\tilde{x})$. We assume rewards $r_T$ are uniformly bounded in $\mathcal{K}$, so we do not have to deal with two different quantities in our bounds and have $LD_\mathcal{K}$ everywhere.

They propose, for single-reward contextual bandits with adversarial contexts and stochastic rewards, a generic randomized exploration scheme that delegates learning to an online regression algorithm. Their exploration/exploitation strategy then has (bandit) regret bounded as a function of the online regret of the regression algorithm. In this section, we extend the SquareCB approach to the case of CBCR. The main interest of this section is that by building on the work of Foster & Rakhlin (2020), we obtain at nearly no cost an algorithm for general reward functions for multi-armed CBCR problems.

This section shows how to extend this algorithm to our setting of concave rewards. To simplify the notation, we consider the case of finite $K$ with atomic actions, i.e., $|\mathcal{A}| = K$. Our algorithm is based on an oracle for multi-dimensional regression RegSq, which provides approximate values for $\mu$:

$$\forall T, \forall x \in \mathcal{X}, \ \ \hat{\mu}_T(x) = \text{RegSq}\big(x, (x_1, a_1, r_1, \ldots, a_{T-1}, r_{T-1})\big). \tag{81}$$

The key assumption is that the problem is realizable and that RegSq has bounded regret:

**Assumption G** *There is a function $T \mapsto R_{\text{oracle}}(T) \in \mathbb{R}$, non-decreasing in $T$,*[12] *and $\Phi$, a class of functions from $\mathcal{X}$ to $\mathbb{R}^{D \times K}$ such that, for every $T \in \mathbb{N}$:*

1. *(Realizability) $\mu \in \Phi$,*
2. *(Regret bound) For every $(x_t, a_t, r_t)_{t \in [\![T]\!]}, \in (\mathcal{X} \times \mathcal{A} \times \mathcal{K})^T$, we have:*

$$\sum_{t=1}^{T} \big\|\hat{\mu}_t(x_t)a_t - r_t\big\|_2^2 - \inf_{\phi \in \Phi} \sum_{t=1}^{T} \big\|\phi(x_t)a_t - r_t\big\|_2^2 \leq R_{\text{oracle}}(T). \tag{82}$$

3. *For every $(x_t, a_t, r_t)_{t \in [\![T]\!]} \in (\mathcal{X} \times \mathcal{A} \times \mathcal{K})^T, \hat{\mu}_T(x_T)a_T \in \mathcal{K}.$*

Assumption G is the counterpart for multidimensional regression of Assumptions 1 and 2a of Foster & Rakhlin (2020), which are the basis of the original SquareCB algorithm.

**Remark 5 (The "informal" assumption used in Table 1)** *Notice that in Table 1, we describe an "informal" version of this assumption, which reads $\sum_{t=1}^{T}\big\|\hat{\mu}_t(x_t)a_t - \mu(x_t)a_t\big\|_2^2 \leq R_{\text{oracle}}(T)$, which is the counterpart for multi-dimensional regression of Assumption 2b by Foster & Rakhlin (2020). Our choice in the table was to simplify the presentation, as this assumption is shorter. Our analysis is also valid under this alternative assumption. Our proofs are made under Assumption G because it is more widely applicable (more discussion of these assumptions can be found in (Foster & Rakhlin, 2020)).*

Algorithm 4 describes how SquareCB principles apply to our framework. We use the framework of the main paper, or, equivalently, the special case of Algorithm 2 where $\forall t \in \mathbb{N}, \rho_t = r_t$ and $z_t = \hat{s}_t$. Note that the algorithm is parameterized by $(\gamma_t)_{t \in \mathbb{N}_*}$ instead of the desired confidence level $\delta'$ to make the analysis more general. Theorem 20 gives a formula for $\gamma_t$ as a function of the desired confidence $\delta'$. As for the previous sections, we describe the algorithm for the general case of smooth approximations of $f$, using $\nabla f_{t-1}$ rather than $\nabla f$ in Line 4 of the algorithm.

At time step $t$, the regression oracle provides an estimate of $\mu(x_t)$, then the algorithm computes $\mathfrak{A}_t$, with a larger probability for the action which maximizes $a \mapsto \langle \nabla f(\hat{s}_{t-1}) \,|\, \hat{\mu}_t(x_t)a \rangle$. The exact formula for these probabilities $\mathfrak{A}_t$ follow the original SquareCB algorithm, with the exception that we use an iteration-dependent $\gamma_t$ instead of a constant $\gamma$.[13]

The main result of this section is the following (see Section H.2 and the next section for intermediate lemmas):

**Theorem 20** *Let $\delta' > 0$. For every $t \in \mathbb{N}_*$, let $\gamma_t = \frac{2}{L}\sqrt{\frac{tK}{R_{\text{oracle}}(t) + 8D_{\mathcal{K}}^2 \ln \frac{4t^2}{\delta'}}}$. Then, under Assumptions E and G, Algorithm 4 satisfies, with probability at least $1 - \delta'$ :*

$$R_T^{\text{gen}} \leq 4L\sqrt{KT\big(R_{\text{oracle}}(T) + 8D_{\mathcal{K}}^2 \ln \frac{4T^2}{\delta'}\big)} + LD_{\mathcal{K}}\sqrt{2T \ln \frac{2}{\delta'}} \tag{83}$$

---

[12]Monotonicity of $R_{\text{oracle}}$ is not required in (Foster & Rakhlin, 2020). We use it in (94) below to deal with time-dependent $\gamma_t$. Meaningful $R_{\text{oracle}}(T)$ are non-decreasing with $T$ since they bound a cumulative regret.

[13]Throughout the paper, we chose to provide anytime bounds rather than bounds that depend on horizon-dependent parameters. The analysis with fixed $\gamma$ is easier.

Recall that Assumption B is a special case of E when $\rho_t = r_t$, as we are here. Thus, the bound on $R_T^{\text{gen}}$ is the same irrespective of whether we use the algorithm for smooth $f$ (in which case $R_T^{\text{scal}} = R_T^{\text{gen}}$) or with smooth approximations (in which case $R_T^{\text{scal,sm}} = R_T^{\text{gen}}$). This is because only the Lipschitzness of $(f_t)_{t \in \mathbb{N}}$ is used in the analysis of $R_T^{\text{gen}}$ for FW-SquareCB.

The following result is a direct corollary of Theorem 20, and gives the order of magnitude we obtain for smooth $f$. Obtaining a similar for smooth approximations of $f$, using Theorem 12 instead of Theorem 11 is straightforward.

*Proof of the FW-SquareCB regret bound of Table 1.* We apply the bound obtained by Theorem 20 within the bound of Theorem 11, using $\delta' := 2\delta/3$ and $\delta := \delta/3$. We obtain:

$$R_T \leq \frac{4L\sqrt{KT\big(R_{\text{oracle}}(T) + 8D_{\mathcal{K}}^2 \ln \frac{12t^2}{\delta}\big)} + 2LD_{\mathcal{K}}\sqrt{2T \ln \frac{3}{\delta}} + \tilde{C}\ln(eT)}{T}. \tag{84}$$

The bound given in the theorem uses the sub-additivity of $\sqrt{.}$ to group the terms in $\sqrt{\ln \delta^{-1}}$ for better readability. $\square$

The proof of Theorem 20 is decomposed into two subsections: in the next subsection, we make the necessary adaptations to the SquareCB analysis to account for multi-dimensional regression. This proof follows essentially the same steps as the original analysis of SquareCB. There are only two changes:

- We use multi-dimensional regression instead of scalar regression, while we need to bound a scalar regret. There is an additional step to go from the scalar regret to the multi-dimensional regression, but it turns out there is no added difficulty (see first line of the proof of Lemma 23).
- For coherence with the overall bounds of the paper, we use an anytime analysis using an increasing sequence of $(\gamma_t)_{t \in [\![T]\!]}$, instead of a fixed exploration parameter $\gamma$ that needs be tuned for a specific horizon determined *a priori*. This introduces a bit more difficulty, where the main tool is Lemma 24. Our choice of anytime bound is more for coherence in the presentation of the paper than an intended contribution.

  Nonetheless, what we gain with our anytime bound is that the exploration parameter $\gamma$ does not depend on a fixed horizon. What we lose, however, is that we need a high-probability bound on cumulative errors based on $R_{\text{oracle}}(t)$ that is valid for every $t$ (see Lemma 23), while the "fixed gamma" case only requires this bound to hold for the horizon $T$. This is the reason for the $\ln T$ factor in our bound, which is not present in the original paper.

In the next sections, we use the following notation:

$$g_t = \nabla f_{t-1}(\hat{s}_{t-1}), \qquad \underline{\mu}_t = g_t^{\mathsf{T}}\mu_t(x_t), \qquad \underline{\mu}_t^* = \max_{a \in \mathcal{A}} \underline{\mu}_t a, \tag{85}$$

$$\underline{\hat{\mu}}_t = g_t^{\mathsf{T}}\hat{\mu}_t(x_t), \qquad \underline{\hat{\mu}}_t^* = \max_{a \in \mathcal{A}} \underline{\hat{\mu}}_t a. \tag{86}$$

## H.1 ADAPTATION OF SQUARECB PROOF TO CBCR

In the SquareCB paper, Foster & Rakhlin (2020) study high probability bounds on a different type of regret, based on average rewards associated to the actions $\mu(x_t)a_t$ rather than observed rewards $r_t$. However, this difference has little influence since we can start with the following inequality, which is similar to (Foster & Rakhlin, 2020, Lemma 2).

**Lemma 21** *Under Assumption E, for every $T \in \mathbb{N}_*$, every $\delta' > 0$, Algorithm 4 satisfies*

$$\sum_{t=1}^{T} \big(\underline{\mu}_t^* - g_t^{\mathsf{T}}r_t\big) \leq \sum_{t=1}^{T} \mathbb{E}_{a \sim \mathfrak{A}_t}\big[\underline{\mu}_t^* - \underline{\mu}_t^{\mathsf{T}}a\big] + LD_{\mathcal{K}}\sqrt{2T \ln(1/\delta')}. \tag{87}$$

*Proof.* The proof is by Azuma's inequality. Let $\mathfrak{F} = (\mathfrak{F}_t)_{t \in \mathbb{N}_*}$ be the filtration where $\mathfrak{F}_t$ is the $\sigma$-algebra generated by $(x_1, a_1, r_1, \ldots, x_{t-1}, a_{t-1}, r_{t-1}, x_t)$, and let us denote $X_T = \sum_{t=1}^{T} \mathbb{E}_{a \sim \mathfrak{A}_t}\big[\underline{\mu}_t^{\mathsf{T}}a\big] - g_t^{\mathsf{T}}r_t$. Then, $(X_T)_{T \in \mathbb{N}}$ is a martingale adapted to filtration $\mathfrak{F}$ and satisfies $|X_t - X_{t-1}| \leq LD_{\mathcal{K}}$. We obtain the result by noticing that $X_T = \sum_{t=1}^{T} \big(\underline{\mu}_t^* - g_t^{\mathsf{T}}r_t\big) - \sum_{t=1}^{T} \mathbb{E}_{a \sim \mathfrak{A}_t}\big[\underline{\mu}_t^* - \underline{\mu}_t^{\mathsf{T}}a\big]$ and applying Azuma's inequality to $X_T$. $\square$

Notice that the difference between (Foster & Rakhlin, 2020, Lemma 2) and our Lemma 21 is that we consider the randomization over actions and rewards, while they only consider the randomization over actions because they study average rewards. However, since it does not change the upper bound on the variations of the martingale, this additional randomness does not change the bound.

The next step is the fundamental step in the proof of the original SquareCB algorithm. Even though the notation differ slightly from the original paper, the proof is the same as in (Foster & Rakhlin, 2020, Appendix B):

**Lemma 22** *(Foster & Rakhlin, 2020, Lemma 3) For every $t \in \mathbb{N}_*$, the choice of $\gamma_t$ and $\mathfrak{A}(h_t, x_t, \delta')$ of Algorithm 4 guarantees:*

$$\mathbb{E}_{a \sim \mathfrak{A}_t} \left[ \underline{\mu}_t^* - \underline{\mu}_t^{\mathsf{T}} a \right] \leq \frac{2K}{\gamma_t} + \frac{\gamma_t}{4} \mathbb{E}_{a \sim \mathfrak{A}_t} \left[ \left( \hat{\underline{\mu}}_t^{\mathsf{T}} a - \underline{\mu}_t^{\mathsf{T}} a \right)^2 \right]. \tag{88}$$

The last step of these preliminary lemmas is to relate the cumulative expected error to the oracle regret bound. We use here the same proof as (Foster & Rakhlin, 2020, Lemma 2). We then have:

**Lemma 23** *Under Assumption E, for every $\delta' > 0$, Algorithm 4 satisfies, w.p. at least $1 - \delta'$:*

$$\forall T \in \mathbb{N}_*, \sum_{t=1}^{T} \mathbb{E}_{a \sim \mathfrak{A}_t} \left[ \left( \hat{\underline{\mu}}_t^{\mathsf{T}} a - \underline{\mu}_t^{\mathsf{T}} a_t \right)^2 \right] \leq 2L^2 R_{\text{oracle}}(T) + 16 L^2 D_{\mathcal{K}}^2 \ln \frac{2T^2}{\delta'} \tag{89}$$

*Proof.* We first notice that $\sum_{t=1}^{T} \mathbb{E}_{a \sim \mathfrak{A}_t} \left[ \left( \hat{\underline{\mu}}_t^{\mathsf{T}} a - \underline{\mu}_t^{\mathsf{T}} a_t \right)^2 \right] \leq L^2 \sum_{t=1}^{T} \mathbb{E}_{a \sim \mathfrak{A}_t} \left[ \left\| \hat{\mu}(x_t) a - \mu(x_t) a \right\|_2^2 \right]$. We then apply the same steps as in the proof of (Foster & Rakhlin, 2020, Lemma 2) to $\sum_{t=1}^{T} \mathbb{E}_{a \sim \mathfrak{A}_t} \left[ \left\| \hat{\mu}(x_t) a - \mu(x_t) a \right\|_2^2 \right]$ (which we do not reproduce here) to obtain: for every every $T \in \mathbb{N}$, every $\delta'_T > 0$, with probability at least $1 - \delta'_T$:

$$\sum_{t=1}^{T} \mathbb{E}_{a \sim \mathfrak{A}_t} \left[ \left( \hat{\underline{\mu}}_t^{\mathsf{T}} a - \underline{\mu}_t^{\mathsf{T}} a_t \right)^2 \right] \leq 2L^2 R_{\text{oracle}}(T) + 16 L^2 D_{\mathcal{K}}^2 \ln \frac{1}{\delta'_T} \tag{90}$$

Let $\delta' > 0$. Applying a union bound and taking $\delta'_t = \frac{\delta'}{2t^2}$ so that $\sum_{t=1}^{T} \delta'_t \leq \frac{\pi^2}{12} \delta' \leq \delta'$, we obtain the desired result. $\square$

Notice the $\log T$ factor in the bound, which appears because the bound is valid for all time steps. This is because we propose anytime convergence bounds, with the exploration parameter that decreases with time, whereas (Foster & Rakhlin, 2020) only prove their result in the case where the exploration parameter is chosen for a specific horizon.

As the main first step for the final result, we need these two lemmas which are the main technical steps to our anytime bound. The proof is deferred to Appendix J.2

**Lemma 24** *Let $(\lambda_t)_{t \in \mathbb{N}} \in \mathbb{R}_+^T$ be a sequence of non-negative numbers, denote $\Lambda_T = \sum_{t=1}^{T} \lambda_t$ and let $(\overline{\Lambda}_T)_{T \in \mathbb{N}}$ such that $\forall T \in \mathbb{N}, \overline{\Lambda}_T > 0$ and $\overline{\Lambda}_T \geq \Lambda_T$.*

$$\sum_{t=1}^{T} \frac{\lambda_t}{\sqrt{\overline{\Lambda}_t}} \leq 2\sqrt{\overline{\Lambda}_T}. \tag{91}$$

We get the following corollary

**Lemma 25** *Let $R'_{\text{oracle}}(T, \delta') = 2L^2 R_{\text{oracle}}(T) + 16 L^2 D_{\mathcal{K}}^2 \ln \frac{2T^2}{\delta'}$. Under the conditions of Lemma 23, assume that there is $\gamma_0 > 0$ such that $\forall t \in [\![T]\!], \gamma_t = \gamma_0 \sqrt{\frac{t}{R'_{\text{oracle}}(t, \delta')}}$. Then, for every $\delta' > 0$, Algorithm 4 satisfies, w.p. at least $1 - \delta'$:*

$$\sum_{t=1}^{T} \gamma_t \mathbb{E}_{a \sim \mathfrak{A}_t} \left[ \left( \hat{\underline{\mu}}_t^{\mathsf{T}} a - \underline{\mu}_t^{\mathsf{T}} a \right)^2 \right] \leq 2\gamma_0 \sqrt{T R'_{\text{oracle}}(T, \delta')}. \tag{92}$$

*Proof.* Using $\gamma_t \leq \gamma_0 \sqrt{\frac{T}{R'_{\mathrm{oracle}}(t,\delta')}}$, the sum on the left hand side of (92) has the form of Lemma 24 multiplied by $\gamma_0 \sqrt{T}$, with probability $1 - \delta'$ by Lemma 23. The result thus follows from applying both Lemmas. □

## H.2 FINAL RESULT

*Proof of Theorem 20.* Notice that the value of $\gamma_t$ given in the theorem is equal to

$$\gamma_t = 2\sqrt{\frac{2tK}{R'_{\mathrm{oracle}}(t,\delta'/2)}}. \tag{93}$$

Using this formula, we have

$$\sum_{t=1}^{T} \frac{2K}{\gamma_t} = \sqrt{\frac{K}{2}} \sum_{t=1}^{T} \sqrt{\frac{R'_{\mathrm{oracle}}(t,\delta'/2)}{t}} \leq \sqrt{\frac{R'_{\mathrm{oracle}}(T,\delta'/2)K}{2}} \sum_{t=1}^{T} \frac{1}{\sqrt{t}} \tag{94}$$

$$\leq \sqrt{2KTR'_{\mathrm{oracle}}(T,\delta'/2)}. \tag{95}$$

Where the first line comes from the monotonicity of $R_{\mathrm{oracle}}(T)$ of Assumption G.

Using Lemmas 22 and 25, we thus have, with probability $1 - \delta'/2$:

$$\sum_{t=1}^{T} \mathbb{E}_{a \sim \mathfrak{A}_t} \left[ \underline{\mu}_t^* - \underline{\mu}_t^\intercal a \right] \leq 2\sqrt{2KTR'_{\mathrm{oracle}}(T,\delta'/2)}. \tag{96}$$

Using a union bound and Lemma 21, we obtain, with probability at least $1 - \delta'$:

$$\sum_{t=1}^{T} \left( \underline{\mu}_t^* - g_t^\intercal r_t \right) \leq 2\sqrt{2KTR'_{\mathrm{oracle}}(T,\delta'/2)} + LD_{\mathcal{K}}\sqrt{2T\ln\frac{2}{\delta'}}. \tag{97}$$

□

## I FW-LINUCBRANK: CBCR FOR FAIR RANKING WITH LINEAR CONTEXTUAL BANDITS

---

**Algorithm 5:** FW-linUCBRank: linear contextual bandits for fair ranking.

**input :** $\delta' > 0, \lambda > 0, \hat{s}_0 \in \mathcal{K}$ $V_0 = \lambda \boldsymbol{I}_d, y_0 = \boldsymbol{0}_d, \hat{\theta}_0 = \boldsymbol{0}_d$

1 **for** $t = 1, \ldots$ **do**
2    Observe context $x_t \sim P$
3    $\forall i, \hat{v}_{t,i} \leftarrow \hat{\theta}_{t-1}^\intercal x_{t,i} + \alpha_t(\frac{\delta'}{3}) \|x_{t,i}\|_{V_{t-1}^{-1}}$      // UCB on $v_i(x_t)$, see Lem. 26 for def. of $\alpha_t$
4    $a_t \leftarrow \text{top-}\overline{k}\{\frac{\partial f_{t-1}}{\partial z_{m+1}}(\hat{s}_{t-1})\hat{v}_{t,i} + \frac{\partial f_{t-1}}{\partial z_i}(\hat{s}_{t-1})\}_{i=1}^{m}$      // FW linear optimization step
5    Observe exposed items $e_t \in \{0,1\}^m$ and user feedback $c_t \in \{0,1\}^m$
6    Update $\hat{s}_t \leftarrow \hat{s}_{t-1} + \frac{1}{t}(r_t - \hat{s}_{t-1})$
7    $V_t \leftarrow V_{t-1} + \sum_{i=1}^{m} e_{t,i}x_{t,i}x_{t,i}^\intercal, \quad y_t \leftarrow y_{t-1} + \sum_{i=1}^{m} c_{t,i}x_{t,i}$ and $\hat{\theta}_t \leftarrow V_t^{-1}y_t$      // regression
8 **end**

---

In this section and following the previous sections, we analyze Algorithm 5 under Assumption E, which is more general than the bound proposed in the main paper, which used Algorithm 1 under Assumption B. The only difference in the algorithms is the use of $f_{t-1}$ instead of $f$ in Line 4 of Algorithm 5. This allows us to provide the algorithm for both smooth and non-smooth objective functions $f$.

The bound is decomposed into two parts: we describe the results for online regression within our observation model for ranking in the next subsection. Then we dive into the final result.

## I.1 RESULTS FOR ONLINE LINEAR REGRESSION (FROM (LI ET AL., 2016))

Even though our linear contextual bandit setup is different from e.g., (Lagrée et al., 2016) for ranking, the availability of the feedback $e_{t,i}$, which tells us whether item $i$ has been exposed, makes the analysis of the online linear regression similar to the general setup of linear bandits. Our approach builds on the confidence intervals developed by Li et al. (2016), which expands the analysis of confidence ellipsoids for linear regression of Abbasi-Yadkori et al. (2011) to cascade user models in rankings.

Each $c_{t,i}$ is $\frac{1}{2}$-subgaussian (because Bernoulli), and is conditionally independent of the other random variables conditioned and on $e_{t,i}$ and $x_{t,i}$. The incremental linear regression of line 7 of Algorithm 5 is the same as (Abbasi-Yadkori et al., 2011). Our observation model satisfies the conditions of the analysis of confidence ellipsoids of Li et al. (2016), from which we obtain:

**Lemma 26** *Under the probabilistic model described in Section 4, and under Assumption C. Let $\delta' > 0$ and $\lambda \geq D_{\mathcal{X}}^2 \overline{k}$, and let*

$$\alpha_T(\delta') = \frac{1}{2}\sqrt{\ln\left(\frac{\det(V_T)}{V_0 \delta'^2}\right)} + \sqrt{\lambda}D_\theta. \tag{98}$$

*Then, under Assumption C and with the notation of Algorithm 5, we have:*

- *((Li et al., 2016, Lemma 4.2)) with probability $\geq 1 - \delta'$, for all $T \geq 0$, $\theta$ lies in the confidence ellipsoid:*

$$C_T = \{\tilde{\theta} \in \mathbb{R}^d : \|\hat{\theta}_T - \tilde{\theta}\|_{V_T} \leq \alpha_T(\delta')\} \tag{99}$$

- *((Li et al., 2016, Lemma 4.4)):*

$$\alpha_T(\delta') \leq \frac{1}{2}\sqrt{2\ln\left(\frac{1}{\delta'}\right) + d\ln\left(1 + \frac{T D_{\mathcal{X}}^2 \overline{k}}{\lambda d}\right)} + \sqrt{\lambda}D_\theta. \tag{100}$$

These results stem from (Li et al., 2016, Lemma A.4 and A.5) that claim that, with the assumptions of Lemma 26, the following inequality holds with probability 1:

$$\sum_{t=1}^{T}\sum_{i=1}^{m}\|x_{t,i}\|_{V_{t-1}^{-1}}^2 e_{t,i} \leq 2\ln\frac{\det V_T}{\det(V_0)} \leq 2d\ln\left(1 + \frac{T D_{\mathcal{X}}^2 \overline{k}}{\lambda d}\right). \tag{101}$$

Notice that terms equivalent to $D_{\mathcal{X}}$ and $D_\theta$ do not appear in (Li et al., 2016) because they assume they are $\leq 1$. The $D_{\mathcal{X}}^2$ term comes from a modification necessary in (Li et al., 2016, Lemma A.4) while $D_\theta$ is required by the initial confidence bound proved by Abbasi-Yadkori et al. (2011). The term $\overline{k}$ plays the constant $C_\gamma$ of (Li et al., 2016).

## I.2 GUARANTEES FOR FW-LINUCB

We start by writing an alternative to Assumption D for the case where $f$ is not smooth to carry out our analysis with as little assumptions on $f$ as possible:

**Assumption $\mathbf{D}'$** *The assumptions of the framework of Sec. 4 hold, as well as Ass. E. Moreover, $\forall t \in \mathbb{N}, \forall z \in \mathcal{K} \frac{\partial f_t}{\partial z_{m+1}}(z) > 0$, and $\forall x \in \mathcal{X}, 1 \geq b_1(x) \geq \ldots \geq b_{\overline{k}}(x) = \ldots = b_m(x) = 0$.*

**Lemma 27** *Under Assumptions $D'$ and $C$ Let $T > 0, \delta' > 0$ and $\lambda \geq D_{\mathcal{X}}^2 \overline{k}$. Then for every $\delta' > 0$, Algorithm 5 satisfies, with probability at least $1 - \delta'$:*

$$R_T^{\text{gen}} \leq 2L\alpha_T(\delta'/3)\sqrt{T\overline{k}}\left(\sqrt{2\ln(\frac{3}{\delta'})} + \sqrt{2d\ln\left(1 + \frac{T D_{\mathcal{X}}^2 \overline{k}}{\lambda d}\right)}\right) + LD_{\mathcal{K}}\sqrt{2T\ln\frac{3}{\delta'}}.$$

*where $\alpha_T$ is defined in Lemma 26.*

*Proof.* Let $g_t = \nabla f_{t-1}(\hat{s}_{t-1})$, and $a_t^* \in \operatorname{argmax}_{a \in \mathcal{A}} \langle g_t \,|\, \mu(x_t)a - r_t \rangle$. Let furthermore $\delta' > 0$. Assume the algorithm uses $\alpha_t(\delta'/3)$, so that $C_t = \{\tilde{\theta} \in \mathbb{R}^d : \|\hat{\theta}_t - \tilde{\theta}\|_{V_t} \leq \alpha_t(\delta'/3)\}$.

Let us define $\hat{\mu}_t$ similarly to Proposition 4, i.e., $\forall t \in \mathbb{N}_*$, $\hat{\mu}_t$ such that $\forall i \in [\![m]\!]$, $\hat{\mu}_{t,i} = \mu_i(x_t)$ and $\hat{\mu}_{t,m+1} = \hat{v}_t b(x_t)^\mathsf{T}$ viewed as a column vector, with $\hat{v}$ defined in line 3 of Algorithm 5. We have:

$$\sum_{t=1}^{T} \max_{a \in \mathcal{A}} \langle g_t \,|\, \mu(x_t)a - r_t \rangle = \sum_{t=1}^{T} \underbrace{\langle g_t \,|\, \mu(x_t)a_t^* - \hat{\mu}_t a_t \rangle}_{:=A_t} + \sum_{t=1}^{T} \underbrace{\langle g_t \,|\, \hat{\mu}_t a_t - \mu(x_t)a_t \rangle}_{:=B_t}$$

$$+ \sum_{t=1}^{T} \underbrace{\langle g_t \,|\, \mu(x_t)a_t - r_t \rangle}_{:=X_t}$$

**Step 1: Upper bound on $\sum_{t=1}^{T} A_t$ via optimism**  Let $t \geq 0$. For $\tilde{\theta} \in \mathbb{R}^d$, denote $\mu_{\tilde{\theta}}(x) \in \mathbb{R}^{D \times K}$ (recall $D = m + 1$), the average reward function where parameters $\tilde{\theta}$ replace $\theta$. We first show that for every $a \in \mathcal{A}$, we have $\max_{\tilde{\theta} \in C_t} \langle g_t \,|\, \mu_{\tilde{\theta}}(x_t)a \rangle \leq \langle g_t \,|\, \hat{v}_t a \rangle$, where $\hat{v}_t$ is given in Line 3 of Algorithm 5.

Given $a \in \mathcal{A}$, let us denote by $\operatorname{mat}(a)$ the view of $a$ as an $m \times m$ permutation matrix (instead of an $m^2$-dimensional column vector). Recalling that $x_t$ is a $m \times d$ matrix and $g_t \in \mathbb{R}^{m+1}$, let us denote by $g_{t,1:m}$ the vector containing the first $m$ dimensions of $g_t$. We have:

$$\langle g_t \,|\, \mu_{\tilde{\theta}}(x_t)a \rangle = g_{t,1:m}^\mathsf{T} \operatorname{mat}(a)b(x_t) + g_{t,m+1} \tilde{\theta}^\mathsf{T} x_t^\mathsf{T} \operatorname{mat}(a)b(x_t), \tag{102}$$

therefore:

$$\max_{\tilde{\theta} \in C_t} \langle g_t \,|\, \mu_{\tilde{\theta}}(x_t)a \rangle = g_{t,1:m}^\mathsf{T} \operatorname{mat}(a)b(x_t) + g_{t,m+1} \max_{\tilde{\theta} \in C_T} \left( g_{t,m+1} \tilde{\theta}^\mathsf{T} x_t^\mathsf{T} \operatorname{mat}(a)b(x_t) \right) \tag{103}$$

$$\leq g_{t,1:m}^\mathsf{T} \operatorname{mat}(a)b(x_t) + g_{t,m+1} \hat{v}_t^\mathsf{T} \operatorname{mat}(a)b(x_t) = \langle g_t \,|\, \hat{\mu}_t a \rangle. \tag{104}$$

The first equality is because $g_{t,m+1} \geq 0$. The second equality is deduced by direct calculation from the definition of $C_t$ in Lemma 26, which gives $\hat{v}_{t,i} = \max_{\tilde{\theta} \in C_t} \tilde{\theta}^\mathsf{T} x_{t,i}$.

By Proposition 4 we have that $a_t$ defined at Line 4 of Algorithm 5 maximizes $\langle g_t \,|\, \hat{\mu}_t a \rangle$ over $a$. We thus have $\max_{a \in \mathcal{A}} \max_{\tilde{\theta} \in C_t} \langle g_t \,|\, \mu_{\tilde{\theta}}(x_t)a \rangle \leq \langle g_t \,|\, \hat{\mu}_t a_t \rangle$.

By Lemma 26, we have $\theta \in C_t$ for all $t \geq 0$ with probability $1 - \delta'/3$. Therefore, with probability $1 - \delta'/3$, we have for all $t \geq 0$: $\langle g_t \,|\, \mu_\theta(x_t)a_t^* \rangle \leq \langle g_t \,|\, \hat{\mu}_t a_t \rangle$. Noting that $\mu_\theta(x_t) = \mu(x_t)$ by definition of $\theta$, we obtain that $\forall t, A_t \leq 0$ and thus $\sum_{t=1}^{T} A_t \leq 0$ with probability $1 - \delta'/3$.

**Step 2: Upper bound on $\sum_{t=1}^{T} B_t$ using linear bandit techniques**  Let $a_{t,i} \in \mathbb{R}^m$ denote the $i$-th row of $\operatorname{mat}(a_t)$, which contains only 0s except a 1 at the rank of item $i$ in $a$. Since $\hat{\mu}_t$ and $\mu(x_t)$ only differ in the last dimension, which is the user utility, we have, using (102):

$$B_t = g_{t,m+1} \left( (\hat{v}_t - v(x_t))^\mathsf{T} \operatorname{mat}(a_t)b(x_t) \right) = g_{t,m+1} \sum_{i=1}^{m} \left( \hat{v}_{t,i} - v_i(x_t) \right) a_{t,i}^\mathsf{T} b(x_t) \tag{105}$$

Denoting $\bar{e}_{t,i} = a_{t,i}^\mathsf{T} b(x_t) \in \mathbb{R}$ the expected exposure of item $i$ in ranking $a_t$ given context $x_t$, we have:

$$B_t = \underbrace{g_{t,m+1}}_{\in [0,L]} \sum_{i=1}^{m} \left( \hat{v}_{t,i} - v_i(x_t) \right) \bar{e}_{t,i} \leq L \sum_{i=1}^{m} \left( (\hat{\theta}_{t-1} - \theta)^\mathsf{T} x_{t,i} + \alpha_t(\delta'/3) \|x_{t,i}\|_{V_{t-1}^{-1}} \right) \bar{e}_{t,i} \tag{106}$$

$$\leq L \sum_{i=1}^{m} \left( \|\hat{\theta}_{t-1} - \theta\|_{V_{t-1}} \|x_{t,i}\|_{V_{t-1}^{-1}} + \alpha_t(\delta'/3) \|x_{t,i}\|_{V_{t-1}^{-1}} \right) \bar{e}_{t,i} \qquad \text{(by Cauchy-Schwarz)}$$

By Lemma 26, we have, with probability $1 - \delta'/3$: $\|\hat{\theta}_{t-1} - \theta\|_{V_{t-1}} \leq \alpha_t(\delta'/3)$, and thus:

$$B_t \leq 2L\alpha_t\left(\frac{\delta'}{3}\right) \sum_{i=1}^{m} \|x_{t,i}\|_{V_{t-1}^{-1}} \overline{e}_{t,i} \tag{107}$$

$$= 2L\alpha_t\left(\frac{\delta'}{3}\right) \left( \underbrace{\left(\sum_{i=1}^{m} \|x_{t,i}\|_{V_{t-1}^{-1}} (\overline{e}_{t,i} - e_{t,i})\right)}_{X'_t} + \left(\sum_{i=1}^{m} \|x_{t,i}\|_{V_{t-1}^{-1}} e_{t,i}\right) \right) \tag{108}$$

We first deal with the sum over $t$ of the right-hand side, using $e_{t,i} \in \{0,1\}$:

$$\sum_{t=1}^{T} \sum_{i=1}^{m} \|x_{t,i}\|_{V_{t-1}^{-1}} e_{t,i} = \sum_{t=1}^{T} \sum_{i=1}^{m} (\|x_{t,i}\|_{V_{t-1}^{-1}} e_{t,i}) \times e_{t,i} \tag{109}$$

$$\leq \sqrt{\sum_{t=1}^{T} \sum_{i=1}^{m} e_{t,i}^2} \sqrt{\sum_{t=1}^{T} \sum_{i=1}^{m} (\|x_{t,i}\|_{V_{t-1}^{-1}}^2 e_{t,i}^2)} \qquad \text{(by Cauchy-Schwarz)}$$

$$\leq \sqrt{T\overline{k}} \sqrt{d\ln\left(1 + \frac{TD_{\mathcal{X}}^2 \overline{k}}{\lambda d}\right)}. \qquad \text{(by 101)}$$

For the left-hand term, we have that $\left(\sum_{t=1}^{T} X'_t\right)_{T \in \mathbb{N}_*}$ is a martingale adapted to the filtration $\overline{\mathfrak{F}} = (\overline{\mathfrak{F}}_T)_{T \in \mathbb{N}_*}$ where $\overline{\mathfrak{F}}_T$ is the $\sigma$-algebra generated by $(x_1, a_1, r_1, \ldots, x_{T-1}, a_{T-1}, r_{T-1}, x_T, a_T)$, with $|X'_t| \leq \frac{D_{\mathcal{X}}\overline{k}}{\sqrt{\lambda}}$. Thus, with probability at least $1 - \delta'/3$, we have

$$\sum_{t=1}^{T} \sum_{i=1}^{m} \sum_{i=1}^{m} \|x_{t,i}\|_{V_{t-1}^{-1}} (\overline{e}_{t,i} - e_{t,i}) \leq \frac{D_{\mathcal{X}}\overline{k}}{\sqrt{\lambda}} \sqrt{2T \ln\frac{3}{\delta'}} \leq \sqrt{2T\overline{k} \ln\frac{3}{\delta'}}. \tag{110}$$

Where the last inequality comes from the assumption $\lambda \geq D_{\mathcal{X}}^2 \overline{k}$. We conclude this step by saying that with probability $1 - 2\delta'/3$, we have:

$$\sum_{t=1}^{T} B_t \leq 2L\alpha_t\left(\frac{\delta'}{3}\right) \sqrt{T\overline{k}} \left( \sqrt{2\ln\frac{3}{\delta'}} + \sqrt{d\ln\left(1 + \frac{TD_{\mathcal{X}}^2 \overline{k}}{\lambda d}\right)} \right). \tag{111}$$

**Step 3: Upper bound on $\sum_{t=1}^{T} X_t$ using Azuma's inequality** Following the same arguments as in the proof of Thm. 19, let $\overline{\mathfrak{F}} = (\overline{\mathfrak{F}}_t)_{t \in \mathbb{N}_*}$ be the filtration where $\overline{\mathfrak{F}}_t$ is the $\sigma$-algebra generated by $(x_1, a_1, r_1, \ldots, x_{t-1}, a_{t-1}, r_{t-1}, x_t, a_t)$. Then $(X_t)_{t \in \mathbb{N}}$ is a martingale difference sequence adapted to $\overline{\mathfrak{F}}$ with $|X_t| \leq LD_{\mathcal{K}}$, so that $\sum_{t=1}^{T} X_t \leq L\sqrt{2T\overline{k}\ln\frac{3}{\delta'}}$ with probability $1 - \delta'/3$.

The final result is obtained using a union bound, considering that Step 1 and Step 2 use the same confidence interval given by Lemma 26 which is valid w.p. $\geq 1 - \delta'/3$, Step 2 uses an addition Azuma inequality valid w.p. $1 - \delta'/3$, and step 3 uses an additional Azuma inequality which valid with probability $\geq 1 - \delta'/3$. $\qquad\square$

**Theorem 5** *Under Assumptions B, C and D, for every $\delta' > 0$, every $T \in \mathbb{N}_*$, every $\lambda \geq D_{\mathcal{X}}^2 \overline{k}$, with probability at least $1 - \delta'$, Algorithm 1 has scalar regret bounded by*

$$R_T^{\text{scal}} = O\left( L\sqrt{T\overline{k}} \sqrt{d\ln(T/\delta')} \left( \sqrt{d\ln(T/\delta')} + D_\theta\sqrt{\lambda} + \sqrt{\overline{k}/d} \right) \right). \tag{13}$$

*Thus, considering only $d, T, \overline{k}$ and $\delta = \delta'$ Alg. 1 has regret $R_T \leq O\left(\frac{d\overline{k}\ln(T/\delta)}{\sqrt{T}}\right)$ w.p. at least $1 - \delta$.*

*Proof.* Let $\delta > 0$ and use $\delta' := 3\delta/4$ and $\delta := \delta/4$ in the bound on $R_T$ obtained by applying Lemma 27 and Theorem 11. Notice that Using $\lambda \geq D_{\mathcal{X}}^2 \overline{k}$ and $D_{\mathcal{K}} = O(\overline{k})$, we have:

$$\overline{R}^{\text{scal}}(T, 3\delta/4) = O\left( L\alpha_T(\delta)\sqrt{T\overline{k}d\ln(T/\delta)} + L\overline{k}\sqrt{T\ln(1/\delta)} \right) \tag{112}$$

and $\alpha_T(\delta) = O\left(\sqrt{d\ln(T/\delta)} + D_\theta\sqrt{\lambda}\right)$.

We thus get

$$\overline{R}^{\text{scal}}(T,\delta) = O\left(L\sqrt{T\overline{k}}\sqrt{d\ln(T/\delta)}\left(\sqrt{d\ln(T/\delta)} + D_\theta\sqrt{\lambda}\right) + L\overline{k}\sqrt{T\ln(1/\delta)}\right) \quad (113)$$

$$= O\left(L\sqrt{T\overline{k}}\sqrt{d\ln(T/\delta)}\left(\sqrt{d\ln(T/\delta)} + D_\theta\sqrt{\lambda} + \sqrt{\overline{k}/d}\right)\right) \quad (114)$$

For the smooth case, the total bound adds $O(L\overline{k}\sqrt{T\ln(1/\delta)} + \tilde{C}\frac{\ln T}{T})$. A bound on the complete regret is thus

$$R_T = O\left(L\sqrt{T\overline{k}}\sqrt{d\ln(T/\delta)}\left(\sqrt{d\ln(T/\delta)} + D_\theta\sqrt{\lambda} + \sqrt{\overline{k}/d} + \tilde{C}\frac{\ln T}{T}\right)\right) \quad (115)$$

$\square$

## J  ADDITIONAL TECHNICAL LEMMAS

### J.1  PROOF OF LEMMA 7 ($\mathcal{S}$ IS COMPACT)

**Lemma 7** *Under Assumption $\tilde{A}$, $\mathcal{S}$ is compact and $\forall T \in \mathbb{N}_*, \forall x_{1:T} \in \mathcal{X}^T, \mathcal{S}(x_{1:T})$ is compact.*

*Proof.* We start with $\mathcal{S}(x_{1:T})$. Let $x_{1:T} \in \mathcal{X}^T$. We notice that $\mathcal{S}(x_{1:T})$ is the image of $\overline{\mathcal{A}}^T$ by the continuous mapping $\phi : (\mathbb{R}^K)^T \to \mathbb{R}^D$ defined by $\phi(a_1, ..., a_T) = \frac{1}{T}\sum_{t=1}^{T}\mu(x_t)a_t$. Since $\overline{\mathcal{A}}$ is compact, $\overline{\mathcal{A}}^T$ is compact as well. $\mathcal{S}(x_{1:T})$ is thus the image of a compact set by a continuous function, and is therefore compact.

For the set $\mathcal{S}$, we provide a proof here using Diestel's theorem (see (Yannelis, 1991)). Consider the set-valued map defined by $G : \mathcal{X} \to \{B \mid B \subseteq \mathbb{R}^D\}$

$$G(x) := \mu(x)\overline{\mathcal{A}} := \{\mu(x)\overline{a} \mid \overline{a} \in \overline{\mathcal{A}}\}. \quad (116)$$

Then, $\mathcal{S}$ can be written as the *Aumann integral* of $G$ over $\mathcal{X}$ w.r.t $P$, i.e.

$$\mathcal{S} = \int_{\mathcal{X}} G\,\mathrm{d}P := \left\{\int_{\mathcal{X}} g\,\mathrm{d}P \,\Big|\, g \in \mathcal{G}\right\}, \quad (117)$$

where $\mathcal{G} \subseteq L^1(\mathcal{X}, P)$ is the collection of all $P$-integrable selections of $G$, i.e. the collection of all $P$-integrable functions $g : \mathcal{X} \to \mathbb{R}$ such that $g(x) \in G(x)$ for $P$-a.e $x \in \mathcal{X}$.

Now, since $\overline{\mathcal{A}}$ is compact, convex and nonempty, the values of the set-valued function $G$ are nonempty, convex, and compact. Moreover, since $\sup_{x \in \mathcal{X}, a \in \overline{\mathcal{A}}}\|\mu(x)a\|_2 < +\infty$ because $\forall x, a, \mu(x)a \in \mathcal{K}$, the set-valued function $G$ is $P$-integrably bounded in the sense of (Yannelis, 1991, Section 2.2). It then follows from *Diestel's Theorem* (Yannelis, 1991, Theorem 3.1) that the collection $\mathcal{G}$ of $P$-integrable selections of $G$ is weakly compact in $L^1(\mathcal{X}, P)$. Finally, since $g \mapsto \int_{\mathcal{X}} g\,\mathrm{d}P$ is a weakly continuous mapping from $L^1(\mathcal{X}, P)$ to $\mathbb{R}^D$, and $\mathcal{S} \subseteq \mathbb{R}^D$ is the image of $\mathcal{G}$ under this mapping (refer to the correspondence (117)), we deduce that $\mathcal{S}$ is weakly compact as a subset of $\mathbb{R}^D$, and therefore compact since $\mathbb{R}^D$ is finite-dimensional. $\square$

### J.2  PROOF OF LEMMA 24

**Lemma 24** *Let $(\lambda_t)_{t \in \mathbb{N}} \in \mathbb{R}_+^T$ be a sequence of non-negative numbers, denote $\Lambda_T = \sum_{t=1}^{T}\lambda_t$ and let $(\overline{\Lambda}_T)_{T \in \mathbb{N}}$ such that $\forall T \in \mathbb{N}, \overline{\Lambda}_T > 0$ and $\overline{\Lambda}_T \geq \Lambda_T$.*

$$\sum_{t=1}^{T}\frac{\lambda_t}{\sqrt{\overline{\Lambda}_t}} \leq 2\sqrt{\overline{\Lambda}_T}. \quad (91)$$

*Proof.* First, we treat the case where $\lambda_0 > 0$. Then $\forall_t \in [\![T]\!], \Lambda_t > 0$. We thus have

$$\sum_{t=1}^{T} \frac{\lambda_t}{\sqrt{\overline{\Lambda}_t}} \leq \sum_{t=1}^{T} \frac{\lambda_t}{\sqrt{\Lambda_t}} \tag{118}$$

We now prove that the right-hand term is $\leq \sqrt{\Lambda_T}$. Let us observe that, for every $\alpha \geq 0, \beta > \alpha$:

$$\frac{1}{2} \frac{\alpha}{\sqrt{\beta}} \leq \sqrt{\beta} - \sqrt{\beta - \alpha}, \tag{119}$$

which is proved using $\sqrt{\beta} - \sqrt{\beta - \alpha} = \int_{\beta-\alpha}^{\beta} \frac{1}{2\sqrt{s}} ds \geq \alpha \frac{1}{2\sqrt{\beta}}$. Using the telescoping sum (with $\Lambda_0 = 0$):

$$\sum_{t=1}^{T} \frac{\lambda_t}{\sqrt{\Lambda_t}} \leq 2 \sum_{t=1}^{T} \left( \sqrt{\Lambda_t} - \underbrace{\sqrt{\Lambda_t - \lambda_t}}_{=\Lambda_{t-1}} \right) = 2\sqrt{\Lambda_T} \leq 2\sqrt{\overline{\Lambda}_T}, \tag{120}$$

we obtain the desired result.

More generally, if $\lambda_0 = 0$, there are two cases:

1. if $\forall_T \in [\![T]\!], \lambda_t = 0$ then the result is true;
2. otherwise, let $T_0 = \min\{t \in [\![T]\!] : \lambda_t > 0\}$. Using the result above, we have:

$$\sum_{t=1}^{T} \frac{\lambda_t}{\sqrt{\overline{\Lambda}_t}} = \sum_{t=T_0}^{T} \frac{\lambda_t}{\sqrt{\overline{\Lambda}_t}} \leq 2\sqrt{\overline{\Lambda}_T}. \tag{121}$$

$\square$

