# OpenReview forum: "Contextual bandits with concave rewards, and an application to fair ranking"
_ICLR.cc/2023/Conference — ICLR 2023 poster_

### Official Review · Reviewer_4Gmp · 2022-10-22

**Confidence:** 3
**Correctness:** 4
**Technical Novelty And Significance:** 3
**Empirical Novelty And Significance:** 3
**Recommendation:** 8

**Clarity, Quality, Novelty And Reproducibility:**

Clarity gets an A+.  Authors cover a *lot* of territory while managing to retain intelligibility and hit the page limit.  Kudos.

Quality and originality also gets an A+.  This is clear improvement over Agarwal 2016.

**Strength And Weaknesses:**

Strengths:
  * Multi-objective criterion arise frequently in practice.
  * Reduction-based algorithm design maximizes reuse of existing and novel componentry and overall utility of approach.
     * e.g., the discussion mentions constraints (good: reduction to CB-knapsack algos might work).
  * Figure 1 (right) is particularly persuasive.

Weaknesses;
  * FW-LinUCB will give you nice theorems, but composing with the infinite action linear version of square cb[2] will give you an algorithm people will actually want to deploy.
    * We never use UCB algos in production here because they are not robust to reward feedback delay.
    * You might also find interesting a composition with a smoothed regret variant of square cb for infinite action spaces [3].
  * It's not clear how to elicit the trade-off function in practice.
    * For example, does the Lin et. al.[1] elicitation technique yield a concave tradeoff function (or, e.g., would preference elicitation be viable over a constrained model class like generalized gini)?

[1] https://arxiv.org/abs/2203.11382

[2] https://arxiv.org/abs/2207.05836

[3] https://arxiv.org/abs/2207.05849

**Summary Of The Paper:**

Authors provide the first algorithms for provably vanishing regret for Contextual Bandits with Concave Rewards.

**Summary Of The Review:**

Exemplary work.

Confidence is just a 3 because I didn't check the proofs.  If I get time I'll circle back to them.  "Math/other details were not carefully checked."

---

> ### Author Response · Authors · 2022-11-10
> **Response to reviewer 4Gmp**
>
> Thanks a lot for your review.
>
> > composing with the infinite action linear version of square cb[2]
>
> Thank you for bringing to us these recent improvements on SquareCB with infinite action spaces [2,3]. These works are definitely relevant to us because we do not assume a finite number of arms (we only require $\mathcal{A}$ to be compact). The strength of our reduction is that it makes it possible to transform new contextual bandit algorithms into new algorithms for CBCR. We updated the discussion in Section 6 to reflect that.
>
>
> > It's not clear how to elicit the trade-off function in practice. For example, does the Lin et. al.[1] elicitation technique  [...] ?
>
> Thank you for this comment, we added a discussion on the elicitation of $f$ in Section 6 (in green). In the paper, we focused on applications where the designer choses $f$, especially in fair ranking (Section 4). In practice, the designer may choose $f$ within a small set of candidates in offline experiments (for example, varying $\beta$ in a small grid in the example of Eq. 2). Other approaches could be used to choose the hyperparameters that define $f$, e.g.:
> * the preference elicitation method of Lin et al. [1] that you suggest, which applies to some classes of concave functions (see [1, App. E.2]),
> * the algorithm of [Bourdache and Perny, 2019] for the Generalized Gini class.
>
> These methods do not consider learning the rewards themselves, so the combination of these works with the bandit setting is an interesting question for future work.
>
> [Bourdache and Perny; 2019] Nadjet Bourdache, Patrice Perny. Active Preference Learning based on Generalized Gini Functions: Application to the Multiagent Knapsack Problem. Thirty-Third AAAI Conference on Artificial Intelligence (AAAI 2019).

---

### Official Review · Reviewer_Gk2g · 2022-10-24

**Confidence:** 3
**Correctness:** 4
**Technical Novelty And Significance:** 2
**Empirical Novelty And Significance:** 3
**Recommendation:** 6

**Clarity, Quality, Novelty And Reproducibility:**

This work has good quality in writing and clarity. I have no problem in understanding the ideas. The novelty and contributions are decent to be published.

**Strength And Weaknesses:**

Strengths:
* This paper proposes an interesting method to reduce CBCR problem to a studied problem, being the first to remove restrictions on the policy space.
* The reduction method has some generality to cover a class of applications that can be modeled by CBCR framework, such as the fair ranking problem mentioned in the paper.

Weaknesses:

Overall, I believe this is a high-quality paper, which has some solid contributions that may benefit future works. Here, I have some minor suggestions and questions:
* In the ``setting'' paragraph in Section~2, the reward is defined as a noisy multi-dimensional reward. I would suggest authors to further declare the noise, e.g., the support of the random noise; the restrictions on the distribution; is it heavy tailed? As the distribution of the noise can heavily affect the policy design, even it may have zero mean. Or, if the only requirement of the noise is zero mean in this work, please also mention that.
* Throughout the paper, the concave function $f$ on the reward is assumed as known by the learner, and the reduction also requires the knowledge of $f$. What if $f$ is unknown so that $f(\sum_t r_t)$ is unknown? How much difficulty will be brought to address the CBCR problem with an unknown $f$? As I think an unknown $f$ is more commonly seen in real world. Or, what will be the case if both the reward $r_t$ and $f$ are unknown and we only know $f(\sum_t r_t)$? Nonetheless, both alternative settings can be considered in future works.

**Summary Of The Paper:**

This paper considers contextual bandits with multi-dimensional rewards, with a known concave function that transforms the cumulative multi-dimensional rewards to a one-dimensional value to be maximized. This paper proposes a method to reduce the CBCR problem to scalar-reward bandits which has been extensively studied, so that no restrictions necessary on the policy space. Also, a smoothing technique is provided in case of non-smooth concave function $f$. Then, the paper introduces an application using their reduction method, which focuses on contextual ranking bandits with fairness-aware objectives, and shows that their method is efficient with regret guarantee. Experimental evaluations are presented.

**Summary Of The Review:**

Overall, I enjoyed reading this work, and I learnt interesting results from it. The idea of the reduction method is novel and general. Related literature is well-discussed in Introduction section.

---

> ### Author Response · Authors · 2022-11-10
> **Response to reviewer Gk2g**
>
> Thank you for your review.
>
> > I would suggest authors to further declare the noise, e.g., the support of the random noise; the restrictions on the distribution
>
> The noise has zero mean (by definition of $\mu$) and bounded support (since rewards are bounded, see Assumption A). It is possible to extend our approach to subgaussian rewards, using e.g., variants of Azuma’s inequality [Shamir, 2011] and making appropriate changes, but it comes at the expense of more technicalities. As in e.g, SquareCB [Foster and Rakhlin, 2020] and multi-objective bandits [Busa-Fekete et al., 2017], we chose to use bounded rewards to keep the presentation clear and focused on the novel aspects.
>
>
> > What if $f$ is unknown so that $f(\sum_t r_t)$ is unknown? […] I think an unknown $f$ is more commonly seen in real world.
>
> Our main motivation for CBCR comes from the problem of fairness in rankings, where various fairness objectives $f$ have been formulated explicitly [Singh and Joachims, 2019; Morik et al, 2020; Biega et al 2018; Zehlike and Castillo, 2018; Do et al 2021] (see also Example 2). CBCR is the framework that allows to incorporate fairness trade-offs in the contextual bandit setting. In Example 1, we give another example application of CBCR derived from music recommendation at Spotify [Mehrotra et al., 2020]: the designer specifies a desired trade-off between the multiple recommendation metrics, which amounts to defining an explicit form for $f$. In practice, the designer may experiment with a small set of possible $f$ (e.g., by varying $\beta$ in a small grid in the example of Eq. 2). We added a discussion on this in Section 6 (in green), see also our response to reviewer 4Gmp.
>
> In bandits with concave rewards (BCR), the trade-off between the rewards is defined by a known function $f$ [Agrawal and Devanur 2014; Agrawal et al. 2016; Busa-Fekete et al, 2017; Mehrotra et al. 2020]. CBCR with unknown $f$ is an interesting venue of investigation, but it may add a significant level of difficulty. In [Berthet and Perchet, 2017] and in bandit convex optimization, $f$ is unknown, but these approaches do not extend to CBCR, because explicit search in policy space is not possible in our case (as discussed in the intro). We added a clarification on the relationship between BCR, CBCR, bandit convex optimization and multi-objective optimization in Appendix A (in green).
>
> [Shamir, 2011] Ohad Shamir. A variant of azuma’s inequality for martingales with subgaussian tails. arXiv preprint arXiv:1110.2392, 2011.

---

> > ### Comment · Reviewer_Gk2g · 2022-11-17
> > **Thanks for the Response**
> >
> > The explanations address my questions. I will keep my rating.

---

### Official Review · Reviewer_joWt · 2022-10-27

**Confidence:** 3
**Correctness:** 3
**Technical Novelty And Significance:** 1
**Empirical Novelty And Significance:** 2
**Recommendation:** 5

**Clarity, Quality, Novelty And Reproducibility:**


The paper refers to $\mu(x_t)a_t$ as "reward" and defines regret as the difference between $f(\text{reward})$ and the best $f(\text{reward})$. This is different from the classical way of defining regret as the difference of reward and may confuse some readers.

Apart from this, the paper is generally well-written and I did not find any apparent typos in main text. Some LaTeX errors in appendix need be fixed, e.g., Lemma ?? in page 25.

The paper appears to supplement enough details for reproduce the results, including the code. I did not run the code, though.


**Strength And Weaknesses:**


### Strength

The paper provides a new setting, designs a new algorithm, proves its theoretical guarantee, and conducts experiments on real-world data to validate its performance. The paepr is rather complete for this problem.

### Weakness

By considering only the stationary context set and fixed action set, the paper is actually more restrictive than most contextual bandit literature that considers possibly adversarial context sets. In particular, the paper shall not claim at the end of Sec. 2 that it "subsumes classical contextual bandits." Also, the "reduction" in Sec. 3.2 is not that rigorous because LinUCB and SquareCB both work for adversarial contexts, meaning that the reduction cannot be understood as that the algorithm in the paper could replace LinUCB or SquareCB. Consequently, I am concerned on the impact of this paper over the bandits learning society.

The paper does not contain much technical novelty that could possibly inspire future work, as the proof seems to be a straightforward combination of Frank-Wolfe algorithm.


**Summary Of The Paper:**

This paper studies a contextual bandit problem with **stationary** context and **fixed** action set. At each step, the environment randomly selects a context $x$, then the player needs to choose an action $a$ that maximizes $f(\mu(x)a)$ where $f(\cdot)$ is a known concave function and $\mu$ is an unknown matrix-valued function. This paper solves this problem by nesting Frank-Wolfe algorithm with standard linear/contextual bandit algorithms.


**Summary Of The Review:**


The setting does not sound interesting or important to me. Besides, I have some concerns as prescribed in "Weakness" section.

---

> ### Author Response · Authors · 2022-11-10
> **Response to reviewer joWt**
>
> Thank you for your review.
>
> > the paper is actually more restrictive than most contextual bandit literature that considers possibly adversarial context sets.
>
>
> CBCR with adversarial contexts is fundamentally difficult, because it challenges the definition of an optimal policy and benchmark. In scalar contextual bandits, optimal policies are simply defined as taking the greedy action at the current context, whether it is stochastic or adversarial. [Langford and Zhang, 2007]. This challenge of CBCR is illustrated by limitations of existing work on BCR which either do not address contexts [Agrawal and Devanur, 2014; Berthet and Perchet, 2017] or restrict themselves to stochastic contexts and *finite policy sets* [Agrawal et al., 2016], a restriction that we remove.
>
>
> > the paper shall not claim at the end of Sec. 2 that it "subsumes classical contextual bandits."
>
> Thanks for pointing out this sentence, which was not precise enough. We made the sentence more precise in the updated document “subsumes classical *stochastic* contextual bandits”. Please note that we are clear throughout the text that we are dealing with stochastic contexts, while our reduction relies on “solving a contextual bandit problem with adversarial contexts and stochastic rewards” (Sec. 3.1, before Th. 2).
>
> > the reduction cannot be understood as that the algorithm in the paper could replace LinUCB or SquareCB.
>
> We would like to point out that the goal of the paper is certainly not to *replace* LinUCB or SquareCB. Rather, through our reduction, we *use* LinUCB or SquareCB as exploration/exploitation methods in our approach to solve corresponding CBCR problems (e.g., LinUCB if the average reward vector $\mu(x)$ is linear w.r.t. contexts),  while LinUCB and SquareCB are initially designed for scalar reward setups.
>
> > The paper does not contain much technical novelty that could possibly inspire future work. [...] The setting does not sound interesting or important to me.
>
> Our contributions are a general solution to CBCR and the first approach for contextual bandits in fair ranking, which are two results of independent interest.
>
> We would like to highlight that the objective functions we consider stem from the more applied literature on recommender systems [Mehrotra et al 2020], and especially the growing literature on *fair recommendation* [Morik et al 2020, Do et al 2021, Wu et al 2022]. Our paper shows that the (C)BCR framework is particularly relevant to these applications, a fact that has been overlooked by this more applied literature, while existing papers on BCR seem to mostly find their audience on the theory side. Related to how future work can build on our paper, we hope that it will increase the interest in the (C)BCR setup, both from the point of view of algorithmic or theoretical development (e.g., combinations with elicitation methods for the trade-off function as suggested by Reviewer 4Gmp) or the usage of these techniques in more applied research communities.
>
> From a technical point of view, the crucial step of our approach is to consider CBCR as an implicit optimization over the set of feasible rewards $\cal{S}$. This novel connection is made by Lemma 1, and leads to a reduction which has a relatively simple formulation. We believe that the simplicity of our solution is a strength: a simple combination of Frank-Wolfe and traditional contextual bandits offers a general solution to the CBCR problem -- including general reward functions and combinatorial action spaces. In particular, this provides enough flexibility to make new developments in contextual bandits readily applicable to CBCR, including the recent (or future) developments in SquareCB pointed out by Reviewer 4Gmp.

---

> > ### Author Response · Authors · 2022-11-16
> > **Did our response address your concerns?**
> >
> > Dear Reviewer joWt,
> >
> > We would be grateful if you could tell us whether our response effectively addressed your concerns, and if there are any remaining issues. Thank you in advance.

---

> > > ### Comment · Reviewer_joWt · 2022-11-24
> > > **Acknowledgement of Rebuttal**
> > >
> > > Thank the authors for the rebuttal. While I made assessment mainly from the perspective of bandits learning, I would consider the authors' claim that the paper merits by its solution to ranking problem.

---

### Official Review · Reviewer_aNts · 2022-10-31

**Confidence:** 2
**Correctness:** 3
**Technical Novelty And Significance:** 3
**Empirical Novelty And Significance:** Not applicable
**Recommendation:** 8

**Clarity, Quality, Novelty And Reproducibility:**

The paper talks about interesting and important topics related to the general concave bandit problem with good quality and clarity. The idea is novel as far as I know together with the techniques (not original but nice apply).


**Strength And Weaknesses:**

This paper is well written. The model (setting) in this paper is interesting, well motivated (based on the provided concrete examples) and more general compared to prior works. The geometric interpretation of CBCR (policy space spanned by a convex set) is a nice explanation and illuminates the optimization directions. This paper provides a nice and elegant reduction from CBCR regret to a scalar-reward bandit problem based on Frank-Wolfe analysis. This bypasses the core challenges in optimization in policy space. However, it is not clear  (at least to the reviewer) whether the obtained regret for CBCR (under this specific setting) is optimal or not. This paper lacks analysis and explanation in this aspect.


**Summary Of The Paper:**

This paper considers contextual bandits with concave rewards where the rewards are defined by a concave objective function with stochastic reward vectors. This setting removes the restrictions on the policy space compared to prior works. Under this setting, this paper proposes an (first) algorithm with the vanishing regret. The theoretical results and techniques could be applied to fair ranking problems. Experimental results also support the derived results.


**Summary Of The Review:**

Overall, I think that the studied model in this paper is well-motivated with more general problem setting. The proposed algorithm is sound  together with the related analysis. Therefore, I recommend “accept”.

---

> ### Author Response · Authors · 2022-11-10
> **Response to reviewer aNts**
>
> Thank you for your review.
>
> > it is not clear [...] whether the obtained regret for CBCR [...] is optimal or not.
>
> Thank you for this valuable remark. The worst-case regrets of multi-armed bandits [Bubeck and Cesa-Bianchi 2012] and contextual bandits with adversarial contexts are $\Omega(\sqrt{T})$ [Dani et al, 2008; Shamir 2015], which gives a lower bound for the worst-case regret of CBCR in $\Omega(\frac{1}{\sqrt{T}})$ (this was mentioned in App C.3). Our main regret bound for CBCR (Theorem 2) includes the regret of the associated scalar bandit algorithm $R_T^{\rm scal}$, plus residual terms in $O(\frac{1}{\sqrt{T}})$. In the worst-case analysis, the dependencies on the problem parameters are all directly derived from the regret bounds $R_T^{\rm scal}$ of the underlying scalar bandit algorithm (LinUCB, SquareCB, ...). Therefore we obtain CBCR algorithms that are near minimax optimal as soon as $R_T^{\rm scal} \leq O(\sqrt{T})$. The residual terms are tied to the use of Azuma’s inequality (Lemma 1) and FW analysis (using Lipschitz and parameters) and the dependencies to these parameters match usual convergence guarantees in optimization [Clarkson, 2008; Lan 2013].
>
> As we rely on a worst-case analysis in deriving our reduction guarantees, it remains an open question whether problem-dependent optimal bounds could be recovered as well.
>
> In the updated paper, we included this discussion in Section 3.1 and Appendix D (in green).

---

> > ### Comment · Reviewer_aNts · 2022-12-12
> > **thanks for the response**
> >
> > Thanks for the response. I will keep my initial rating after carefully reading the response.

---

### Author Response · Authors · 2022-11-10
**General response**

We thank all reviewers for their reviews. We re-uploaded the paper with a few changes (colored in green) based on the reviewers’ suggestions:
* [Reviewer aNts] In Section 3.1 and Appendix D, we added a discussion on minimax optimality,
* [Reviewer 4Gmp] In Section 6, we highlight that recent contextual bandit algorithms can be incorporated in our reduction-based approach,
* [Reviewer 4Gmp, Gk2g] In Section 6, we discuss how $f$ could be elicited in practice,
* [Reviewer Gk2g] In Appendix A, we clarified the relationship between BCR, CBCR, Bandit Convex Optimization and Multi-Objective Optimization.
* [Reviewer joWt] We also fixed minor typos in Section 2 and Appendix E.

---

### Decision · Program_Chairs · 2023-01-20

**Decision:**

Accept: poster

**Justification For Why Not Higher Score:**

The acceptance can be either as poster or spotlight. The paper could be potentially interesting to many audiences as it considers a general problem (contextual bandits with concave rewards) with popular applications such as fair ranking and beyond.

**Justification For Why Not Lower Score:**

All reviewers appreciate the novel and general problem setting and the theoretical analysis of proposed solutions with high ratings.

**Metareview: Summary, Strengths And Weaknesses:**

This paper considers a new problem: contextual bandits with concave rewards (CBCR) where the rewards are defined by a concave objective function with stochastic reward vectors.  The authors proposed a general reduction from CBCR to scalar-reward bandits and solved the problem by new algorithms combining Frank-Wolfe with existing contextual bandit algorithms.  The authors proved vanishing regret of the proposed solution. The authors also showed that CBCR can be applied to ranking problem with fairness-aware objectives.

Overall, the reviewers appreciate the problem setting and the theoretical analysis of proposed solutions. The application of ranking with fairness of exposure is also an important contribution. The authors are encouraged to include the discussion on novelty and limitation during the response period in the final version. I recommend acceptance.

**Note From Pc:**

if the above contains the word "oral" or "spotlight" please see: "oral" presentation means -> notable-top-5% and "spotlight" means -> notable-top-25%. As stated in our emails, we are disassociating presentation type from AC recommendations